# II-Bench: An Image Implication Understanding Benchmark for Multimodal Large Language Models

**Ziqiang Liu**[1,2]* **Feiteng Fang**[1,3]* **Xi Feng**[1,3]* **Xinrun Du**[4,14]* **Chenhao Zhang**[1,6]*
**Zekun Wang**[12,14] **Yuelin Bai**[1,2] **Qixuan Zhao**[1,3] **Liyang Fan**[1] **Chengguang Gan**[7]
**Hongquan Lin**[1,3] **Jiaming Li**[1,2] **Yuansheng Ni**[9] **Haihong Wu**[1,3] **Yaswanth Narsupalli**[5]
**Zhigang Zheng**[1] **Chengming Li**[10] **Xiping Hu**[10] **Ruifeng Xu**[11] **Xiaojun Chen**[8] **Min Yang**[1]
**Jiaheng Liu**[12] **Ruibo Liu**[13] **Wenhao Huang**[14] **Ge Zhang**[4,14,15]† **Shiwen Ni**[1]†

[1]Shenzhen Institutes of Advanced Technology, Chinese Academy of Sciences
[2]University of Chinese Academy of Sciences
[3]University of Science and Technology of China   [4]M-A-P   [5]IIT Kharagpur
[6]Huazhong University of Science and Technology   [7]Yokohama National University
[8]Shenzhen University   [9]Zhejiang University   [10]Shenzhen MSU-BIT University
[11]Harbin Institute of Technology (Shenzhen)   [12]Beihang University
[13]Dartmouth College   [14]01.ai   [15]University of Waterloo

## Abstract

The rapid advancements in the development of multimodal large language models (MLLMs) have consistently led to new breakthroughs on various benchmarks. In response, numerous challenging and comprehensive benchmarks have been proposed to more accurately assess the capabilities of MLLMs. However, there is a dearth of exploration of the higher-order perceptual capabilities of MLLMs. To fill this gap, we propose the **I**mage **I**mplication understanding **Bench**mark, **II-Bench**, which aims to evaluate the model's higher-order perception of images. Through extensive experiments on II-Bench across multiple MLLMs, we have made significant findings. Initially, a substantial gap is observed between the performance of MLLMs and humans on II-Bench. The pinnacle accuracy of MLLMs attains 74.8%, whereas human accuracy averages 90%, peaking at an impressive 98%. Subsequently, MLLMs perform worse on abstract and complex images, suggesting limitations in their ability to understand high-level semantics and capture image details. Finally, it is observed that most models exhibit enhanced accuracy when image sentiment polarity hints are incorporated into the prompts. This observation underscores a notable deficiency in their inherent understanding of image sentiment. We believe that II-Bench will inspire the community to develop the next generation of MLLMs, advancing the journey towards expert artificial general intelligence (AGI). II-Bench is publicly available at `https://huggingface.co/datasets/m-a-p/II-Bench`.

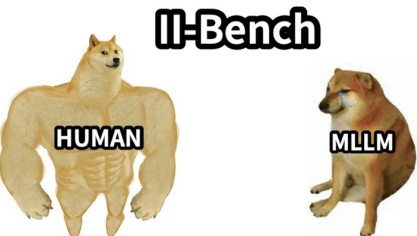

Figure 1: Implication: a significant gap exists between humans and MLLMs on II-Bench.

*Equal Contribution.
†Corresponding authors.

38th Conference on Neural Information Processing Systems (NeurIPS 2024) Track on Datasets and Benchmarks.

# 1 Introduction

In recent years, the development of Multimodal Large Language Models (MLLMs)[52, 12, 33, 66] has significantly advanced our ability to understand and generate content across various modalities, including text, images, and audio. Leveraging sophisticated architectures and vast amounts of data, MLLMs have demonstrated remarkable performance in image captioning[18, 24, 45], visual question answering[44, 51], video understanding and generation[46, 70], etc.

Nevertheless, comprehensively evaluating the performance of these models remains a challenge. While benchmarks exist for multimodality, such as ScienceQA[43], MMMU[67], there is a dearth of exploration of the higher-order perceptual capabilities[57] of MLLMs, which refer to nuanced emotional understanding and profound meaning extraction.

Philosopher Suzanne Langer once noted, "Art is the creation of forms symbolic of human feeling." This profoundly summarizes how images often embody human emotions and serve as a conduit for personal views and cultural narratives. Therefore, understanding the meaning of images requires not only meticulous observation but also an exploration of the human emotions and cultural contexts they reflect. In real life, many artworks, comics, and posters are imbued with rich meanings, and artists convey their insights to the audience through these works. These abstract and complex images pose a significant challenge for MLLMs, as the models must possess advanced higher-order perceptual capabilities to accurately understand the human emotions conveyed in the pictures and infer the deeper meanings the creators intend to express. The evaluation of high-order perception is crucial for artificial general intelligence (AGI) because it encompasses the ability to understand complex, non-literal aspects of visual information, much like humans do, which includes recognizing and processing implications, emotional cues, synesthesia or other cognitive features that go beyond mere object recognition or factual knowledge extraction [6, 55]. This implies that a machine's understanding of the world should also incorporate these implications to achieve a more human-like comprehension, which fosters a deeper, more nuanced understanding, serving as an important means of exploration from perceptual intelligence to cognitive intelligence.

However, an effective benchmark for higher-order perceptual capability measurement is notably absent in the current landscape. To fill this gap, we introduce **II-Bench**, a comprehensive benchmark designed to assess MLLMs' higher-order perceptual, reasoning and comprehension abilities. This holistic evaluation enables us to gain a deeper insight into the models' true capabilities, thereby fostering advancements in multimodal AI research.

As illustrated in Figure 2, II-Bench comprises 1,222 images, spanning six domains: life, art, society, psychology, environment and others. Furthermore, II-Bench encompasses diverse categories of images, including illustrations, memes, posters, comics, logos and paintings. By utilizing images from multiple domains and categories, the model's comprehension and reasoning abilities can be evaluated more objectively and comprehensively.

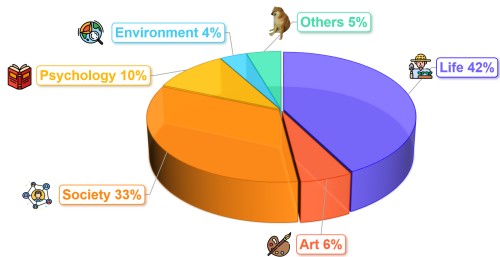

Figure 2: Composition of II-Bench.

We conduct extensive experiments to evaluate II-Bench on 20 MLLMs. Our main contributions and findings are as follows:

- We introduce II-Bench, the first Image Implication Understanding Benchmark, which is very challenging for current MLLMs.
- A significant difference exists in performance between humans and MLLMs: the highest accuracy achieved by the model is 74.8%, whereas the average accuracy for humans is 90%, with the highest reaching 98%.
- Closed-source models often outperform open-source ones, while the performance gap between the leading closed-source model and the leading open-source model is minimal, only about 1%.
- Models perform worse in domains containing abstract and complex information, such as Art and Psychology, compared to Environment, Life, Society and other domains.
- Incorporating additional emotional polarity information of images into prompts generally enhances model scores, indicating that models lack sufficient emotional understanding of images, leading to misinterpretation of implicit meanings.

Our aim with II-Bench is to evaluate MLLMs' higher-order perception of images. We believe that II-Bench will inspire the community to create the next generation of MLLMs, propelling us further on the path toward sophisticated artificial general intelligence (AGI).

## 2 Related Work

### 2.1 Multimodal Large Language Models

Given that advanced large language models (LLMs) exhibit sophisticated reasoning abilities, strong generality, and extensive world knowledge [48, 49], current multimodal LLMs (MLLMs) typically involve integrating additional modules to align non-textual modality features with the language space. For example, BLIP-2 [35] encodes images using ViT [16] and employs a Q-Former to map visual features into the language space. Similarly, LLaVA [38, 7, 9] utilizes an MLP as the connector between the visual encoder and the LLM backbone. These architectural designs not only incorporate visual representations into the LLMs but also preserve the advanced capabilities inherent to LLMs. Recent studies have demonstrated that current MLLMs are capable of understanding human minds, reasoning with scientific figures, etc. [4, 49], due to the success of unlocking the abilities of LLM backbones in multimodal settings. Nonetheless, despite the strong implication understanding abilities of LLMs [59], there is limited research on the implication understanding of images by current MLLMs, and our work addresses this gap for the first time.

### 2.2 MLLM Benchmarks

The evolution of MLLMs has underscored the importance of comprehensive evaluations within the research community. Initial benchmarks primarily targeted singular tasks, such as the visual question answering (VQA) task [2, 19, 29, 54, 27] and the image captioning task [36, 1, 50]. While notable achievements have been recorded on these benchmarks, they fall short of thoroughly evaluating MLLMs across the broader spectrum of multimodal perception and reasoning. To bridge this gap, recent studies have aimed at evaluating models from various perspectives [40, 32, 31, 62, 17, 43, 5, 71, 20]. For example, MMBench [40] and SEED [32, 31] explore models' capabilities through common-sense questions, featuring multiple-choice questions across various dimensions of ability. To assess specialized expertise, MMMU [67] and CMMMU [69] leverage content from exams and textbooks to enhance domain-specific knowledge evaluation.

However, MMStar [8] pointed out that the model can answer some benchmarks' questions without images, and there is a risk of data leakage during training. We find that these benchmarks mostly test knowledge or just simple image understanding and don't assess logic and reasoning skills. Image implication understanding represents a more challenging task compared to image understanding, necessitating multi-hop reasoning ability and theory of mind (ToM) [14, 23, 63, 72, 56, 57]—the sophisticated capability intrinsic to human cognition. II-Bench is a benchmark designed to evaluate MLLMs' prowess in both image understanding and reasoning through image implication.

## 3 The II-Bench

### 3.1 Overview of II-Bench

We introduce the **I**mage **I**mplication Understanding **Bench**mark (**II-Bench**), a new benchmark measuring the higher-order perceptual, reasoning and comprehension abilities of MLLMs when presented with complex implication images. These images, including abstract artworks, comics and posters, possess visual implications that require an understanding of visual details and reasoning ability. II-Bench reveals whether current MLLMs, leveraging their inherent comprehension abilities, can accurately decode the implications embedded within the complex and abstract information presented in these images.

II-Bench contains a total of 1,222 various images. The specific image types and domain statistics can be seen in Figure 5 of the Appendix A. These images are manually collected and annotated by 50 undergraduate students from various disciplines and institutions, with sources from multiple renowned illustration websites. Each image is manually designed with one to three multiple-choice questions, each with six options and only one correct answer. The questions cover the metaphors,

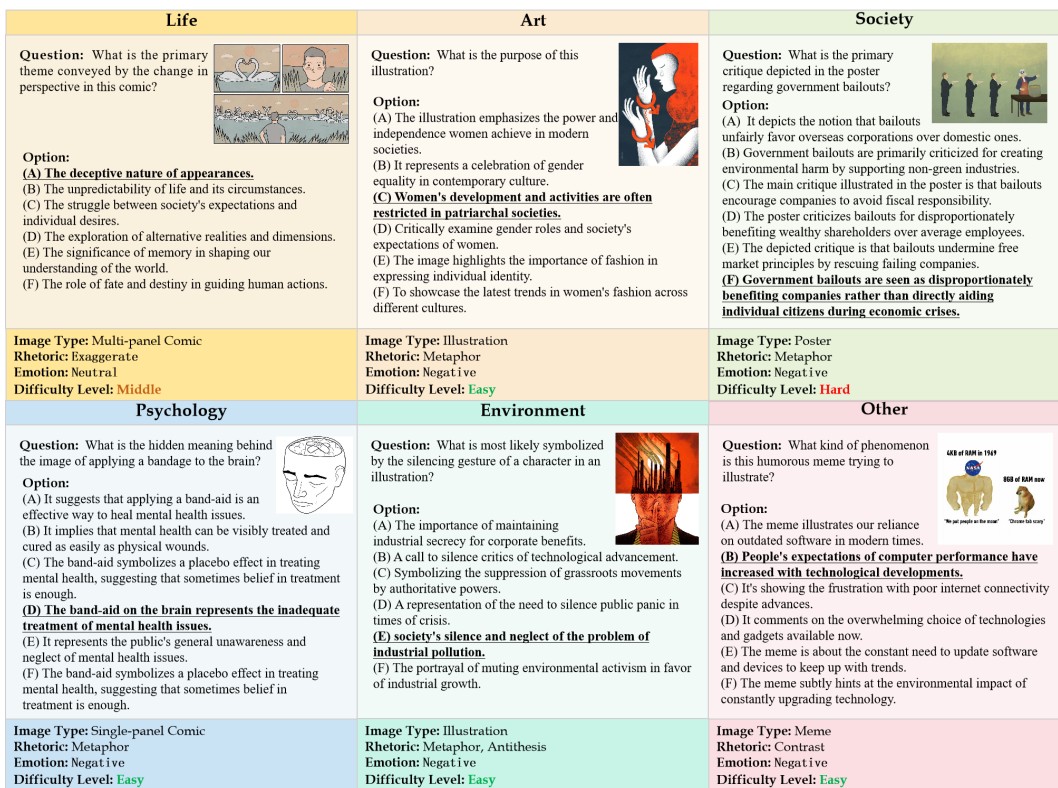

Figure 3: II-Bench examples sampled from each domain. The pictures include life, art, society, psychology, environment and other domains. Understanding these images and completing the corresponding questions require a certain level of comprehension.

symbolism, and detailed understanding of the images. The benchmark includes a total of 1,434 multiple-choice questions, with 1,399 questions used to construct the test set and 35 questions used to construct the development and validation set for few-shot tasks. Figure 3 shows representative examples of II-Bench.

## 3.2 Data Curation Process

**Data Collection.** We collect 20,150 raw images from various renowned illustration websites, ensuring a sufficiently extensive raw dataset. Our collectors are well instructed to adhere to copyright and license regulations, avoiding data from sites prohibiting copy and redistribution. We have also taken special measures to prevent data leakage. For detailed information on the specific websites from which we collect images and our data leakage prevention measures, please refer to Appendix B.

**Data Filtration.** After collecting the raw images, we carefully design a three-stage data filtration procedure. In **Stage 1**, dedicated to *image deduplication*, we utilize image similarity algorithms to perform pixel-based comparisons which allows the identification and elimination of copies and close variants, rendering the dataset unique. In **Stage 2**, focused on *text-to-image ratio control*, we use Optical Character Recognition (OCR) to locate text portions in the images. We then calculate the area occupied by text relative to the total image area. Images are removed if the text-to-image ratio breaches the threshold, ensuring that the dataset remains visually dominant. In **Stage 3**, an exhaustive visual inspection is conducted by humans. Our specific screening protocol is mandated to identify and discard images lacking implications. This strategic exclusion ensures that irrelevant and poor-quality images are weeded out, enhancing the meaningfulness and quality of data retained for further processing. After these filtration stages, we have eliminated over 90% of the original images, leaving us with fewer than 2,000 images.

**Data Annotation.** We forward the annotation sources to the crowdsourcing annotators and perform three steps of data annotation using our carefully devised annotation protocol. The annotators mark the images with their difficulty, image type, domain, and corresponding rhetoric first. An explanation of contained visual implications is then drafted for each image. Finally, the annotators devise 1-3 fine-grained questions per image, each with only one correct answer and five distractor options related to the implication nuances. The detailed annotation protocol is in Appendix B.

**Data Quality Assurance.** To ensure the quality, difficulty of questions and distractor options and consistency among annotators, we have implemented a rigorous multi-step process. Specifically, each question and option undergoes multiple rounds of meticulous manual annotation to ensure the distractors are sufficiently challenging and not easily distinguishable from the correct option and ensure consistency across different annotators. If any question is found to be insufficiently challenging or answers from different annotators are not consistent, it is revised accordingly. This iterative process continues until at least 4 out of 5 independent reviewers agree that the question is accurate, reasonable, difficult and reach a consensus.

### 3.3 Dataset Statistics

II-Bench comprises 1,222 images, each accompanied by 1 to 3 multiple-choice questions, totaling 1,434 questions. We randomly select 35 of these questions to construct a few-shot development set and validation set. The average question length is approximately 17 words, and the average option length is 14 words. Each image also includes a description manually annotated by annotators, explaining the human interpretation of the image's implication.

II-Bench encompasses images from six distinct domains: Life, Art, Society, Psychology, Environment and Others. It features a diverse array of image types, including Illustrations, Memes, Posters, Multi-panel Comics, Single-panel Comics, Logos and Paintings. The images are classified based on human understanding into three levels of difficulty: Easy, Middle and Hard. Additionally, they are categorized by the emotional tone they convey: Positive, Neutral or Negative. Furthermore, each image is manually annotated with rhetorical devices such as Metaphor, Exaggeration, Symbolism, Contrast, Visual Dislocation, Antithesis, Analogy, Personification and Others. The detailed statistical information can be found in Table 1.

| Statistics | | Statistics | |
|---|---|---|---|
| Total Questions | 1,434 | Life | 516 (42.23%) |
| Total Images | 1,222 | Art | 70 (5.73%) |
| Dev : Validation : Test | 15 : 20 : 1,187 | Society | 408 (33.39%) |
| Easy : Medium : Hard | 708 : 385 : 129 | Psychology | 127 (10.39%) |
| | | Environment | 44 (3.60%) |
| Average Question Length | 16.91 | Other | 57 (4.66%) |
| Average Option Length | 14.05 | | |
| Average Explanation Length | 170.47 | Positive | 169 (13.83%) |
| | | Neutral | 702 (57.45%) |
| Metaphor | 955 | Negative | 351 (28.72%) |
| Exaggerate | 191 | | |
| Symbolism | 236 | Illustration | 374 (28.70%) |
| Visual Dislocation | 71 | Meme | 269 (20.64%) |
| Antithesis | 27 | Poster | 111 (8.52%) |
| Analogy | 38 | Multi-panel Comic | 311 (23.87%) |
| Personification | 108 | Single-panel Comic | 90 (6.91%) |
| Contrast | 226 | Logo | 59 (4.53%) |
| Other | 47 | Painting | 89 (6.83%) |

Table 1: Statistics of II-Bench.

## 4 Experiment

We conduct experiments on II-Bench using both open-source and closed-source MLLMs. For each model, we employ eight different settings: 1-shot, 2-shot, 3-shot, zero-shot (None), CoT,

Domain, Emotion and Rhetoric. "Emotion" denotes prompts where the model is informed about the emotional polarity of the images(e.g., positive, negative), "Domain" involves adding information about the image's domain (e.g., life, environment) to the prompt, and "Rhetoric" signifies prompt with information about the rhetorical devices used in the image (e.g., metaphor, personification), while "None" indicates the use of standard prompts without any additional information. Uniform prompts are applied across all MLLMs, with detailed specifications available in the Appendix C. All experiments are conducted on NVIDIA A800 GPUs.

## 4.1 Baselines

**MLLMs.** Table 2 provides an overview of the studied MLLMs, highlighting differences in their architectures and parameters. Notably, InternLM-XComposer2[15] attempts to modify the projection module in LLaVA architecture to better align multiple modalities. Meanwhile, CogVLM2[60] integrates a visual expert into the large language model, enabling a deep fusion of vision and language features without compromising performance on NLP tasks.

| Model | Size | ViT | Projection Module | LLM |
|---|---|---|---|---|
| CogVLM2-Llama3-Chat [60] | 19.5B | EVA2-CLIP-E | MLP | Llama-3-8B + Visual Expert |
| MiniCPM-Llama3-2.5 [25] | 8.5B | SigLip-400M | Perceiver Resampler | Llama3-8B |
| InternVL-Chat-1.5 [10] | 25.5B | InternViT-6B | MLP | InternLM2-20B |
| InternLM-XComposer2-VL [15] | 7B | OpenAI ViT-Large | PLoRA | InternLM-2 |
| DeepSeek-VL-Chat-7B [42] | 7.3B | SAM-B + SigLIP-L | MLP | DeepSeek-LLM-7B |
| InstructBLIP-T5 [13] | 4.0B/12.3B | ViT-g/14 | MLP | FLAN T5 XL/XXL |
| BLIP-2 FLAN-T5 [35] | 4.1B/12.1B | ViT-g/14 | MLP | FLAN T5 XL/XXL |
| mPLUGw-OWL2 [64] | 8.2B | ViT-L/14 | Visual Abstractor | Llama-2-7B |
| Qwen-VL-Chat [3] | 9.6B | ViT-bigG | VL Adapter | Qwen-7B |
| Yi-VL-34B-Chat [65] | 7.1B/35.4B | CLIP ViT-H/14 | MLP | Yi-34B-Chat |
| LLaVA-1.6-34B [37] | 34.8B | ViT-L/14 | MLP | Nous-Hermes-2-Yi-34B |
| Mantis-8B-siglip-llama3 [28] | 8.5B | SigLIP | MLP | Llama-3-8B |
| Idefics2-8B [30] | 8.4B | SigLIP | MLP | Mistral-7B |

Table 2: The architecture and size of different models.

**Evaluation.** Accuracy is used as our main evaluation metric. Given that II-Bench comprises entirely multiple-choice questions, the evaluation merely involves extracting the selected options from the model's responses, thereby simplifying the rule design complexity. Notably, when the model employs chain-of-thought (CoT) prompting, the responses generate intermediate steps. This necessitates that the designed rules possess sufficient robustness or that the model outputs answers in a fixed format. If the options cannot be extracted from the model's response, it is deemed that the model has answered the current question incorrectly. For the robust answer extraction method and the detailed statistics of the model output, please see Appendix E. For reference, we also assessed human performance on II-Bench.

## 4.2 Main Results

In this section, we present a comprehensive comparison of different MLLMs and humans on II-Bench. The detailed results of different domains and emotions are in Table 3. The detailed results of different image types, levels of difficulty, and rhetoric are in Appendix D. The main experimental results and findings are summarized below:

### 4.2.1 Gap between Humans and MLLMs

The results indicate a significant disparity between humans and MLLMs on II-Bench. Human participants achieve an average accuracy of 90.3%, with the highest accuracy reaching 98.2%. In comparison, the best closed-source model, Qwen-VL-MAX, achieves an accuracy of 74.8%, while the best open-source model, LLaVA-1.6-34B, scores 73.8%. These results highlight the substantial gap between human capabilities and current state-of-the-art models in understanding image implications. The highest accuracy of the models is substantially lower than the average human score, underscoring the challenges that MLLMs face in this domain.

| | Overall (1,399) | Life (585) | Art (85) | Society (461) | Psy. (152) | Env. (51) | Others (65) | Positive (196) | Neutral (789) | Negative (414) |
|---|---|---|---|---|---|---|---|---|---|---|
| *Open-source Models* | | | | | | | | | | |
| InstructBLIP-T5-XL | 47.3 | 45.6 | 48.2 | 48.8 | 44.7 | 52.9 | 50.8 | 46.9 | 48.3 | 45.4 |
| BLIP-2 FLAN-T5-XL | 52.8 | 53.0 | 58.8 | 52.5 | 42.8 | 64.7 | 58.5 | 56.1 | 52.9 | 51.0 |
| mPLUGw-OWL2 | 53.2 | 54.0 | 56.5 | 50.5 | 52.0 | 60.8 | 56.9 | 55.6 | 52.6 | 53.1 |
| Qwen-VL-Chat | 53.4 | 53.2 | 49.4 | 52.1 | 50.0 | 60.8 | 72.3 | 56.1 | 52.6 | 53.6 |
| InstructBLIP-T5-XXL | 56.7 | 56.2 | 58.8 | 58.6 | 45.4 | 64.7 | 64.6 | 63.3 | 56.1 | 54.6 |
| Mantis-8B-siglip-Llama3 | 57.5 | 56.8 | 61.2 | 57.5 | 53.9 | 64.7 | 61.5 | 59.2 | 58.0 | 55.6 |
| BLIP-2 FLAN-T5-XXL | 57.8 | 57.1 | 63.5 | 57.0 | 53.3 | 66.7 | 66.2 | 67.9 | 57.2 | 54.3 |
| DeepSeek-VL-Chat-7B | 60.3 | 59.0 | 58.8 | 58.4 | 61.8 | 68.6 | 76.9 | 65.8 | 60.1 | 58.0 |
| Yi-VL-6B-Chat | 61.3 | 60.9 | 63.5 | 60.7 | 56.6 | 66.7 | 72.3 | 61.7 | 61.7 | 60.1 |
| InternLM-XComposer2-VL | 62.1 | 61.7 | 62.4 | 62.3 | 58.6 | 70.6 | 66.2 | 65.8 | 63.0 | 58.7 |
| InternVL-Chat-1.5 | 66.3 | 63.6 | 65.9 | 68.5 | 65.8 | 64.7 | 76.9 | 73.5 | 65.4 | 64.5 |
| Idefics2-8B | 67.7 | 67.2 | **74.1** | 67.7 | 62.5 | 74.5 | 70.8 | 68.9 | 67.0 | 68.4 |
| Yi-VL-34B-Chat | 67.9 | 67.5 | 70.6 | 67.7 | 63.8 | 70.6 | 76.9 | 74.0 | 68.2 | 64.5 |
| MiniCPM-Llama3-2.5 | 69.4 | 68.4 | 71.8 | 69.4 | 64.5 | **80.4** | 78.5 | 75.0 | 69.3 | 66.9 |
| CogVLM2-Llama3-Chat | 70.3 | 68.9 | 68.2 | 70.9 | 67.8 | 72.5 | **86.2** | 69.9 | 71.1 | 69.1 |
| LLaVA-1.6-34B | **73.8** | **73.8** | 71.8 | **73.3** | **71.1** | 78.4 | 81.5 | **79.1** | **72.9** | **72.9** |
| *Closed-source Models* | | | | | | | | | | |
| GPT-4V | 65.9 | 65.0 | 69.4 | 65.3 | 59.9 | 76.5 | 80.0 | 69.4 | 66.0 | 64.0 |
| GPT-4o | 72.6 | 72.5 | 72.9 | 73.3 | 68.4 | 76.5 | 75.4 | 78.6 | 71.2 | 72.5 |
| Gemini-1.5 Pro | 73.9 | 73.7 | **74.1** | 74.4 | 63.2 | **80.4** | 83.1 | **80.1** | 70.8 | **75.4** |
| Qwen-VL-MAX | **74.8** | **74.7** | 71.8 | **74.6** | **73.0** | 76.5 | **84.6** | **80.1** | **74.5** | 72.9 |
| *Humans* | | | | | | | | | | |
| Human_avg | 90.3 | 90.0 | 88.2 | 91.4 | 86.6 | 96.1 | 92.3 | 84.7 | 89.1 | 92.2 |
| Human_best | **98.2** | **97.9** | **98.8** | **98.3** | **97.4** | **100.0** | **100.0** | **98.0** | **98.0** | **98.8** |

Table 3: Overall results of different MLLMs and humans on different domains and emotions. The best-performing model in each category is **in-bold**, and the second best is underlined.

### 4.2.2 Disparity between Open-source and Closed-source Models

The results on II-Bench reveal that closed-source models generally perform better, with open-source models exhibiting a larger variance. However, some open-source models show excellent performance. The highest scores for open-source and closed-source models are LLaVA-1.6-34B (73.8%) and Qwen-VL-MAX (74.8%), respectively. Top open-source models like CogVLM2-Llama3-Chat-19B, MiniCPM-Llama3-2.5, Yi-VL-34B-Chat, Idefics2-8B, and InternVL-Chat-1.5 outperform the closed-source model GPT-4V's 65.9% accuracy but fall short of GPT-4o's 72.6%.

According to our analysis, the image implication understanding not only tests the model's image understanding ability but also tests the model's multi-hop reasoning ability. From the image understanding perspective, top open-source MLLMs perform closely to GPT-4V on various OCR-related benchmarks[41, 47, 53] and general multimodal benchmarks[69, 67, 40, 32, 31]. In terms of logical reasoning, multi-hop reasoning ability is crucial, and LLMs used in MLLMs like Llama3-Chat-8B, InternLM2-Chat-20B, and Yi-34B-Chat exhibit strong performance in reasoning and mathematics benchmarks[58, 68, 22, 39, 11]. Conversely, InstructBLIP-T5-XL, with weaker multi-hop reasoning ability from its language model Flan-T5-XL, shows the lowest accuracy at 47.3%.

### 4.2.3 Model Performance across Different Domains and Emotions

In terms of domain performance, our results in Table 3 indicate that the models generally perform better in the Environment, Other, Life and Society domains, achieving higher accuracy. Conversely, the accuracy is lower in the Art and Psychology domains, which suggests that while the models generalize well in common domains, they struggle with the more abstract and logically demanding information found in Art and Psychology.

From an emotional perspective, the models tend to exhibit higher accuracy when the image implications convey positive emotions, while accuracy is the lowest for images with negative emotions. This discrepancy highlights that the models' preferences do not align with those of humans, as humans are significantly more sensitive to negative implications. Additionally, the results suggest that the models are overly biased towards positive responses, potentially reflecting a positive emotion bias in the training data.

| Models | None | CoT | Domain | Emotion | Rhetoric |
|---|---|---|---|---|---|
| *Open-source Models* | | | | | |
| InstructBLIP-T5-XL | 47.3 | 30.0 | 47.8 | 49.8 | 47.6 |
| BLIP-2 FLAN-T5-XL | 52.8 | 42.0 | 51.4 | 51.8 | 51.5 |
| mPLUGw-OWL2 | 53.2 | 54.2 | 54.5 | 55.0 | 55.7 |
| Qwen-VL-Chat | 53.4 | 51.6 | 54.9 | 57.0 | 54.0 |
| InstructBLIP-T5-XXL | 56.7 | 50.8 | 56.7 | 58.7 | 56.0 |
| Mantis-8B-siglip-Llama3 | 57.5 | 56.7 | 57.1 | 57.0 | 58.0 |
| BLIP-2 FLAN-T5-XXL | 57.8 | 42.5 | 57.5 | 58.4 | 57.3 |
| DeepSeek-VL-Chat-7B | 60.3 | 59.2 | 60.4 | 63.3 | 59.8 |
| Yi-VL-6B-Chat | 61.3 | 60.8 | 60.8 | 62.8 | 60.4 |
| InternLM-XComposer2-VL | 62.1 | 60.7 | 60.9 | 61.5 | 61.6 |
| InternVL-Chat-1.5 | 66.3 | 63.3 | 66.6 | 67.4 | 65.6 |
| Idefics2-8B | 67.7 | 67.7 | 67.0 | 68.6 | 66.6 |
| Yi-VL-34B-Chat | 67.9 | 67.6 | 67.7 | 70.1 | 67.6 |
| MiniCPM-Llama3-2.5 | 69.4 | 67.4 | 70.3 | 70.8 | 69.3 |
| CogVLM2-Llama3-Chat-19B | 70.3 | **69.3** | 69.1 | 71.7 | 69.3 |
| LLaVA-1.6-34B | **73.8** | 60.0 | **73.1** | **75.3** | **73.3** |
| *Closed-source Models* | | | | | |
| GPT-4V | 65.9 | 68.4 | 66.0 | 68.3 | 69.3 |
| GPT-4o | 72.6 | **75.7** | 72.6 | 74.2 | 71.3 |
| Gemini-1.5 Pro | 73.9 | 68.2 | 73.1 | 70.5 | 71.3 |
| Qwen-VL-MAX | **74.8** | 74.1 | **74.1** | **75.5** | **73.6** |

Table 4: Overall results of different prompts on II-Bench. The label(*Emotion, Domain, Rhetoric*) means providing corresponding information for the images in the prompt. The best-performing model in each category is **in-bold**, and the second best is underlined.

### 4.2.4 Analysis on different prompt skills

We present a comprehensive analysis of prompt skills, with detailed results in Table 4.

**Analysis of Chain-of-Thought (CoT).** The Chain-of-Thought (CoT) prompting skill is evaluated to determine its impact on model performance in Table 4. The results indicate that CoT has no significant effect on improving accuracy. In some cases, particularly with smaller open-source models, the accuracy even decline when CoT is used. For example, CogVLM2-Llama3-Chat-19B scores 70.3% without CoT and drops to 69.3% with CoT, InternVL-Chat-1.5 scores 66.3% and 63.3% as the same.

One primary reason for the observed decline in performance with CoT prompting is that many models fail to adhere to the required format. Specifically, CoT prompts require models to output both an analysis and a final answer in a fixed format. However, many models only provide the analysis without concluding with the final answer. We manually check the outputs and find that models either fail to explicitly generate the answer option after the analysis (instead of generating the content of the answer) or select multiple options, which reflect the decline in instruction following ability, leading to the failure of regex matching. An obvious example is BLIP-2 FLAN-T5-XXL, where using the CoT prompt results 15.8% increase in responses that fail to match our regex compared to the direct answer prompt. This issue is highlighted in Appendix E, where we present statistics on model outputs. The lack of a final answer in CoT responses leads to extraction failures, which negatively impacts performance.

Another contributing factor is that CoT prompting does not universally enhance performance across all types of tasks. In evaluations such as MMLU [61], C-Eval [26], where the primary focus is not on logical reasoning or mathematical problem-solving, CoT prompting often does not lead to better results. The same phenomenon is also observed in our experiments, which contrast with tasks that inherently benefit from step-by-step reasoning where CoT can be more effective. These findings align with other benchmarks [69, 34, 21].

**Analysis of Different Types and Domains.** To evaluate the impact of different label information on model accuracy, we conduct an ablation study by providing corresponding label information (Emotion, Domain, Rhetoric) for the images in the prompt. The results in Table 4 indicate that

Emotion labels significantly enhance model accuracy, followed closely by Domain and Rhetoric labels, which exhibit similar effectiveness.

This outcome is consistent with the human perspective of image implication comprehension. Emotion labels likely provide more intuitive and salient cues that align closely with human interpretative processes, thereby facilitating better model performance. In contrast, Domain and Rhetoric labels, while still beneficial, are not as immediately intuitive or universally applicable, thus resulting in slightly lower effectiveness in improving model accuracy. At the same time, from the perspective of model training, the model has a normal understanding of emotion, unlike the specific nouns we define ourselves in the Rhetoric and Domain labels. The model does not see many descriptions of such specific nouns during pre-training, which does not help improve accuracy.

**Analysis of Few-shot Examples.** The results in Table 5 demonstrate that few-shot examples do not enhance the accuracy of the models. Specifically, the performance tends to drop as more examples are provided. This can be attributed to the models' inferior multi-image capabilities compared to their single-image capabilities, leading to a decline in accuracy with an increasing number of shots. Additionally, as the number of shots increases, the input length becomes longer, and the model's long text ability is insufficient, resulting in poor long context performance. An example is Qwen-VL-Max, where inputs exceeding 6,000 tokens cause errors. Moreover, chat models generally exhibit good instruction following ability, reducing the necessity for few-shot examples.

| Model | 0-shot | 1-shot | 2-shot | 3-shot |
|---|---|---|---|---|
| Qwen-VL-Chat | 53.4 | 43.3 | 47.9 | 41.1 |
| Mantis-8B-siglip-Llama3 | 57.5 | 55.3 | 54.2 | 54.9 |
| GPT-4V | 65.9 | 65.5 | 67.7 | 67.1 |
| Idefics2-8B | 67.7 | 64.1 | 62.4 | 59.5 |
| Gemini-1.5 Pro | 73.9 | 73.2 | 73.8 | 74.1 |
| Qwen-VL-Max | 74.8 | 74.5 | 69.6 | 53.6* |

Table 5: Few-shot results of different models on the II-Bench. ∗ means exceeds the context length.

### 4.3 Error Analysis

In order to perform a comprehensive error analysis of GPT-4V's performance on II-Bench, we randomly select 100 erroneous samples from each domain, in proportion to their representation in the dataset. These samples are meticulously analyzed by expert annotators. As illustrated in Figure 4, GPT-4V's errors can be categorized into the following types: Metaphorical Misunderstanding, Detail Misunderstanding, Detail Ignorance, Surface-Level Interpretation, Reasoning Error, Reject to Answer and Answer Extraction Error. This error analysis is crucial for gaining deeper insights into the capabilities of MLLMs and identifying the current limitations in image comprehension tasks. Understanding these shortcomings can guide researchers in developing and training more robust and performant models in the future. A selection of 77 notable cases, along with detailed analyses, is included in Appendix G, providing further insights into the nature of these errors. **Reminder: although we filtered and sifted as much as possible, some of the negative cases in the appendix are offensive to certain groups of people.**

**Metaphorical Misunderstanding (36%):** Metaphorical Misunderstanding is a common error that GPT-4V makes when generating responses based on image comprehension. This indicates that the model has misunderstood the implications or symbolic meanings within the images. There are two main reasons for this. First, the model might grasp certain aspects of the image's meaning, but its overall understanding of the image's theme is incorrect, as exemplified by Fig.G28. Second, some implications and hidden meanings require specific knowledge to be understood, and the model's internal knowledge might not cover

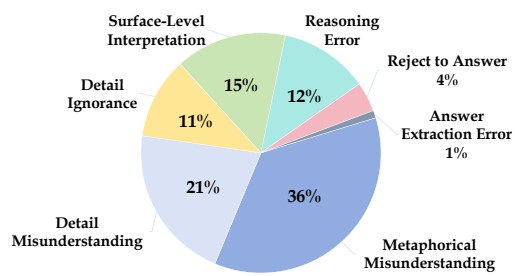

Figure 4: GPT-4V error response distribution.

these areas, leading to an incorrect interpretation
of the image's deeper meaning.

**Detail Misunderstanding (21%):** Detail Misunderstanding is another common mistake made by GPT-4V. Understanding details is very important for models, as inaccuracies in understanding details can sometimes affect how the model interprets the meaning of images. For instance, in Fig.G11, GPT-4V has an error in understanding the details, resulting in an incorrect response.

**Other Errors:** The remaining errors are detail ignorance (11%), surface-level interpretation (15%), reasoning error (12%), reject to answer (4%), and answer extraction error (1%). The description of these errors can be found in Appendix F.

## 5 Conclusion

The development of II-Bench for assessing the capabilities of MLLMs represents a significant milestone in the journey towards achieving Expert AGI, marking a step into higher-order theory of mind in the exploration of the capabilities of MLLMs. The experimental results show that the current state-of-the-art MLLMs are good at understanding the surface content of image, but the gap between the understanding of image implication and humans is still huge. We found that including information about the emotional polarity of the image in the prompts usually improves the model score, suggesting that the model lacks sufficient emotional understanding of the image, which leads to misinterpretation of the implied meaning. Moreover, we found that humans would implicitly understand neutral and negative emotions much better than models. The vast majority of MLLMs perceive positive emotions better than neutral and negative emotions, and we think that the distribution of training data for MLLMs is more skewed toward positive emotions. We believe II-Bench will stimulate the community to build next generation multimodal foundation models towards expert AGI.

## Limitations

We acknowledge several limitations in our study. While II-Bench is comprehensive, the inclusion of subjective elements can lead to varying interpretations, potentially affecting result consistency. Additionally, our benchmark focuses on specific domains, covering only a portion of human knowledge. The evaluation metrics might not entirely reflect the sophisticated understanding and reasoning abilities of advanced AI systems. These limitations highlight the need for ongoing refinement and expansion of our benchmarks. In future work, we aim to develop and incorporate more stringent and objective test sets to enhance reliability and validity of our benchmark.

## Ethics Statement

In developing II-Bench, we strictly adhere to ethical guidelines and legal regulations, ensuring fairness, transparency, inclusivity and respect for all stakeholders. We stress the importance of safeguarding privacy and intellectual property rights, underscoring our commitment to responsible and lawful data management. We have taken steps to anonymize any personal data to protect privacy and and have made every effort to minimize harmful or biased content. However, we recognize that biases can inadvertently arise and some information may be potentially offensive. We are committed to continuous monitoring and improvement to mitigate such biases. Furthermore, we encourage users of our dataset to employ it responsibly and to consider the ethical implications of their work, particularly in applications that may impact individuals or communities.

## Acknowledgement

This work was supported by National Key Research and Development Program of China (2022YFF0902100), China Postdoctoral Science Foundation (2024M753398), Postdoctoral Fellowship Program of CPSF (GZC20232873), Guangdong Basic and Applied Basic Research Foundation (2023A1515110718 and 2024A1515012003), and Shenzhen Basic Research Foundation (JCYJ20210324115614039).

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

# A  Image type and domain statistics

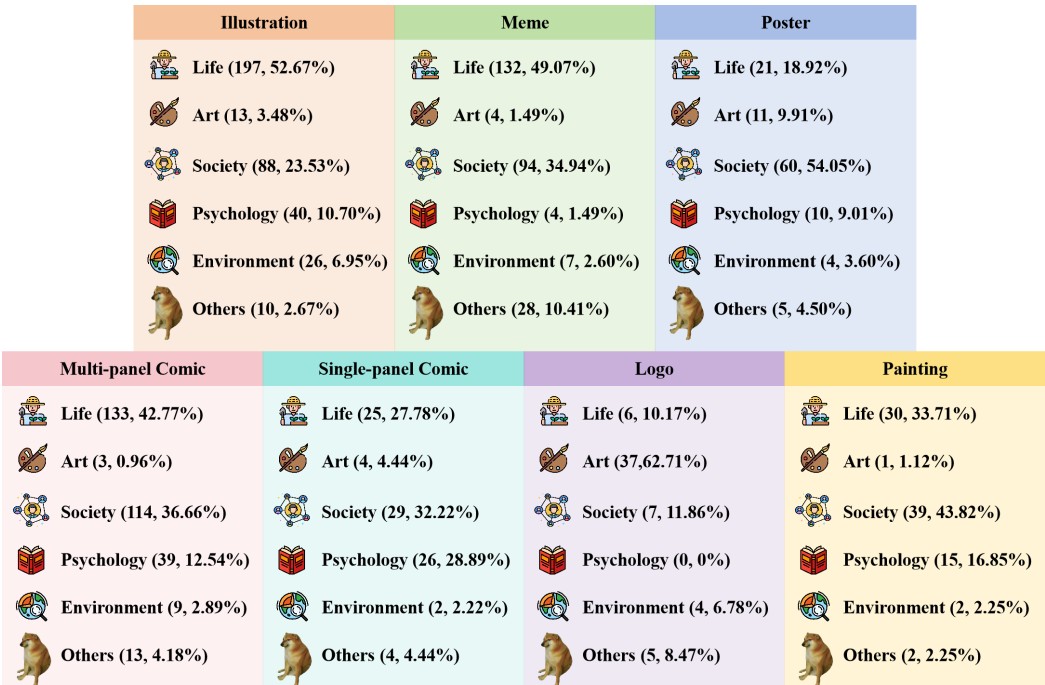

Figure 5: II-Bench specific image type and domain statistics.

# B  Data Annotation Protocol

This document outlines a comprehensive protocol for annotating a dataset consisting of questions that explore the metaphorical implications of images.

## B.1  Data Collection

Some websites from which we collect data are as follows:

- `https://www.davidebonazzi.com`
- `https://www.boredpanda.com`
- `https://themindsjournal.com`
- `https://naldzgraphics.net/satirical-illustrations-agim-sulaj`
- `https://www.pinterest.co.uk`
- `https://www.asafhanuka.com/the-realist`

## B.2  Data Leakage Prevention

We have made efforts to mitigate data leakage risk:

- **Recent Data Collection**: We have made an effort to collect data from a recent time, which aims to reduce the likelihood that these specific images are included in the training sets of existing models.
- **Brand new, Manually Constructed Questions**: The questions we posed are entirely new and manually constructed, which are not present in any model's training set. Even if the images themselves are part of the training data, it is unlikely to correctly answer our high-level implication questions.

- **Multimodal Data Leakage Assessment**: We conduct evaluations where models are tested without providing the corresponding images. The experimental results are shown in the table 6, which indicate that models perform badly without the images, suggesting that the models do not possess pre-existing knowledge of the specific questions and images, thereby indicating that there is no risk of data leakage.

These measures collectively help ensure that our evaluations are fair and that the risk of data leakage is minimized.

| Model | Mode | Accuarcy |
|-------|------|----------|
| deepseek-vl-7b-chat | w. image | 60.3 |
| deepseek-vl-7b-chat | w/o. image | 36.8(**-23.5**) |
| Qwen-VL-Chat | w. image | 53.4 |
| Qwen-VL-Chat | w/o. image | 38.0(**-15.4**) |
| idefics2-8b | w. image | 67.7 |
| idefics2-8b | w/o. image | 47.8(**-19.9**) |

Table 6: The architecture and size of different models.

### B.3 General Guidelines

**General Principles:**

- Annotations should be accurate and consistent.
- All questions, options and explanation should be written in English.
- Any images without metaphorical implications should be discarded.

**Specific Instructions:**

- Each image needs to be categorized as one of the following image types: single-panel comic, multi-panel comic, poster, logo, meme, illustration or painting.
- Each image needs to be categorized as one of the following difficulty levels from a human understanding perspective: easy, middle, or hard.
- Each image needs to be categorized as one of the following domains: life, art, society, psychology, environment or others.
- Each image needs to be categorized as one of the following emotions: positive, neutral or negative.
- Each image needs to be categorized as one or more of the following rhetoric: metaphor, exaggerate, symbolism, contrast, visual dislocation, antithesis, analogy, personification or others.
- Each image needs a human explanation.
- Each image needs 1-3 questions about the fine-grained metaphorical implications of the image, each with one correct answer and five distractor options.

### B.4 Additional Data Quality Assurance

To further ensure the quality and reliability of the data, the annotated datasets were double-checked and cross-validated. Each question was manually validated by at least four annotators. Any inconsistencies or misinterpretations found were thoroughly examined and resolved by consensus of the annotation team, thus improving the reliability of the dataset while ensuring consistency of the annotations. In total, we conducted four rounds of data quality checks to ensure data quality and ultimately obtain II-Bench.

### B.5 Ethical Considerations

**Copyright and Licensing.** It is essential to strictly follow all copyright and licensing regulations. Data from sources that do not permit copying or redistribution will be explicitly excluded.

**Data Privacy.** Adherence to privacy laws and ethical standards in data handling is crucial. Annotators must avoid collecting questions that contain any personal information.

# C   Prompts

In experiments, the prompts of different settings are as follows:

## C.1   None

```
instruction: "Instruction: Please try to answer the single-answer multiple choice
↪   question below based on the picture provided."
prompt_format:
  - |
    Question: {}
    (A) {}
    (B) {}
    (C) {}
    (D) {}
    (E) {}
    (F) {}
    Answer:
```

## C.2   Few-shot

```
instruction:
  - |
    Instruction: Please try to answer the single-answer multiple choice question
    ↪   below based on the example(with answer) and the corresponding picture.
  - |
    Instruction: Please try to answer the single-answer multiple choice question
    ↪   below based on the examples(with answers) and the corresponding pictures.
prompt_format:
  - |
    Question: In the comic image, what deeper societal commentary might Barry's
    ↪   costume choice at the party represent?
    Picture: <Picture {}>
    (A) The backlash faced when challenging traditional roles.
    (B) The struggle to fit in while also standing out in social circles.
    (C) The challenge of maintaining personal identity in group dynamics.
    (D) The discomfort caused by confronting controversial or taboo topics in
    ↪   social settings.
    (E) The effects of poor decision-making on interpersonal relationships.
    (F) The significance of color coordination in party costumes to enhance the
    ↪   festive atmosphere.
    Answer: (D)
  - |
    Question: What hidden message can be inferred about the dynamics of fame and
    ↪   the collective cultural memory from the text and images of Brendan Fraser
    ↪   within the meme?
    Picture: <Picture {}>
    (A) The meme suggests that the public and media often overlook certain
    ↪   celebrities in favor of others due to shifting trends and narratives in
    ↪   popular culture.
    (B) The imagery suggests that personal struggles of celebrities are often
    ↪   overlooked by the public and media.
    (C) It points to a discrepancy between the talent and contributions of
    ↪   celebrities and their recognition in the media.
    (D) The focus on Brendan Fraser is meant to highlight how male fashion trends
    ↪   drastically changed from the 90s to the present.
    (E) Brendan Fraser is depicted as the quintessential 90s figure, indicating
    ↪   that he defined the entire decade's style and sensibilities.
    (F) The meme indicates that celebrities who maintain a consistent public
    ↪   image are more likely to remain in the spotlight.
```

```
   Answer: (A)
 - |
   Question: What is the metaphorical significance of the glowing eye in this
   ↪  image?
   Picture: <Picture {}>
   (A) It represents the ever-present nature of surveillance in society.
   (B) It symbolizes enlightenment and the pursuit of knowledge.
   (C) It signifies wisdom and the foresight of a leader.
   (D) It depicts the uninterrupted attention and care from protectors.
   (E) It represents the vigilance and unending watchfulness of authority.
   (F) It conveys the omnipresent gaze of societal norms and expectations.
   Answer: (E)
 - |
   Question: {}
   Picture: <Picture {}>
   (A) {}
   (B) {}
   (C) {}
   (D) {}
   (E) {}
   (F) {}
   Answer:
```

## C.3  Keywords

The keywords here include one of the following: emotion, domain, rhetoric.

```
instruction: "Instruction: Please try to answer the single-answer multiple choice
↪  question below based on the picture and the key words."
prompt_format:
  - |
    Key words: {}
    Question: {}
    (A) {}
    (B) {}
    (C) {}
    (D) {}
    (E) {}
    (F) {}
    Answer:
```

## C.4  CoT

```
instruction: "Instruction: Please try to answer the single-answer multiple choice
↪  question below based on the picture provided. Let's think through each
↪  option. Let's think step by step."
prompt_format:
  - |
    Question: {}
    (A) {}
    (B) {}
    (C) {}
    (D) {}
    (E) {}
    (F) {}
    Explanation:
    Answer:
```

# D    Results on Different Types, Difficulties and Rhetoric

In this section, we report the performance of different MLLMs and humans on different types of images, levels of difficulty, and rhetoric.

## D.1    Image Types and Difficulty

We present a comprehensive comparison of different MLLMs and humans on image types and different levels of difficulty, with detailed results in Table 7.

| | Overall (1,399) | Illu. (436) | Meme (292) | Poster (133) | MPC (359) | SPC (104) | Logo (71) | Paint. (101) | Easy (786) | Mid. (465) | Hard (148) |
|---|---|---|---|---|---|---|---|---|---|---|---|
| *Open-source Models* | | | | | | | | | | | |
| InstructBLIP-T5-XL | 47.3 | 40.8 | 53.8 | 51.9 | 47.4 | 45.2 | 57.8 | 44.6 | 50.1 | 44.7 | 39.9 |
| BLIP-2 FLAN-T5-XL | 52.8 | 44.5 | 59.6 | 60.2 | 54.3 | 54.8 | 69.0 | 47.5 | 56.1 | 49.3 | 46.0 |
| mPLUGw-OWL2 | 53.2 | 43.1 | 63.0 | 59.4 | 56.3 | 55.8 | 63.4 | 43.6 | 56.0 | 50.5 | 46.6 |
| Qwen-VL-Chat | 53.4 | 42.7 | 64.0 | 61.7 | 56.3 | 43.3 | 57.8 | 55.5 | 56.7 | 51.8 | 40.5 |
| InstructBLIP-T5-XXL | 56.7 | 47.9 | 67.1 | 63.2 | 58.5 | 51.9 | 60.6 | 54.5 | 58.8 | 55.9 | 48.0 |
| Mantis-8B-siglip-llama3 | 57.5 | 47.7 | 66.1 | 65.4 | 59.6 | 58.6 | 69.0 | 55.5 | 58.9 | 56.6 | 52.7 |
| BLIP-2 FLAN-T5-XXL | 57.8 | 47.7 | 66.1 | 65.4 | 59.6 | 58.6 | 69.0 | 55.5 | 58.9 | 56.6 | 52.7 |
| DeepSeek-VL-Chat-7B | 60.3 | 47.7 | 70.2 | 72.2 | 65.7 | 59.6 | 67.6 | 51.5 | 64.3 | 57.0 | 49.3 |
| Yi-VL-6B-Chat | 61.3 | 53.2 | 68.5 | 63.9 | 62.4 | 63.5 | 74.6 | 59.4 | 64.1 | 59.4 | 52.0 |
| InternLM-XComposer2-VL | 62.1 | 53.0 | 68.8 | 65.4 | 66.6 | 60.6 | 74.7 | 60.4 | 65.3 | 60.4 | 50.7 |
| InternVL-Chat-1.5 | 66.3 | 54.6 | 78.1 | 71.4 | 71.6 | 66.4 | 71.8 | 59.4 | 69.7 | 64.3 | 54.1 |
| Idefics2-8B | 67.7 | 58.5 | 77.4 | 76.7 | 68.8 | 59.6 | **81.7** | 66.3 | 68.8 | 69.5 | 56.1 |
| Yi-VL-34B-Chat | 67.9 | 56.7 | 81.9 | 70.7 | 71.6 | 60.6 | 77.5 | 58.4 | 71.1 | 66.7 | 54.7 |
| MiniCPM-Llama3-2.5 | 69.4 | 61.9 | 80.5 | 79.0 | 69.1 | 65.4 | 77.5 | 63.4 | 70.2 | 69.7 | **64.2** |
| CogVLM2-Llama3-Chat | 70.3 | 60.8 | 82.9 | 75.9 | 73.5 | 66.4 | 74.7 | 60.4 | 74.2 | 66.9 | 60.8 |
| LLaVA-1.6-34B | **73.8** | **62.8** | **84.6** | **80.5** | **80.5** | **67.3** | 80.3 | **67.3** | **77.5** | **71.4** | 61.5 |
| *Closed-source Models* | | | | | | | | | | | |
| GPT-4V | 65.9 | 55.1 | 79.8 | 73.7 | 69.1 | 64.4 | 67.6 | 58.4 | 69.6 | 61.9 | 58.8 |
| GPT-4o | 72.6 | 64.7 | 81.2 | 78.2 | 76.9 | **72.1** | 80.3 | 66.3 | 76.6 | 67.5 | **67.6** |
| Gemini-1.5 Pro | 73.9 | **66.7** | 82.2 | 79.7 | 74.6 | 70.2 | **81.7** | 74.3 | 75.1 | **74.2** | 66.9 |
| Qwen-VL-MAX | **74.8** | 65.1 | **84.3** | **85.0** | **78.0** | 71.2 | 73.2 | **75.3** | **77.4** | 73.3 | 66.2 |
| *Humans* | | | | | | | | | | | |
| Human_avg | 90.3 | 90.3 | 89.6 | 88.4 | 90.8 | 92.3 | 92.3 | 93.6 | 90.7 | 90.1 | 88.5 |
| Human_best | **98.2** | **98.4** | **99.3** | **99.3** | **96.7** | **97.1** | **100.0** | **99.0** | **98.1** | **98.3** | **98.7** |

Table 7: Overall results of different MLLMs and humans on different image types and different levels of difficulty. The best-performing model in each category is **in-bold**, and the second best is underlined. For brevity, Illu. refers to Illustration, MPC refers to Multi-panel Comic, SPC refers to Single-panel Comic, Paint. refers to Painting and Mid. refers to Middle.

## D.2    Rhetoric

We present a comprehensive comparison of different MLLMs and humans on on different rhetoric, with detailed results in Table 8.

| | Overall (1,399) | Meta. (1106) | Exag. (227) | Symb. (271) | VisD. (88) | Anti. (35) | Anal. (42) | Pers. (128) | Contrast (274) | Others (55) |
|---|---|---|---|---|---|---|---|---|---|---|
| *Open-source Models* | | | | | | | | | | |
| InstructBLIP-T5-XL | 47.3 | 47.6 | 44.9 | 49.8 | 45.5 | 57.1 | 42.9 | 50.8 | 50.7 | 41.8 |
| BLIP-2FLAN-T5-XL | 52.8 | 53.6 | 48.9 | 52.8 | 46.6 | 54.3 | 45.2 | 54.7 | 58.4 | 49.1 |
| mPLUGw-OWL2 | 53.2 | 53.4 | 51.5 | 49.8 | 44.3 | 45.7 | 47.6 | 55.5 | 50.7 | 56.4 |
| Qwen-VL-Chat | 53.4 | 52.9 | 52.9 | 50.2 | 45.5 | 45.7 | 59.5 | 57.8 | 55.1 | 47.3 |
| InstructBLIP-T5-XXL | 56.7 | 57.8 | 57.3 | 53.5 | 51.1 | 51.4 | 42.9 | 63.3 | 60.2 | 50.9 |
| Mantis-8B-siglip-llama3 | 57.5 | 56.6 | 56.8 | 53.1 | 58.0 | 48.6 | 64.3 | 60.9 | 60.2 | 63.6 |
| BLIP-2FLAN-T5-XXL | 57.8 | 58.4 | 55.1 | 56.5 | 56.8 | 54.3 | 52.4 | 64.1 | 59.9 | 52.7 |
| DeepSeek-VL-Chat-7B | 60.3 | 59.8 | 56.8 | 54.6 | 53.4 | 65.7 | 54.8 | 61.7 | 66.1 | 60.0 |
| Yi-VL-6B-Chat | 61.3 | 61.1 | 59.0 | 59.0 | 58.0 | 54.3 | 64.3 | 61.7 | 63.1 | 54.5 |
| InternLM-XComposer2-VL | 62.1 | 61.1 | 57.3 | 62.4 | 56.8 | 54.3 | 66.7 | 71.1 | 63.9 | 67.3 |
| InternVL-Chat-1.5 | 66.3 | 65.7 | 64.8 | 64.2 | 60.2 | 57.1 | 64.3 | 76.6 | 68.2 | 65.5 |
| Idefics2-8B | 67.7 | 67.7 | 67.8 | 63.5 | 68.2 | 77.1 | 66.7 | 66.4 | 70.8 | 70.9 |
| Yi-VL-34B-Chat | 67.9 | 67.7 | 64.8 | 60.5 | **69.3** | 65.7 | 71.4 | 73.4 | 70.8 | 65.5 |
| MiniCPM-Llama3-2.5 | 69.4 | 69.6 | 68.7 | 66.1 | 63.6 | 68.6 | 69.0 | 65.6 | 72.3 | 65.5 |
| CogVLM2-Llama3-Chat | 70.3 | 70.8 | 72.2 | 64.2 | 62.5 | 71.4 | **78.6** | 70.3 | 72.6 | 60.0 |
| LLaVA-1.6-34B | **73.8** | **73.1** | 73.1 | **68.6** | 68.2 | 80.0 | 71.4 | 77.3 | 75.5 | 74.5 |
| *Closed-source Models* | | | | | | | | | | |
| GPT-4V | 65.9 | 65.2 | 60.8 | 61.6 | 67.0 | 80.0 | 69.0 | 72.7 | 68.2 | 70.9 |
| GPT-4o | 72.6 | 71.3 | 69.2 | **70.5** | 63.6 | 71.4 | 78.6 | 78.1 | 72.6 | 74.5 |
| Gemini-1.5 Pro | 73.9 | **74.0** | 75.8 | 68.3 | **70.5** | 68.6 | 78.6 | 75.0 | 74.5 | 69.1 |
| Qwen-VL-MAX | **74.8** | 73.9 | 74.0 | 67.5 | 68.2 | 74.3 | 71.4 | 78.9 | 79.2 | **81.8** |
| *Humans* | | | | | | | | | | |
| Human_avg | 90.3 | 90.1 | 89.9 | 91.3 | 88.6 | 88.6 | 86.9 | 94.1 | 90.0 | 88.2 |
| Human_best | **98.2** | **98.1** | **98.2** | **98.9** | **100.0** | **94.3** | **97.6** | **97.7** | **96.7** | **100.0** |

Table 8: Overall results of different MLLMs and humans on different rhetoric. The best-performing model in each category is **in-bold**, and the second best is underlined. For brevity, Meta. refers to Metaphor, Exag. refers to Exaggerate, Symb. refers to Symbolism, VisD. refers to Visual Dislocation, Anti. refers to Antithesis, Anal. refers to Analogy and Pers. refers to Personification.

# E  Additional Details of Results

## E.1  Answer Extraction

To mitigate the issue of models understanding the question but failing to output the correct option, we have implemented a robust answer extraction method and we use the same method for both CoT and non-CoT responses to ensure consistency. Specifically, our extraction method involves the following steps:

- Pattern Matching: We first search for the presence of option labels within the model's output using specific patterns.
- Frequency Analysis: If multiple option labels are found, we count their occurrences and select the most frequently occurring label.

## E.2  Statistics of Output

We do detailed statistics of the model output. The results are shown in Table 9 to 12. *Miss* is mainly caused by two situations, one is that the model does not give an answer, and the other is the regex is not matched. The *Miss* rate of most models is controlled below 2%, which is an acceptable ratio. In the *CoT* setting, some models do not follow instructions well and do not provide the expected letters as answer, which cannot be matched and will be considered a *Miss*. For convenience of presentation, some model names are abbreviated. The specific meanings of these abbreviations are consistent with the full model names used elsewhere in the paper.

| Mode | Metric | BLIP2-XL | BLIP2-XXL | CogVLM2 | DeepSeek | InsBLIP-XL |
|------|--------|----------|-----------|---------|----------|------------|
| CoT | Acc | 42.0 | 42.5 | 69.3 | 59.2 | 30.0 |
|  | Error | 0.0 | 0.0 | 0.0 | 0.0 | 0.0 |
|  | Miss | 11.7 | 15.8 | 0.0 | 0.2 | 11.7 |
| Domain | Acc | 51.4 | 57.5 | 69.1 | 60.4 | 47.8 |
|  | Error | 0.0 | 0.0 | 0.0 | 0.0 | 0.0 |
|  | Miss | 0.0 | 0.0 | 0.0 | 0.0 | 0.0 |
| Emotion | Acc | 51.8 | 58.4 | 71.7 | 63.3 | 49.8 |
|  | Error | 0.0 | 0.0 | 0.0 | 0.0 | 0.0 |
|  | Miss | 0.1 | 0.1 | 0.0 | 0.0 | 0.0 |
| None | Acc | 52.8 | 57.8 | 70.3 | 60.3 | 47.3 |
|  | Error | 0.0 | 0.0 | 0.0 | 0.0 | 0.0 |
|  | Miss | 0.0 | 0.0 | 0.0 | 0.1 | 0.0 |
| Rhetoric | Acc | 51.5 | 57.3 | 69.3 | 59.8 | 47.6 |
|  | Error | 0.0 | 0.0 | 0.0 | 0.0 | 0.0 |
|  | Miss | 0.0 | 0.0 | 0.0 | 0.0 | 0.0 |

Table 9: Accuracy, Error and Miss rate of different models under different settings.(1/4)

| Mode | Metric | InsBLIP-XXL | XComposer2 | InternVL | LLaVA-1.6 | MiniCPM-2.5 |
|---|---|---|---|---|---|---|
| CoT | Acc | 50.8 | 60.7 | 63.3 | 60.0 | 67.4 |
| | Error | 0.0 | 0.0 | 0.0 | 0.0 | 0.0 |
| | Miss | 2.2 | 2.3 | 0.1 | 12.4 | 0.0 |
| Domain | Acc | 56.7 | 60.9 | 66.6 | 73.1 | 70.3 |
| | Error | 0.0 | 0.0 | 0.0 | 0.0 | 0.0 |
| | Miss | 0.4 | 0.0 | 0.0 | 0.0 | 0.0 |
| Emotion | Acc | 58.7 | 61.5 | 67.4 | 75.3 | 70.8 |
| | Error | 0.0 | 0.0 | 0.0 | 0.0 | 0.0 |
| | Miss | 0.4 | 0.1 | 0.0 | 0.0 | 0.0 |
| None | Acc | 56.7 | 62.1 | 66.3 | 73.8 | 69.4 |
| | Error | 0.0 | 0.0 | 0.0 | 0.0 | 0.0 |
| | Miss | 0.4 | 0.0 | 0.0 | 0.1 | 0.0 |
| Rhetoric | Acc | 56.0 | 61.6 | 65.6 | 73.3 | 69.3 |
| | Error | 0.0 | 0.0 | 0.0 | 0.0 | 0.0 |
| | Miss | 0.4 | 0.0 | 0.0 | 0.0 | 0.0 |

Table 10: Accuracy, Error and Miss rate of different models under different settings.(2/4)

| Mode | Metric | mPLUGw-OWL2 | GPT-4o | Yi-VL-34B | Yi-VL-6B |
|---|---|---|---|---|---|
| CoT | Acc | 54.2 | 75.7 | 67.6 | 60.8 |
| | Error | 0.0 | 0.1 | 0.0 | 0.0 |
| | Miss | 0.2 | 10.7 | 0.0 | 0.0 |
| Domain | Acc | 54.5 | 72.6 | 67.7 | 60.8 |
| | Error | 0.0 | 0.0 | 0.0 | 0.0 |
| | Miss | 0.0 | 5.2 | 0.0 | 0.1 |
| Emotion | Acc | 55.0 | 74.2 | 70.1 | 62.8 |
| | Error | 0.0 | 0.1 | 0.0 | 0.0 |
| | Miss | 0.0 | 0.3 | 0.0 | 0.1 |
| None | Acc | 53.2 | 72.6 | 67.9 | 61.3 |
| | Error | 0.0 | 0.0 | 0.0 | 0.0 |
| | Miss | 0.0 | 0.2 | 0.0 | 0.0 |
| Rhetoric | Acc | 54.6 | 71.3 | 67.6 | 60.4 |
| | Error | 0.0 | 0.1 | 0.0 | 0.0 |
| | Miss | 0.0 | 0.1 | 0.0 | 0.0 |

Table 11: Accuracy, Error and Miss rate of different models under different settings.(3/4)

| Mode | Metric | GPT-4V | Qwen-Chat | Qwen-MAX | Gemini1.5 | Mantis | Idefics2 |
|---|---|---|---|---|---|---|---|
| CoT | Acc | 68.4 | 51.6 | 74.1 | 68.2 | 56.7 | 67.7 |
| | Error | 0.4 | 0.0 | 0.4 | 0.0 | 0.0 | 0.0 |
| | Miss | 0.4 | 10.7 | 0.1 | 0.3 | 0.0 | 0.1 |
| Domain | Acc | 66.0 | 54.9 | 74.1 | 73.1 | 57.1 | 67.0 |
| | Error | 0.1 | 0.0 | 0.6 | 0.0 | 0.0 | 0.0 |
| | Miss | 2.6 | 5.2 | 0.0 | 1.3 | 0.0 | 0.1 |
| Emotion | Acc | 68.3 | 57.0 | 75.5 | 70.5 | 57.0 | 68.6 |
| | Error | 0.6 | 0.0 | 0.6 | 0.0 | 0.0 | 0.0 |
| | Miss | 2.0 | 5.1 | 0.1 | 2.1 | 0.0 | 0.1 |
| None | Acc | 65.9 | 53.4 | 74.8 | 73.9 | 57.5 | 67.7 |
| | Error | 0.9 | 0.0 | 0.4 | 0.0 | 0.0 | 0.0 |
| | Miss | 4.2 | 7.0 | 0.1 | 1.3 | 0.0 | 0.0 |
| Rhetoric | Acc | 69.3 | 54.0 | 73.6 | 71.3 | 58.0 | 66.6 |
| | Error | 0.1 | 0.0 | 0.6 | 0.0 | 0.0 | 0.0 |
| | Miss | 1.9 | 6.4 | 0.4 | 1.1 | 0.0 | 0.0 |
| 1-shot | Acc | 65.5 | 43.3 | 74.5 | 73.2 | 55.3 | 64.1 |
| | Error | 0.3 | 0.0 | 0.4 | 0.1 | 0.0 | 0.0 |
| | Miss | 1.6 | 1.9 | 0.6 | 0.7 | 0.0 | 0.0 |
| 2-shot | Acc | 67.7 | 47.9 | 69.6 | 73.8 | 54.2 | 62.4 |
| | Error | 0.2 | 0.0 | 0.4 | 0.1 | 0.0 | 0.0 |
| | Miss | 0.5 | 5.7 | 1.0 | 0.4 | 0.0 | 0.0 |
| 3-shot | Acc | 67.1 | 41.1 | 53.6 | 74.1 | 54.9 | 59.5 |
| | Error | 0.6 | 0.0 | 16.7 | 0.0 | 0.0 | 0.0 |
| | Miss | 0.4 | 5.4 | 0.1 | 0.2 | 0.0 | 0.0 |

Table 12: Accuracy, Error and Miss rate of different models under different settings.(4/4)

# F  Other Errors

**Detail Ignorance (11%):** Detail Ignorance refers to GPT-4V overlooking certain crucial details or elements in images. This oversight can sometimes lead to 'Metaphorical Misunderstanding.' Many images convey implications and meanings through their details. By fully utilizing the content of an image and not ignoring any details, one can better understand and uncover the hidden meanings within the image.

**Surface-Level Interpretation (15%):** This error occurs when GPT-4V focuses only on the superficial information in images while ignoring their deeper meanings. It may only offer a superficial interpretation of the images, failing to grasp their complexity or multifaceted meanings. For example, in Fig.G12, GPT-4V noticed only the surface-level information of 'Internet' and overlooked the deeper element of 'the emotional value of the meme,' thus providing an incorrect response.

**Reasoning Error (12%):** Reasoning errors may arise even when GPT-4V accurately understands the content of an image, such as in the illustration of Fig.G18. In such instances, errors occur during complex problem-solving that demands advanced logical and mathematical reasoning. This type of error often stems from the model's limited capabilities in handling intricate logic and mathematics, highlighting areas where GPT-4V requires further refinement to improve its reasoning accuracy.

**Reject to Answer (4%):** Reject to Answer is a common error encountered in GPT-4V. This phenomenon typically manifests for two primary reasons. First, the model may determine that the information provided in the question and accompanying images is insufficient to deduce the underlying meanings or implications, thereby rendering it incapable of ascertaining the correct answer, such as Fig.G38. The second reason involves the model's evaluation of the content as potentially harmful or inappropriate. In such cases, the model opts to withhold a response as a preventive measure against disseminating sensitive or damaging information. This safety mechanism is crucial in maintaining ethical standards and preventing the propagation of harmful content. However, this can also lead to frustrations when users expect a response that the model is programmed to avoid for ethical reasons.

**Answer Extraction Error (1%):** Answer Extraction Error refers to the phenomenon of extracting incorrect answers using a regularization formula from GPT-4V's output. This issue may arise due to GPT-4V's weak instruction-following capabilities on certain questions, where it fails to generate answers in the correct format according to the rules.

# G Case Study

The appendix is our sample analysis of GPT-4V, including an analysis of 49 error examples and 28 correct examples.

**List of Case Study Figures**

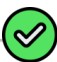

## Life

**Question:** What does the caption "YES, BUT" imply about travel? <image 1>
**Option:**
(A) Cultural experiences are not affected by the traveler's level of immersion.
(B) Travel experiences are homogenous and offer similar cultural depths across all destinations.
(C) The physical arrival of travel does not equate to the depth of cultural experience.
(D) Tourist attractions are the core of cultural experience when traveling.
(E) Physical presence in a location is all that is required for a meaningful travel experience.
(F) Engagement with the culture of a travel destination is not necessary for a fulfilling travel experience.

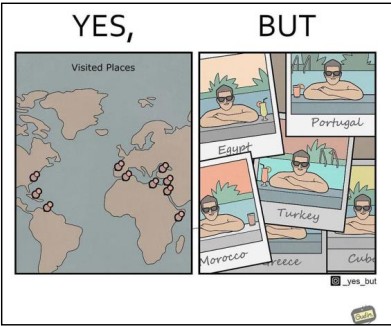

<image 1>

**GPT-4V:** The caption "YES, BUT" in the context of the picture, which shows a map with marked places and snapshots of ostensibly the same person in different locations but with their presence obscured, implies: (C) The physical arrival of travel does not equate to the depth of cultural experience.

**Ground Truth: (C) The physical arrival of travel does not equate to the depth of cultural experience.**

Figure G1: A sample correct case of *Life* domain.
Back to List of figures

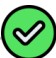

## Life

**Question:** What is the message of repetitive depictions of adult life in comics? <image 1>
**Option:**
(A) These comics are meant to illustrate how every day in adult life is uniquely exciting.
(B) The intent is to show how adults have ample free time for hobbies and personal interests.
(C) These depictions aim to highlight the unpredictable and constantly changing nature of adult life.
(D) Adult life is often seen as a monotonous cycle that repeats itself.
(E) These comics are trying to communicate that adults always have clear and simple decisions to make.
(F) They suggest that adult life is full of joyous adventures and new discoveries every day.

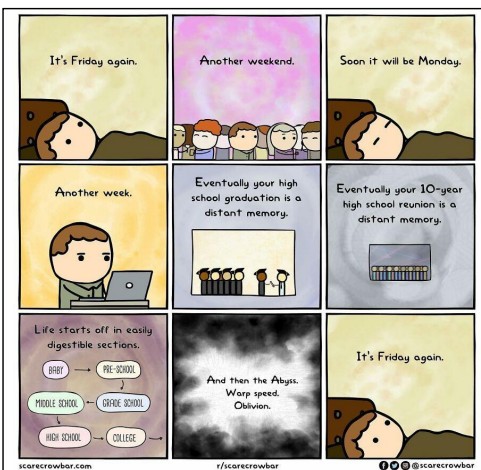

<image 1>

**GPT-4V:** The message of repetitive depictions of adult life in this comic seems to be (D) Adult life is often seen as a monotonous cycle that repeats itself. The comic shows a sequence of recurring events and the concept of time passing quickly, with little emphasis on new experiences or excitement, suggesting a routine that continuously loops with a sense of inevitability.

**Ground Truth: (D) Adult life is often seen as a monotonous cycle that repeats itself.**

Figure G2: A sample correct case of *Life* domain.
Back to List of figures

**Life**

**Question:** The metaphor for this image is? <image 1>
**Option:**
(A) The puzzle pieces symbolize the unresolved issues and conflicts between the father and son, indicating a broken relationship.
(B) The missing puzzle pieces in the father figure represent the parts of his life and experiences that he imparts to his son, helping to shape the son's identity and future.
(C) The pieces represent the cultural and family traditions the father wishes to preserve and pass down to his son.
(D) The puzzle pieces represent the father's dreams and aspirations for his son, hoping he achieves more than he did.
(E) They signify the genetic traits and characteristics inherited from the father, emphasizing the biological bond between them.
(F) They indicate the fragmented memories and experiences the father shares with his son, bonding them together.

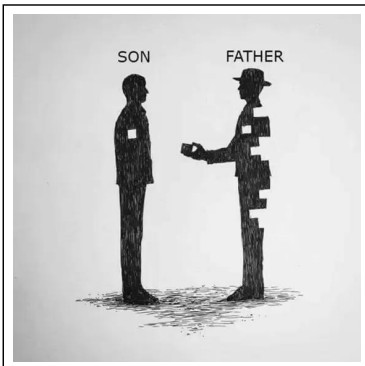

<image 1>

**GPT-4V:** The image shows two silhouettes, one labeled "SON" and the other labeled "FATHER." The father figure has puzzle pieces missing, and one piece is being handed over to the son. This could be open to interpretation, but given the choices you've provided:
(B) The missing puzzle pieces in the father figure represent the parts of his life and experiences that he imparts to his son, helping to shape the son's identity and future.
(B) seems to be the most fitting metaphor based on the visual representation. The act of handing over a puzzle piece suggests the father is passing down something to his son.

**Ground Truth: (B) The missing puzzle pieces in the father figure represent the parts of his life and experiences that he imparts to his son, helping to shape the son's identity and future.**

Figure G3: A sample correct case of *Life* domain.
Back to List of figures

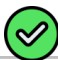 **Life**

**Question:** What hidden meaning can be inferred from the juxtaposition of the optimistic phrase "FAILURE IS NOT AN OPTION" with the ironic addendum "IT'S MY WHOLE LIFE" in the context of the cheerful illustration of the man? <image 1>

**Option:**

(A) The juxtaposition reveals a critique of the overemphasis on positivity and success, pointing out the unrealistic nature of such expectations.

(B) It suggests a critique of the unrealistic standards of success in society while promoting acceptance and humor in the face of personal shortcomings.

(C) The phrase is a commentary on the relentless pursuit of success, which often overlooks the integral role of failure in personal growth.

(D) It underscores the importance of persistence in the face of challenges, suggesting that enduring failure is a necessary step towards eventual success.

(E) The illustration conveys a message that while society may push for constant success, embracing one's full narrative, including failures, is crucial for genuine happiness.

(F) It expresses a form of self-deprecation used as a coping mechanism to deal with the pressures of achieving success in a competitive society.

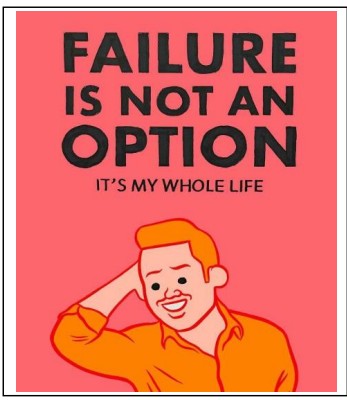

<image 1>

**GPT-4V:** The hidden meaning that can be inferred from the juxtaposition of the optimistic phrase "FAILURE IS NOT AN OPTION" with the ironic addendum "IT'S MY WHOLE LIFE" in the context of the cheerful illustration of the man, which displays an individual striking a pose that could imply confidence or self-assurance, seems to suggest a critique of the unrealistic standards of success in society while promoting acceptance and humor in the face of personal shortcomings. Therefore, the most appropriate answer from the given options that reflects this sentiment would be:
(B) It suggests a critique of the unrealistic standards of success in society while promoting acceptance and humor in the face of personal shortcomings.

**Ground Truth: (B) It suggests a critique of the unrealistic standards of success in society while promoting acceptance and humor in the face of personal shortcomings.**

Figure G4: A sample correct case of *Life* domain.

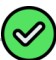

**Life**

**Question:** Based on the analysis of this comic, what is the implied meaning conveyed in the picture? <image 1>

**Option:**

(A) It suggests that working overtime is a festive tradition and celebrated within the company culture.

(B) It suggests that the character is choosing to work overtime voluntarily for personal satisfaction.

(C) The comic illustrates the joy and celebration of holidays within the corporate environment.

(D) The comic reflects the frustration of a person who still needs to work overtime at the company during the holidays.

(E) The character's frustration is actually a disguise for their excitement about holiday work.

(F) The comic is an ode to workaholics who find joy in their dedication to work during holidays.

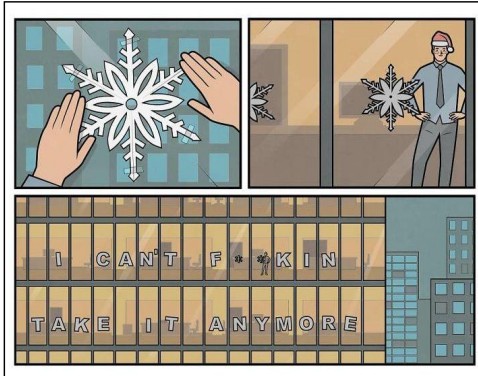

<image 1>

**GPT-4V:** Based on the analysis of this comic, the implied meaning conveyed in the picture is:

(D) The comic reflects the frustration of a person who still needs to work overtime at the company during the holidays.

The image shows a message in an office window, displaying a profanity-laden declaration of being unable to take it anymore, which suggests the character's frustration rather than a celebration or joy regarding working during the holidays.

**Ground Truth: (D) The comic reflects the frustration of a person who still needs to work overtime at the company during the holidays.**

Figure G5: A sample correct case of *Life* domain.

Back to List of figures

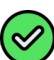

**Life**

**Question:** What could the depicted skeleton symbolize? <image 1>
**Option:**
(A) Endless curiosity and the never-ending pursuit of knowledge.
(B) Symbolizing the eternal life of knowledge.
(C) The joy found in constant academic discovery.
(D) People's sense of boredom and emotional exhaustion from repetitive studying.
(E) A representation of the wisdom gained from lifelong learning.
(F) The exhilaration of learning new things.

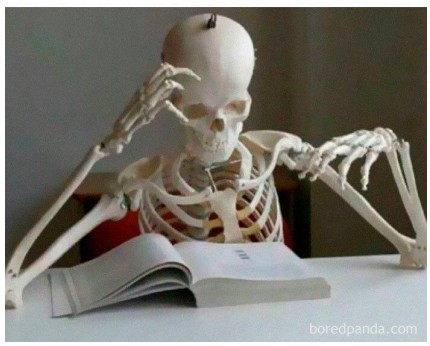

<image 1>

**GPT-4V:** The depicted skeleton could symbolize (D) People's sense of boredom and emotional exhaustion from repetitive studying. The image shows a skeleton looking at an open book, and the caption refers to the situation of having to read the same page repeatedly due to a lack of concentration, which is often associated with boredom or fatigue.

**Ground Truth: (D) People's sense of boredom and emotional exhaustion from repetitive studying.**

Figure G6: A sample correct case of *Life* domain.
Back to List of figures

**Question:** What metaphorical concept is humorously illustrated in the comic? <image 1>
**Option:**
(A) The evolution and complexity of fragrance production technologies.
(B) A symbol of the often overpowering and unappealing nature of cologne worn by men.
(C) It illustrates a satirical commentary on the absurdity and confusion caused product naming tactics.
(D) A playful critique of societal expectations regarding personal hygiene and fragrance.
(E) A reflection on societal expectations of hygiene and personal scent.
(F) A critique of the overuse of colognes and perfumes in modern society.

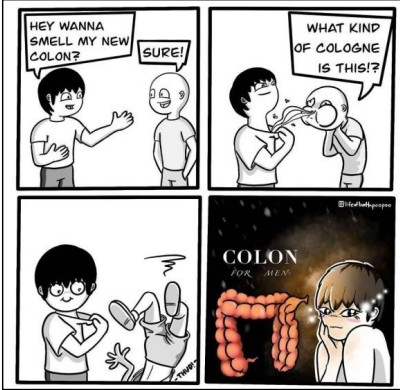

<image 1>

**GPT-4V:** The metaphorical concept humorously illustrated in the comic is:
**(B) A symbol of the often overpowering and unappealing nature of cologne worn by men.**
The comic plays on the word "cologne," which is homophonous with "colon" in English, and the punchline revolves around the literal interpretation of colon, suggesting an overpowering and unpleasant scent associated with it, which humorously criticizes some colognes worn by men.

**Error Category: Metaphorical Misunderstanding**
**Error Reason:** GPT-4V analyzed the connection between "colon" and "cologne", but only understood the surface meaning of the picture, lacked in-depth reasoning, and incorrectly interpreted the metaphor as (B).

**Ground Truth: (C) It illustrates a satirical commentary on the absurdity and confusion caused product naming tactics.**

Figure G7: A sample error case of *Life* domain.

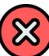

## Life

**Question:** Why does the character have four expressive masks when his face is expressionless?
<image 1>

**Option:**

(A) The character is a master of disguise, using the masks for espionage and intelligence gathering without revealing his real appearance.

(B) He believes the masks hold magical properties that can protect him from negative energies and prefers to wear them rather than showing his own face.

(C) The character uses the masks as tools for performing in plays, needing to quickly shift between roles without relying on facial expressions.

(D) He is part of a secret society where members must conceal their identity at all times, so he uses the masks to hide his true self.

(E) The masks serve as a metaphor for the various roles he plays in his life, symbolizing his adaptability and multifaceted personality.

(F) This may represent the character trying to hide his or her true emotions, using a mask to cover up inner sadness or pain.

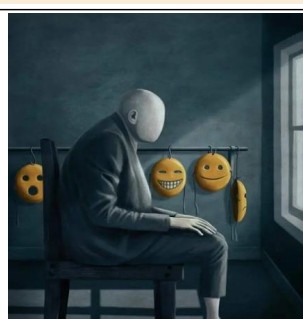

<image 1>

**GPT-4V:** In the image provided, we see a character with an expressionless face sitting next to four expressive masks. Each option suggests a different interpretation of this scene. However, since I am to give my best guess based on the image:

**(E) The masks serve as a metaphor for the various roles he plays in his life, symbolizing his adaptability and multifaceted personality.**

This choice seems the most fitting given the contrast between the character's blank expression and the varied emotions displayed on the masks. It suggests a depth and complexity to the character's experiences or behavior that is not immediately visible from their outward appearance. The setting is somewhat somber, which may imply an introspective or contemplative mood, further aligning with this metaphorical interpretation.

**Error Category: Metaphorical Misunderstanding**

**Error Reason:** GPT-4V understood the surface meaning of the comic — a character with an expressionless face sitting next to four expressive masks. GPT-4V's analysis of the mask was correct. The mask is a metaphor for the various roles the protagonist plays in life.

However, the image does not convey the adaptability and multifaceted personality of the protagonist. Based on the gloomy background and depressing color palette of the picture, the purpose is to create a sad atmosphere. Therefore, the mask may be used by the protagonist to hide inner sadness or pain. The correct metaphorical interpretation of this comic is (F).

**Ground Truth: (F) This may represent the character trying to hide his or her true emotions, using a mask to cover up inner sadness or pain.**

Figure G8: A sample error case of *Life* domain.
Back to List of figures

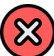

## Life

**Question:** Which aspect of scientific research is implied in this picture? <image 1>
**Option:**
(A) The influence of personal life events on scientific productivity.
(B) The debate over the quantization of energy and its reception in the scientific community.
(C) Real scientists are very serious about research, and every scientific discovery is not easy.
(D) The role of technological advancements in the progression of scientific research.
(E) The dichotomy between theoretical and practical implications in Planck's work.
(F) The constant evolution of scientific methodologies over time.

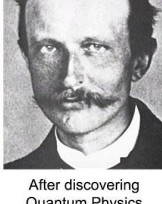

Theoretical physicist
Max Planck

Before discovering
Quantum Physics

After discovering
Quantum Physics

<image 1>

**GPT-4V:** The picture implies **(B) The debate over the quantization of energy and its reception in the scientific community**, as it references Max Planck, who was a key figure in the development of quantum physics, which involves the concept of quantized energy.

**Error Category: Metaphorical Misunderstanding, Detail Ignorance**
**Error Reason:** GPT-4V failed to understand the contrast between the two sets of pictures and mistakenly understood the picture metaphor as (B). Based on the image and the explanation, we can clearly see that the appearance of the scientist has changed greatly after studying science, reflecting the picture metaphor is (C).

**Ground Truth: (C) Real scientists are very serious about research, and every scientific discovery is not easy.**
**Explanation:** By comparing the appearance of scientists before and after scientific research, we can deduce that the discovery of quantum physics is difficult and requires a long period of struggle and hard work.

Figure G9: A sample error case of *Life* domain.
Back to List of figures

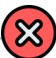

# Life

**Question:** Which psychosocial phenomenon does this illustration reveal? <image 1>
**Option:**
(A) The influence of individual positivity on collective mood and behavior.
(B) The bystander effect in crisis situations.
(C) The formation of in-group versus out-group biases.
(D) The effect of authoritarian leadership on group morale.
(E) The role of intrinsic motivation in personal satisfaction.
(F) The impact of social proof on conformity.

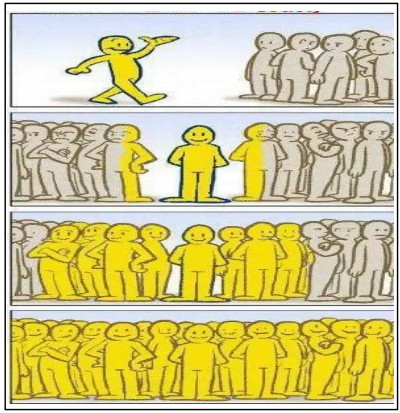

<image 1>

**GPT-4V:** Answer: **(F) The impact of social proof on conformity.**
The illustration appears to show a sequence where one individual's behavior is initially distinct from that of the group, but gradually the group members begin to adopt the same behavior as the individual. This suggests that the group's actions are influenced by the behavior of one member, reflecting the impact of social proof on conformity, where individuals in a group look to each other for cues on how to behave, often leading to a conformity effect.

**Error Category: Metaphorical Misunderstanding, Detail Ignorance**
**Error Reason:** GPT-4V understood part of the surface meaning of the picture — Group members gradually begin to adopt the same behavior as individuals. However, GPT-4V ignores the expression changes of the characters. The emotional changes of the group can be found by comparing the before and after expressions of the characters in the picture.. Therefore GPT-4V only analyzes changes in behavior and incorrectly chooses (F).

**Ground Truth: (A) The influence of individual positivity on collective mood and behavior.**
**Explanation:** This picture shows the effect of individual positivity on the group through the behavior and expressions of the group before and after.

Figure G10: A sample error case of *Life* domain.
Back to List of figures

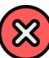

## Life

**Question:** What hidden meaning can be interpreted from the contrast between the two panels of the image, particularly centering around the "Pull" sign on the door? <image 1>

**Option:**

(A) It implies the dilemma of offering public accessibility while safeguarding personal privacy and security.

(B) It points to the contradiction between advocating for health safety and the lack of supportive infrastructure.

(C) It highlights the disparity between the intention to engage in environmental sustainability and the practical obstacles presented by public facilities.

(D) It emphasizes the importance of social distancing measures, yet the infrastructure does not accommodate it.

(E) It signifies the necessity of maintaining personal hygiene rituals but public design fails to support it.

(F) It exposes the irony of emphasizing cleanliness in public messaging while failing to provide adequate facilities for it.

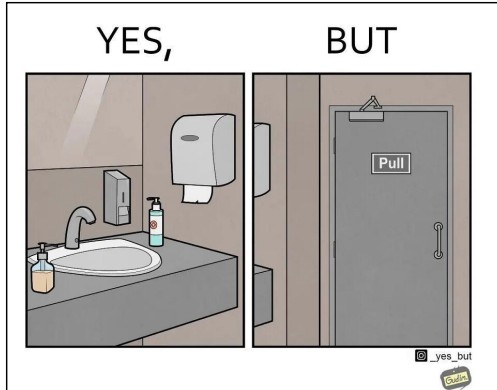

<image 1>

**GPT-4V:** The image depicts a contrast between a clean handwashing station with soap and the act of having to pull a door to exit, potentially after having just cleaned one's hands. This presents an irony since the infrastructure doesn't seem to support the hygiene practice completely by potentially causing recontamination when touching the door handle. The most fitting answer choice to this scenario would be:

**(F) It exposes the irony of emphasizing cleanliness in public messaging while failing to provide adequate facilities for it.**

**Error Category: Detail Misunderstanding, Reasoning Error**

**Error Reason:** GPT-4V correctly understood the metaphorical content of the comic, that is, the infrastructure cannot fully support the promoted personal hygiene practices. However, GPT-4V has an error in understanding the details. The picture shows hand sanitizer instead of soap. And there were errors in logical reasoning, the picture metaphor could not reason about the irony of not providing adequate facilities. The option derived from correct logical reasoning should be (E).

**Ground Truth: (E) It signifies the necessity of maintaining personal hygiene rituals but public design fails to support it.**

Figure G11: A sample error case of *Life* domain.
Back to List of figures

## Life

**Question:** What hidden meaning can be discerned from the transition between using the internet for entertainment to sharing a hand-drawn meme in the narrative of the comic? 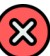

**Option:**

(A) It conveys a misplaced nostalgia for less effective forms of expression that disregard the benefits of the internet.

(B) It hints at a romanticized view of manual effort that overlooks the practical shortcomings of non-digital solutions.

(C) It reflects a personal challenge against the dominant culture of digital immediacy, preferring slower forms of sharing.

(D) It suggests a regression in communication methods, favoring outdated techniques over advanced digital tools.

(E) It implies that human adaptation can find ways to fulfill emotional needs, even without modern technology.

(F) It implies a resistance to embracing the global reach and inclusivity facilitated by the internet.

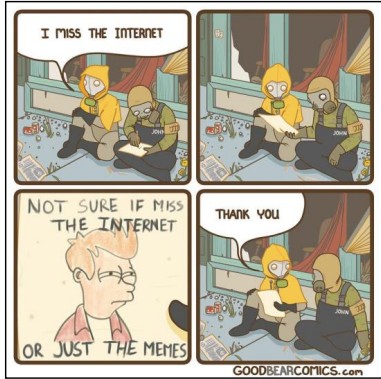

<image 1>

**GPT-4V:** Based on the picture provided, the hidden meaning that can be discerned from the transition between using the internet for entertainment to sharing a hand-drawn meme in the narrative of the comic is:

**(C) It reflects a personal challenge against the dominant culture of digital immediacy, preferring slower forms of sharing.**

The narrative in the comic strip shows a character initially missing the internet and then experiencing a personal connection through the sharing of a hand-drawn meme. This intimates a preference for a more personal and tangible form of sharing over the instantaneous digital method, highlighting the value of personal effort and connection.

**Error Category: Surface-Level Interpretation**

**Error Reason:** GPT-4V understood the surface meaning of the picture—a character missing the internet and then sharing a hand-drawn meme. But it misunderstood the metaphorical content and chose (C). The picture is a metaphor for gaining emotional support for sharing a meme, whether in person or on the Internet.

**Ground Truth: (E) It implies that human adaptation can find ways to fulfill emotional needs, even without modern technology.**

Figure G12: A sample error case of *Life* domain.
Back to List of figures

**Life**

**Question:** What does the pile of food in the dog's dream symbolize in the comic strip? <image 1>
**Option:**
(A) A metaphor for the dog's hidden talents and unexplored potential.
(B) The wealth and prosperity that the dog aspires to achieve in the future.
(C) The dog's past experiences of going on a picnic with its owner.
(D) The dog's dissatisfaction with its regular diet and a desire for more variety.
(E) The owner's neglect and the dog's feelings of abandonment.
(F) The significance of material comforts that seem unfulfilling without emotional bonds.

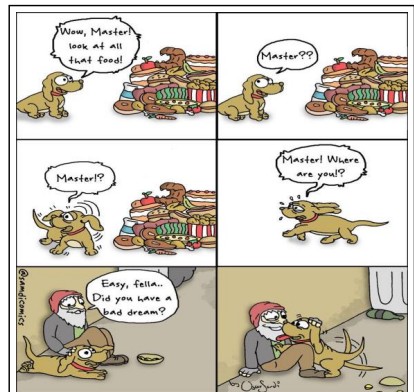
<image 1>

**GPT-4V:** Based on the picture provided, the pile of food in the dog's dream symbolizes: **(D) The dog's dissatisfaction with its regular diet and a desire for more variety.**
The comic strip shows the dog dreaming of a large pile of various delicious foods, but the dream shifts to a feeling of panic when the dog can't find its owner. This suggests that the food represents something the dog wishes for but realizes that it isn't as important as the companionship and presence of its owner.

**Error Category: Surface-Level Interpretation, Metaphorical Misunderstanding**
**Error Reason:** GPT-4V understood the surface meaning of the picture — The dog dreamed of delicious things but could not find its owner. However, GPT-4V only focused on food and ignored the emotional metaphor of the picture, and mistakenly interpreted the delicious food that the dog dreamed of as (D).

**Ground Truth: (F) The significance of material comforts that seem unfulfilling without emotional bonds.**
**Explanation:** The picture tells us that emotional connection is more meaningful than material comfort through the contrast between the dog's love for delicious food and the abandonment of food because he cannot find his owner.

Figure G13: A sample error case of *Life* domain.
Back to List of figures

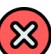

**Life**

**Question:** What societal commentary might be inferred from the repetitive nature of the doctor's instructions to "breathe out" without allowing for the patient to breathe in, as depicted in the comic strip? <image 1>

**Option:**
(A) It suggests the societal value placed on continuous productivity at the expense of personal health and wellbeing.
(B) It hints at the medical industry's disregard for patient autonomy and the importance of following professional advice without question.
(C) It exposes the dichotomy between the push for unrelenting productivity and the critical importance of self-care and relaxation.
(D) It shows the contrast between the need for continuous progress and performance versus the essential human requirement for downtime.
(E) It represents the overwhelming demands modern society places on individuals without giving them a chance to recover.
(F) It highlights the unrealistic demands society places on its members to always be active, neglecting the need for periods of stillness and recovery.

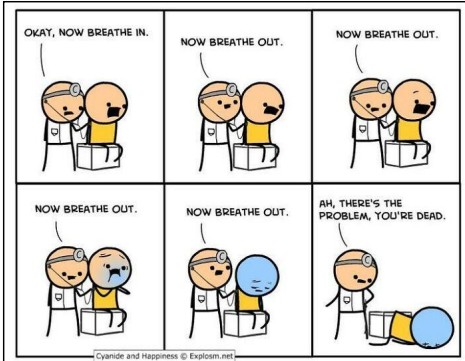

<image 1>

**GPT-4V:** All the options (A, B, C, D, E, F) could potentially be inferred from the comic strip, as they all touch on themes of unrealistic societal expectations, neglect of personal well-being, and the pressure to be constantly productive. However, the most direct commentary based on the imagery and text in the comic strip seems to be **option (F)**, as it directly addresses the impossibility of continuously exhaling (i.e., being productive) without the opportunity to inhale (i.e., rest and recover).

**Error Category: Surface-Level Interpretation , Metaphorical Misunderstanding**
**Error Reason:** GPT-4V understood the surface meaning of the comic — The doctor made the patient exhale and eventually led to his death. However, GPT-4V only noticed the superficial meaning of comic without deeply understanding the comic metaphor, and made the wrong choice (F).
The comic uses black humor to compare the overwhelming demands placed on individuals in modern society to a doctor asking a patient to keep breathing out, satirizing its irrationality. The correct metaphorical interpretation of this comic is (E).

**Ground Truth: (E) It represents the overwhelming demands modern society places on individuals without giving them a chance to recover.**

Figure G14: A sample error case of *Life* domain.
Back to List of figures

**Question:** What hidden meaning might be suggested by the explicit rejection of the inmate's last meal request for "just pineapple" on a pizza in the final panel of the comic? <image 1>

**Option:**

(A) It implies a critique on the artificial standards of purity in culinary tastes, mocking the rigidity of traditional food choices.

(B) The rejection represents people's condemnation of unconventional choices, even in the most personal matters.

(C) The pineapple symbolizes exotic desires that are too strange for society to accept, hinting at the broader rejection of foreign influences.

(D) The denial underscores the character's isolation from societal norms, highlighting a deeper societal fear of the unknown.

(E) The refusal reflects a misunderstanding between different generations' preferences, suggesting a gap that cannot be bridged.

(F) The rejection of the pizza request symbolizes the trivialization of individual preferences in the face of collective judgments.

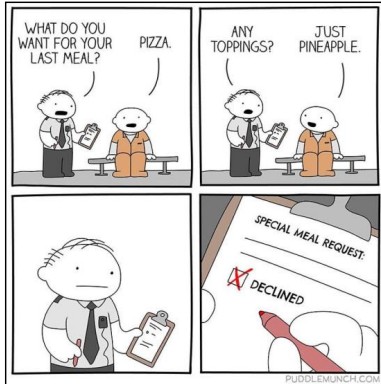

<image 1>

**GPT-4V:** The comic depicts a conversation between a prison inmate and what appears to be a prison official regarding the inmate's last meal request. The inmate requests a pizza with "just pineapple" as a topping, which is declined in the final panel. The hidden meaning suggested by this explicit rejection could be:

**(F) The rejection of the pizza request symbolizes the trivialization of individual preferences in the face of collective judgments.**

This choice seems to capture the essence of the situation: an individual's unusual choice, in this case, a polarizing pizza topping, is not honored, perhaps reflective of society's reluctance to accept unconventional preferences.

**Error Category: Reasoning Error**

**Error Reason:** GPT-4V correctly understood the metaphorical content of the comic, that is, society's reluctance to accept unconventional preferences. However, there were errors in logical reasoning and it was unable to obtain the perspective of the trivialization of individual preferences in the face of collective judgments. The option derived from correct logical reasoning should be (B).

**Ground Truth: (B) The rejection represents people's condemnation of unconventional choices, even in the most personal matters.**

Figure G15: A sample error case of *Life* domain.
Back to List of figures

**Question:** In the humorous twist of the final panel of the comic, what hidden meaning can be derived from Edgar the dog's statement? <image 1>

**Option:**

(A) The quote could be misunderstood as suggesting pets enjoy human misery rather than them recognizing and responding to emotional cues.

(B) Edgar's words could be taken to mean that pets have a mischievous nature, enjoying human discomfort as a form of entertainment.

(C) It hints at the dark side of the puppy and is an expression of black humor. The puppy may not want to comfort the owner, but just likes to see the owner sad.

(D) It implies a lack of empathy in animals, suggesting they find humor in human sadness rather than offering comfort.

(E) Edgar's statement reveals that animals, particularly pets, have a complex understanding of human emotions and provide comfort with a hint of humor.

(F) Edgar's statement suggests that animals are indifferent to human emotions, focusing solely on their own needs.

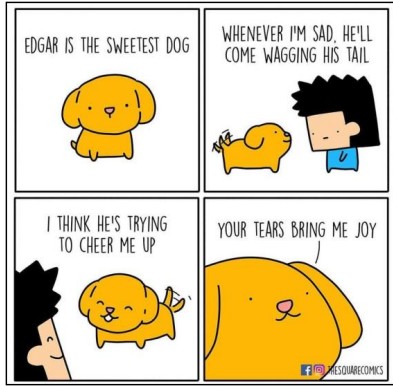

<image 1>

**GPT-4V:** Based on the image provided, the humorous twist in the final panel of the comic where Edgar the dog says "Your tears bring me joy" can be derived as the following hidden meaning:

**(A) The quote could be misunderstood as suggesting pets enjoy human misery rather than them recognizing and responding to emotional cues.**

Edgar's statement is a play on the typical behavior of pets like dogs that are known to provide comfort when their owners are sad. However, the joke subverts this expectation by suggesting that Edgar seems to enjoy the owner's sadness, giving the statement a humorous twist that contrasts with pets' usual empathetic nature.

**Error Category: Reasoning Error**

**Error Reason:** GPT-4V correctly understood the metaphorical content of the comic, that is, dogs provide comfort when their owners are sad. This comic uses black humor to playfully express typical dog behaviors. However, there were errors in logical reasoning, the dark humor failed to reason that pets enjoy human misery and dogs cannot recognize and respond to emotional cues. The option derived from correct logical reasoning should be (C).

**Ground Truth: (C) It hints at the dark side of the puppy and is an expression of black humor. The puppy may not want to comfort the owner, but just likes to see the owner sad.**

Figure G16: A sample error case of *Life* domain.
Back to List of figures

**Question:** What hidden meaning can be inferred from the character's action of sweeping up broken glass in the last panel of the comic? <image 1>

**Option:**

(A) It indicates the character's hope to find something valuable among the shattered remnants.

(B) It symbolizes the character's attempt to piece back together a relationship that's been damaged.

(C) It conveys the character's belief in the importance of facing the aftermath of one's actions head-on.

(D) It depicts the character's struggle with accepting the irreversible nature of certain decisions.

(E) It shows the character's desire to correct mistakes from the past by literally cleaning up.

(F) It signifies that rebellious behavior often leads to consequences that one must eventually address.

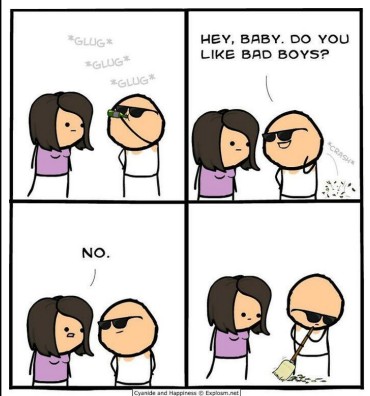

<image 1>

**GPT-4V:** Based on the last panel of the comic, where the character is sweeping up broken glass, the hidden meaning that can be inferred from the character's action is:

**(D) It depicts the character's struggle with accepting the irreversible nature of certain decisions.**

This seems to suggest that after attempting to portray themselves as a "bad boy" and not getting the desired response, they are left to deal with the aftermath of their actions, which in this case is quite literal with the broken bottle.

**Error Category: Reasoning Error**

**Error Reason:** GPT-4V correctly understood the metaphorical content of the comic, that is, the need to deal with the consequences of one's actions after an act of rebellion. However, there were errors in logical reasoning, and the character's struggle with accepting the irreversible nature of certain decisions was not reflected in the metaphor. The option derived from correct logical reasoning should be (F).

**Ground Truth: (F) It signifies that rebellious behavior often leads to consequences that one must eventually address.**

Figure G17: A sample error case of *Life* domain.
Back to List of figures

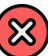

## Life

**Question:** What does the silhouette of the woman most likely symbolize in the picture? <image 1>
**Option:**
(A) The silhouette is meant to symbolize a creative spirit, suggesting a passion for art and cultural expression.
(B) The silhouette portrays an ambitious career-oriented individual, suggesting a woman's professional growth and drive.
(C) The silhouette of the woman most likely symbolizes independence and personal freedom, emphasizing the importance of self-identity.
(D) The silhouette symbolizes adventure and exploration, reflecting a desire to travel and experience new horizons.
(E) The silhouette conveys the concept of individuality, representing women's right to pursue their unique goals and aspirations.
(F) The silent supportive role of a mother in the family.

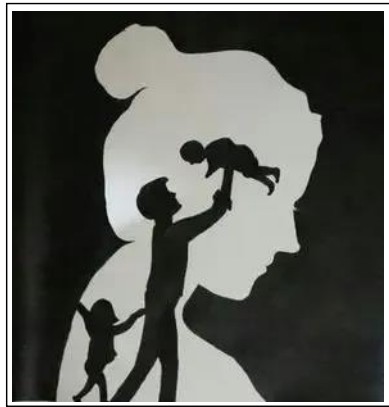

<image 1>

**GPT-4V:** The silhouette of the woman in the picture most likely symbolizes **(C) independence and personal freedom, emphasizing the importance of self-identity.** The image of a woman with a visible inner silhouette of her caring for children might also suggest the multifaceted role of women encompassing both personal identity and their traditional role in nurturing and caring for a family, which relates to (F) the silent supportive role of a mother in the family. Based on this image alone, it is not explicitly indicative of a creative spirit, professional ambition, adventure and exploration, or the concept of unique individual goals, so options (A), (B), (D), and (E) would not be as strongly supported.

**Error Category: Reasoning Error**
**Error Reason:** GPT-4V correctly understood the metaphorical content of the comic, that is, the traditional role of women taking care of the family and silently supporting it. However, there were errors in logical reasoning, and the idea of female independence and personal freedom cannot be derived from the metaphor of the picture. The pictures show more of the family than the individual. The option derived from correct logical reasoning should be (F).

**Ground Truth: (F) The silent supportive role of a mother in the family.**

Figure G18: A sample error case of *Life* domain.
Back to List of figures

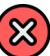

## Life

**Question:** In the illustration, the anthropomorphized lamp directs a beam of yellow light upon the individual. What hidden meaning might this spotlight represent? <image 1>

**Option:**

(A) It could be seen as a warning or foresight, suggesting the character is about to face significant challenges or decisions.

(B) It could be interpreted as the glare of public attention, focusing on the external rather than the internal transformation of the character.

(C) It represents the character's fear of being exposed or scrutinized by society, suggesting a sense of vulnerability or shame.

(D) Enlightenment and self-realization.

(E) It may symbolize the character's isolation, as if they are under a spotlight that separates them from their surroundings and peers.

(F) The spotlight may represent a moment of fame or recognition, distracting from the deeper, introspective journey of the individual.

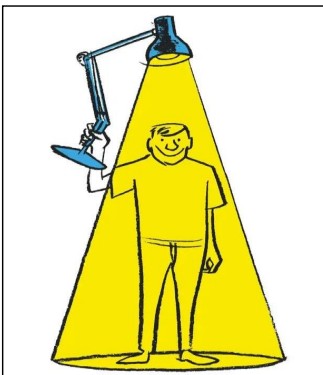

<image 1>

**GPT-4V:** Based on the provided illustration, the anthropomorphized lamp casting a yellow light upon the individual could be open to various interpretations. Since the individual appears to be standing calmly and smiling under the spotlight, this suggests a more positive connotation. Therefore, the hidden meanings of the spotlight could potentially be:

**(A) It could be seen as a warning or foresight, suggesting the character is about to face significant challenges or decisions.**

Options (B), (C), and (E) suggest a more negative experience or feeling, which the character's relaxed and happy demeanor does not reflect, thus making them less likely interpretations in this context.

**Error Category: Reasoning Error**

**Error Reason:** GPT-4V correctly understood the metaphorical content of the comic, that is, a person standing in the spotlight and smiling, indicating a positive meaning. However, there were errors in logical reasoning, and the meaning of spotlight is a warning cannot be taken from the metaphor of the picture. It is impossible to deduce that the characters are about to face a major challenge or decision. What we can get from the picture is that light can enlighten the characters' thoughts and is a positive expression. The option derived from correct logical reasoning should be (D).

**Ground Truth: (D) Enlightenment and self-realization.**

Figure G19: A sample error case of *Life* domain.
Back to List of figures

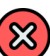

**Life**

**Question:** What hidden meaning could be interpreted from the grandmother's dual portrayal in the comic strip? <image 1>

**Option:**

(A) The comic humorously conveys the multifaceted nature of humanity, illustrating that everyone can exhibit vastly different traits in varying situations.

(B) This contrast emphasizes that even seemingly simple characters may possess complex inner worlds and unknown sides

(C) Through this unexpected behavior, the comic creates a humorous effect, prompting reflections on the disparity between the traditional image of the elderly and their actual actions.

(D) The grandmother's dual life could be that of a secret agent, depicting that even older individuals can lead thrilling and adventurous lives.

(E) By showing the grandmother as gentle in front of her grandson and tough at a crime scene, it challenges traditional views on the personality and behavior of older people.

(F) It emphasizes the concept that individuals may have hidden aspects to their personalities or lives that are in stark contrast to their outward appearance.

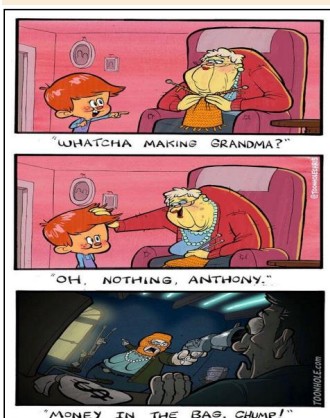

<image 1>

**GPT-4V:** Based on the picture provided, the hidden meaning that could be interpreted from the grandmother's dual portrayal in the comic strip is:

**(C) Through this unexpected behavior, the comic creates a humorous effect, prompting reflections on the disparity between the traditional image of the elderly and their actual actions.**

This choice is most accurate because the comic strip directly shows the grandmother in two contrasting scenarios—first, as a benign grandmother figure with her grandson and then as a tough individual involved in some sort of crime or secret operation. The humorous twist lies in the unexpected subversion of the typical, gentle elderly image.

**Error Category: Reasoning Error**

**Error Reason:** GPT-4V correctly understood the metaphorical content of the comic, that is, the unexpected subversion of the image of the grandmother—from kind to tough. However, there were errors in logical reasoning, and the picture metaphor could not lead to reflection on the gap between the image and actual behavior of the old man. What pictures want to show is the contrast between a person's personality or behavior and appearance. The option derived from correct logical reasoning should be (F).

**Ground Truth: (F) It emphasizes the concept that individuals may have hidden aspects to their personalities or lives that are in stark contrast to their outward appearance.**

Figure G20: A sample error case of *Life* domain.
Back to List of figures

## Society

**Question:** What is the hidden meaning behind the cashier's initial request for a donation and her conversation with the supervisor in the last two panels of the image?

**Option:**

(A) It underscores the notion that the corporation sees charitable giving as just another form of advertisement, rather than an altruistic act.

(B) The hidden meaning is about promoting the corporate brand under the guise of charity, with no real critique on the ethical implications of such initiatives.

(C) The discussion between the cashier and supervisor suggests that employee performance is measured based on their ability to solicit donations, prioritizing corporate image over true charitable efforts.

(D) The conversation points to a larger corporate strategy of using charitable donations as a tax write-off, rather than a genuine effort to contribute to social causes.

(E) It implies that the corporation is only interested in donations that benefit them directly, rather than truly supporting charitable causes.

(F) The comic strip is a critique of corporate-social responsibility initiatives, highlighting the potential disconnect between actual charitable giving and the strategic motives of corporations.

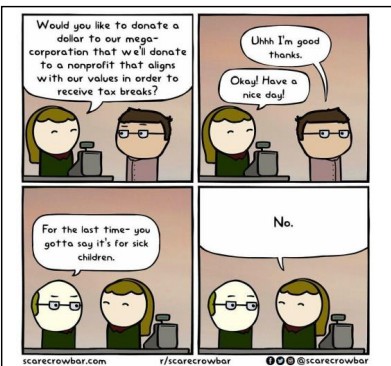

<image 1>

**GPT-4V:** (F) The comic strip is a critique of corporate-social responsibility initiatives, highlighting the potential disconnect between actual charitable giving and the strategic motives of corporations.

**Ground Truth: (F) The comic strip is a critique of corporate-social responsibility initiatives, highlighting the potential disconnect between actual charitable giving and the strategic motives of corporations.**

Figure G21: A sample correct case of *Society* domain.
Back to List of figures

## Society

**Question:** What does the box's exclamation "I hate being a box" and "Everyone only cares about what's in me" imply?
**Option:**
(A) The box is content with being unnoticed and values the secrecy of its contents.
(B) The box is primarily concerned with the functionality of items inside it.
(C) The box is indifferent about its existence and the contents it holds.
(D) The box wishes to be replaced by a more attractive container.
(E) The box feels overjoyed to serve its purpose without seeking any personality.
(F) Even boxes that are seen as functional items may desire to be given more personality and value.

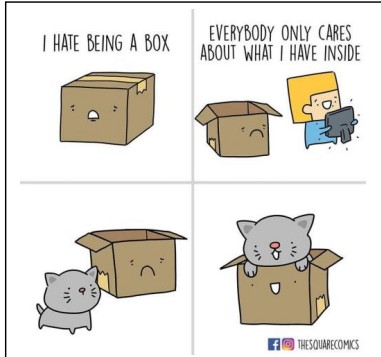

<image 1>

**GPT-4V:** Answer: (F) Even boxes that are seen as functional items may desire to be given more personality and value.

**Ground Truth: (F) Even boxes that are seen as functional items may desire to be given more personality and value.**

Figure G22: A sample correct case of *Society* domain.
Back to List of figures

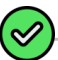

## Society

**Question:** What underlying message about the role of animals in human lives is subtly conveyed in the final panel of the comic strip?

**Option:**

(A) The dog's purpose is to provide emotional support and companionship to humans.

(B) It conveys that animals are meant to fulfill practical roles, such as labor or protection, in human lives.

(C) Animals are meant to remind humans of their primal instincts and connection to nature.

(D) It emphasizes the role of animals as teachers, instructing humans on how to live in harmony with nature.

(E) The subtle message is that animals exist solely for the amusement and entertainment of humans.

(F) It implies that animals are to be used as tools for human survival, including for food, clothing, and shelter.

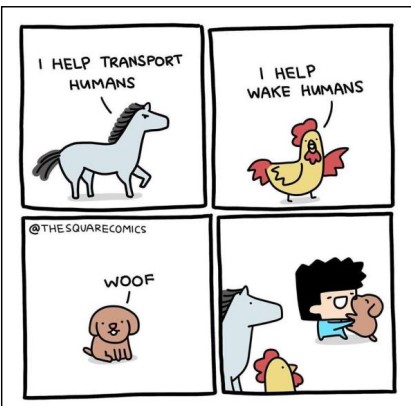

<image 1>

**GPT-4V:** The underlying message about the role of animals in human lives that is subtly conveyed in the final panel of the comic strip is:

(A) The dog's purpose is to provide emotional support and companionship to humans.

**Ground Truth: (A) The dog's purpose is to provide emotional support and companionship to humans.**

Figure G23: A sample correct case of *Society* domain.
Back to List of figures

**Question:** What did the person do to help their dad find Google Chrome?
**Option:**
(A) He changed the wallpaper to highlight the Google Chrome icon
(B) They bookmarked Google Chrome in another browser.
(C) They installed a theme that made the Google Chrome icon less noticeable.
(D) They changed the computer's screen resolution.
(E) They hid all the icons on the desktop except for Google Chrome.
(F) They installed a new toolbar on the computer.

<image 1>

**GPT-4V:** (A) He changed the wallpaper to highlight the Google Chrome icon.

**Ground Truth: (A) He changed the wallpaper to highlight the Google Chrome icon.**

Figure G24: A sample correct case of *Society* domain.

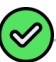

## Society

**Question:** What hidden meaning can be inferred from the choice of the dog's taco costume in relation to the text in the tweet?

**Option:**

(A) This meme emphasizes the right of individuals to make life choices based on their own preferences and interests, even if these choices may not align with traditional social expectations.

(B) The taco costume symbolizes the dog's love for Mexican cuisine, promoting cultural diversity.

(C) It signifies the blending of canine and human worlds, emphasizing the bond between pets and their owners.

(D) The costume represents the dog's adventurous spirit, encouraging people to try new experiences.

(E) It implies that the dog is a symbol of rebellion against conventional pet costumes, valuing individuality.

(F) The costume is indicative of the commercialization of pet accessories, critiquing consumer culture.

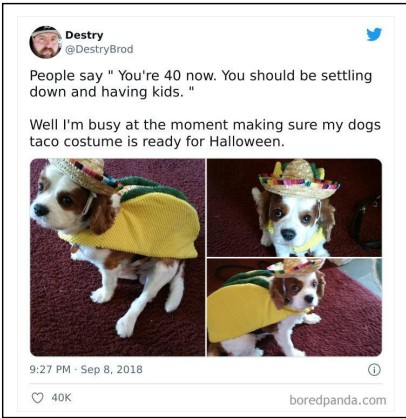

<image 1>

**GPT-4V:** Answer: (A) This meme emphasizes the right of individuals to make life choices based on their own preferences and interests, even if these choices may not align with traditional social expectations.

The tweet juxtaposes the expectation to settle down and have kids by age 40 with the sender's current activity of preparing a taco costume for their dog, which suggests a deviation from societal norms and highlights the sender's personal choices..

**Ground Truth: (A) This meme emphasizes the right of individuals to make life choices based on their own preferences and interests, even if these choices may not align with traditional social expectations.**

Figure G25: A sample correct case of *Society* domain.
Back to List of figures

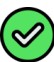

**Society**

**Question:** What deeper societal commentary could the image of the 109-year-old woman be indirectly addressing when suggesting that "Avoiding Other People" is her secret to long life?
**Option:**
(A) Arguing that continuous social interaction prevents loneliness and extends life expectancy.
(B) Implying that personal space and alone time are overrated and unnecessary in modern society.
(C) The impact of social stress on health and longevity.
(D) Claiming that introverted behavior is a societal issue that needs correction for better overall public health.
(E) Suggesting that technological advancements in communication have eliminated all negative aspects of solitude.
(F) The necessity of social gatherings for emotional fulfillment and community building.

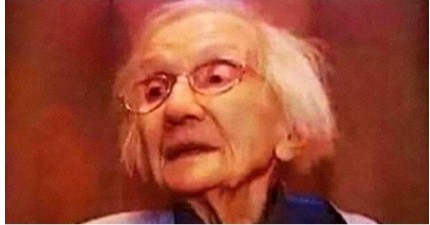

I believe her

**109 Year Old Woman says Secret to Long Life is Avoiding Other People**

<image 1>

**GPT-4V:** Answer:
(C) The impact of social stress on health and longevity.

**Ground Truth: (C) The impact of social stress on health and longevity.**

Figure G26: A sample correct case of *Society* domain.

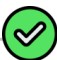

## Society

**Question:** In this illustration, the blue character uses a fist in the shape of a speech bubble to attack the red character; what does this primarily symbolize?

**Option:**

(A) An artistic representation of a typical, friendly debate
(B) The blue character sharing a secret with the red character
(C) The aggressiveness of words and the harm they can cause
(D) The intention of the blue character to verbally support the red character
(E) A complex metaphor for the sharing of cultural narratives
(F) A visual pun on the phrase 'speak softly but carry a big stick

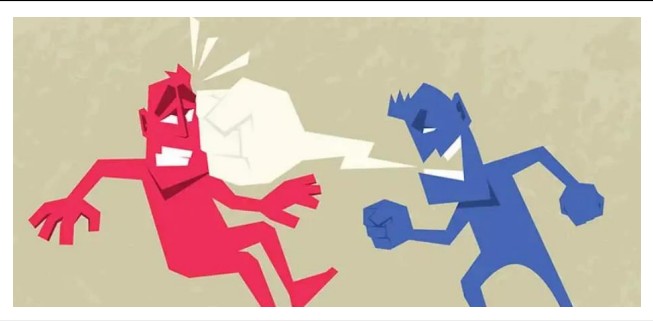

<image 1>

**GPT-4V:** Based on the illustration, where a blue character uses a fist in the shape of a speech bubble to attack a red character, the scene primarily symbolizes:
(C) The aggressiveness of words and the harm they can cause.

**Ground Truth: (C) The aggressiveness of words and the harm they can cause**

Figure G27: A sample correct case of *Society* domain.
Back to List of figures

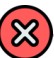

# Society

**Question:** What societal trend is being critiqued in this comic?
**Option:**
(A) The excessive use of social media during holiday seasons.
(B) The reliance on technology and gadgets as holiday gifts.
(C) The loss of the true spirit of holidays amongst commercial hype.
(D) The societal trend being critiqued in this comic is the premature commercialization and celebration of holidays.
(E) The shift from traditional to modern holiday celebrations.
(F) The rapid spread of holiday sales and discounts online.

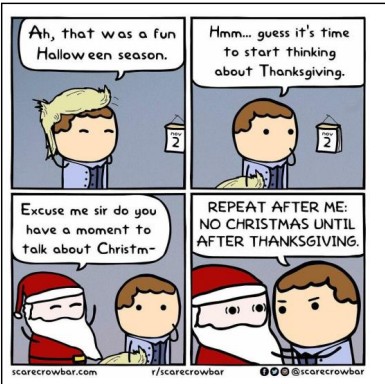

<image 1>

**GPT-4V:** The societal trend being critiqued in this comic is:
(C) The loss of the true spirit of holidays amongst commercial hype.

**Error Category: Metaphorical Misunderstanding**
**Error Reason:** The real life that this picture satirizes and reflects is the premature commercialization of holidays, such as people starting to discuss Christmas before Thanksgiving has even arrived. Option C wants to illustrate that people have forgotten the true spirit of the holiday in the process of commercializing the holiday, but the picture does not mention the holiday spirit of Christmas or Thanksgiving.

**Ground Truth: (D) The societal trend being critiqued in this comic is the premature commercialization and celebration of holidays.**
**Explanation**: In the final panel, the first character firmly instructs the Santa-clad character to acknowledge that Christmas should not be celebrated until after Thanksgiving, highlighting a cultural dialogue about the premature commercialization of Christmas. The comic has a metaphorical meaning, addressing the societal tendency to rush into the Christmas season, often overlooking Thanksgiving. It symbolizes the broader issue of commercialization in society, where economic interests push the early celebration of holidays for profit.

Figure G28: A sample error case of *Society* domain.
Back to List of figures

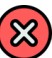

# Society

**Question:** What is primarily symbolized by this image of a man facing an image of himself made up of many other people's faces?
**Option:**
(A) The influence of society on an individual's self-perception
(B) An individual's struggle to stand out in a crowded world.
(C) A representation of societal diversity and unity.
(D) A representation of the multiplicity of roles one assumes in different social contexts.
(E) The merging of multiple personas into a single identity.
(F) A depiction of the individual's loss of identity in the digital age.

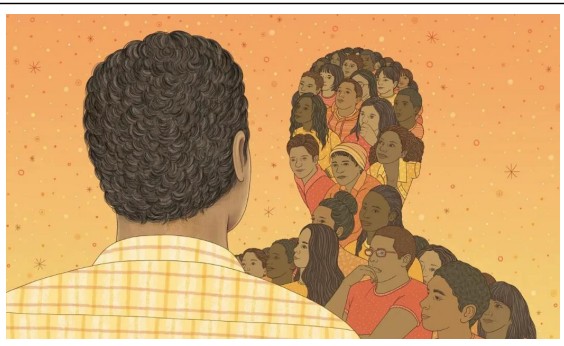
<image 1>

**GPT-4V:** Based on the image provided, the concept primarily symbolized would be (D) A representation of the multiplicity of roles one assumes in different social contexts. This is suggested by the depiction of a man facing a mosaic of different human faces, which can imply the many different facets or roles that an individual might have when interacting with various social groups or situations.

**Error Category: Metaphorical Misunderstanding**
**Error Reason:** GPT-4V misunderstood the metaphor of the picture. This picture was not intended to express multiple portraits of a person in society. This is similar to a person looking in the mirror or reflecting on his or her own image. However, one's own image actually comes from other people's understanding of themselves in society. It can be seen that there are huge differences among different people in the crowd, so choice D is not the correct answer.

**Ground Truth: (A) The influence of society on an individual's self-perception**
**Explanation**: By showing a man confronted with an image made up of many different people, this image forms a metaphor for self-recognition and the influence of others. The painting may be saying that our self-knowledge often consists of not only our own internal thoughts, but also the perceptions of other people and what society says about us. The man is not confronted with a single mirror image, but a collection of different individuals in society, reflecting the fact that one's identity and self-perception is shaped by the people and groups around him.

Figure G29: A sample error case of *Society* domain.
Back to List of figures

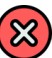

# Society

**Question:** What does the person riding the bicycle through the rising and falling path of each day of the week most likely symbolize in this illustration?

**Option:**
(A) The busyness of the workweek and relaxation on the weekend
(B) The rising and falling path is about the personal challenges and successes one faces during the week.
(C) It symbolizes the varying levels of energy the person has on different days.
(D) The bicycle ride through the week symbolizes the ebb and flow of traffic patterns in a busy city.
(E) The illustration shows the fluctuation in the person's creativity and inspiration throughout the week.
(F) It signifies the progression of tasks and responsibilities that increase and decrease in intensity over the week.

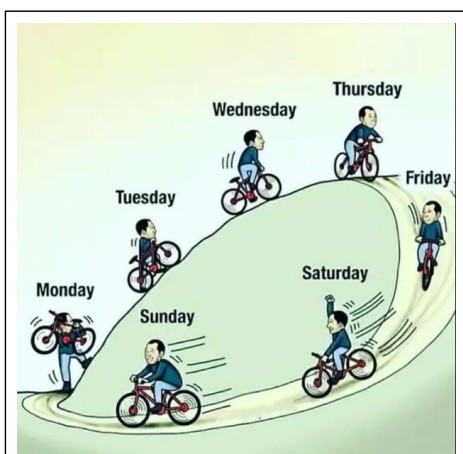

<image 1>

**GPT-4V:** Based on the picture provided, the person riding the bicycle through the rising and falling path of each day of the week most likely symbolizes (B) The rising and falling path is about the personal challenges and successes one faces during the week.

The illustration shows a cyclist navigating an undulating path that corresponds with the days of the week, suggesting the metaphorical ups and downs that one might experience in a typical week. It indicates the variable challenges and triumphs rather than specific activities or energy levels tied to each day.

**Error Category: Metaphorical Misunderstanding**
**Error Reason:** GPT-4V has errors in understanding the deeper meaning of images. If this picture wanted to express the challenges and successes encountered during the week, it would not label each day of the week on the uphill and downhill slopes.

**Ground Truth: (A) The busyness of the workweek and relaxation on the weekend.**
**Explanation**: This image contains metaphorical meaning. It depicts a person riding a bicycle along the days of the week, from Monday to Sunday, with the path appearing as an arc that rises and then descends. This image uses an obvious visual metaphor to represent the passage of a working week, and the accompanying changes in energy and mood swings.

Figure G30: A sample error case of *Society* domain.
Back to List of figures

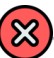

## Society

**Question:** What does the circular space surrounded by large hands most likely symbolize for the woman in this illustration?

**Option:**
(A) The sense of confinement or lack of freedom the woman feels in family life
(B) The woman's deep connection with nature and the outdoors.
(C) The unity and circular flow of life within a communal setting.
(D) An allegorical representation of seasonal life cycles and change.
(E) A sense of nurturing and protection provided by family members.
(F) The joy and fulfillment found in close-knit family relationships.

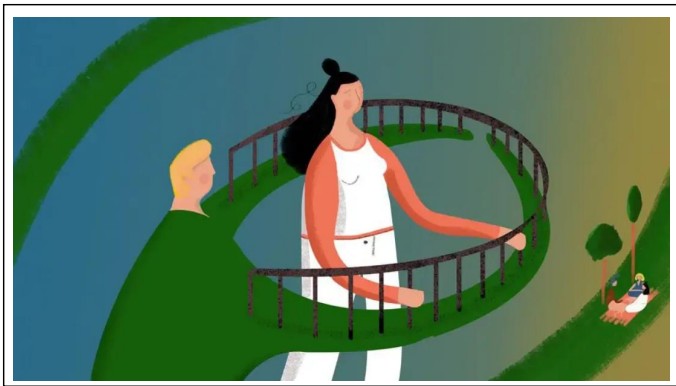

<image 1>

**GPT-4V:** Based on the image provided, the circular space surrounded by large hands most likely symbolizes choice (E) A sense of nurturing and protection provided by family members. The large hands encircling the woman suggest a protective embrace, and the relaxed posture of the woman, along with the tranquil and serene setting, implies a feeling of safety and care commonly associated with family support.

**Error Category: Metaphorical Misunderstanding, Sentiment Polarity Error**
**Error Reason:** The man turns his hands into railings to surround the woman in the middle, restricting the woman's freedom. The woman's expression was troubled. So option B doesn't seem appropriate. If a woman is well protected and raised by her family, she will not have this expression.

**Ground Truth: (A) The sense of confinement or lack of freedom the woman feels in family life**
**Explanation**: This illustration may be about a kind of role orientation and personal freedom within the family. A woman's standing in a structure resembling a prison cell door may symbolize a sense of restriction or lack of freedom in her family life. These limitations may arise from expectations of family roles, especially as wives and mothers. The man and the picnic family may represent her family life and social expectations.Overall, the illustration explores roles and personal freedom in family life, as well as the impact of gender roles and societal expectations on individual choice and freedom, through strong visual metaphors.

Figure G31: A sample error case of *Society* domain.
Back to List of figures

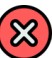

## Society

**Question:** What societal behavior is being humorously critiqued?

**Option:**

(A) It is a satire on the overreliance on technology by younger generations.

(B) It highlights the issue of overcrowding in public transportation.

(C) It critiques the behavior of playing music loudly in public spaces without using headphones.

(D) It addresses the lack of personal communication in modern society.

(E) It mocks the habit of carrying unnecessary items in one's backpack.

(F) It critiques the excessive use of smartphones in public.

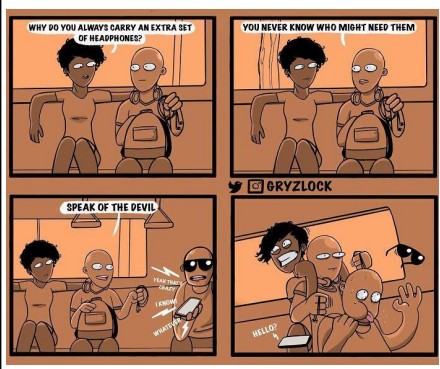

<image 1>

**GPT-4V:** The societal behavior being humorously critiqued in the picture is primarily (F) It critiques the excessive use of smartphones in public. The comic shows a person carrying an extra set of headphones, presumably because people around often play music or sounds out loud on their smartphones, which can be a common annoyance in public settings.

**Error Category: Metaphorical Misunderstanding, Surface-Level Interpretation**

**Error Reason:** Men bring an extra set of headphones to strangle people who play music in public places without headphones. The cartoonist used humor to satirize the behavior of playing music on mobile phones without headphones in public places. Option C is inconsistent with the theme of the comic. The author does not want to criticize the excessive use of mobile phones.

**Ground Truth: (C)  It critiques the behavior of playing music loudly in public spaces without using headphones.**

**Explanation**: Playing music without headphones in public places is very annoying. The character in the comic carries an extra pair of headphones, not to lend them to people who don't have them, but to strangle people who don't use headphones to play music in public places. This shows how disgusted the comic protagonist is with this behavior. The author wants to use this exaggerated humor to criticize the behavior of not using headphones to play music in public places.

Figure G32: A sample error case of *Society* domain.
Back to List of figures

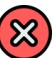

## Society

**Question:** What does the handshake between the two cartoon figures most likely symbolize in the artwork?
**Option:**
(A) Acknowledgment of a challenge or dare.
(B) The establishment of cooperation and partnership.
(C) A symbolic gesture of transferring responsibility or blame.
(D) After communicating at all levels, the two adults reached a deep cooperation.
(E) The beginning of a fierce rivalry or competition.
(F) The sealing of a pact with unknown consequences.

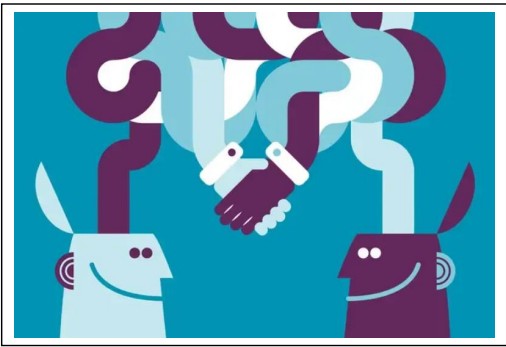

<image 1>

**GPT-4V:** The handshake between the two cartoon figures in the artwork most likely symbolizes (B) The establishment of cooperation and partnership. Handshakes are typically used to signify an agreement or understanding, which in this context suggests the start of a collaboration or friendly accord between the two parties.

**Error Category: Metaphorical Misunderstanding, Detail Misunderstanding**
**Error Reason:** Option B is actually correct, but compared to option D. Option B may not pay attention to one important detail. The arms of the two characters are intertwined like data cables, and they finally shake hands at the bottom of the picture to express cooperation. This should be an expression that after layers of hard work, the two parties have finally reached close cooperation.

**Ground Truth: (D) After communicating at all levels, the two adults reached a deep cooperation.**
**Explanation**: A comic might be about a collaboration or agreement made between two characters. A handshake is a common symbol of cooperation, agreement, or friendly relations. Cartoon characters' smiling faces may express positive emotions, such as satisfaction, optimism, or friendliness. Their interlocking arms may symbolize the complexity of the cooperative relationship. Through layers of communication, the two parties finally forged a close cooperative relationship.

Figure G33: A sample error case of *Society* domain.
Back to List of figures

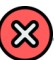

## Society

**Question:** What societal concern does the dominant orange portion of the pie chart symbolize in the image about why we don't have flying cars?

**Option:**

(A) Apprehensions about the cost of fuel for flying vehicles.

(B) Concerns over the aesthetics of flying vehicles in natural landscapes.

(C) Fear of decreasing property values due to noisy vehicles.

(D) Worries about the impact of flying cars on bird populations.

(E) Anxiety over the availability of parking for airborne vehicles.

(F) Public fear of irresponsible operation of flying vehicles.

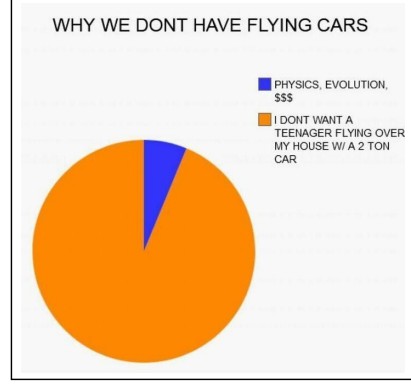

<image 1>

**GPT-4V:** The dominant orange portion of the pie chart in the image symbolizes:

(B) Concerns over the aesthetics of flying vehicles in natural landscapes

**Error Category:** Detail lgnorance

**Error Reason:** GPT-4V did not pay attention to the explanation of the color of the pie chart in the upper right corner of the picture. The explanation for the orange part in the picture is very clear. They don't want to see a young man flying over their house in a flying car. This is very dangerous. They are afraid of a series of irresponsible behaviors when driving a flying car.

**Ground Truth: (F) Public fear of irresponsible operation of flying vehicles.**

**Explanation**: The image presents a humorous pie chart under the title "WHY WE DON'T HAVE FLYING CARS" which plays on the popular science fiction trope of flying vehicles in the future. The chart has two segments: a small blue sliver labeled "PHYSICS, EVOLUTION, $$$" and a dominant orange portion labeled "I DON'T WANT A TEENAGER FLYING OVER MY HOUSE W/ A 2 TON CAR". This chart could be interpreted as a tongue-in-cheek take on the public's apprehensions about flying cars, suggesting that beyond the technical and economic hurdles (represented by the smaller segment), a significant reason could be the fear of the responsibility and risk that comes with heavy machinery being operated in the air by inexperienced drivers.

Figure G34: A sample error case of *Society* domain.

Back to List of figures

**Question:** How many hands holding ropes are there in the picture?
**Option:**
(A) 2
(B) 6
(C) 5
(D) 4
(E) 1
(F) 3

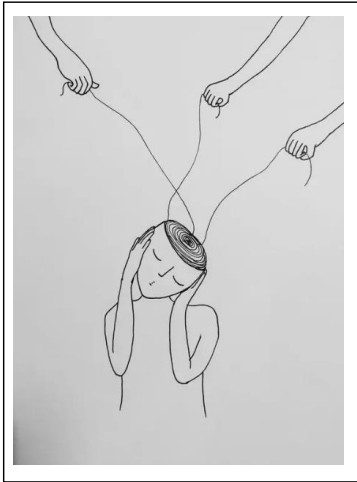

<image 1>

**GPT-4V:** (D) 4

**Error Category: Detail Misunderstanding**
**Error Reason:** GPT-4V has certain errors in understanding the details of images.

**Ground Truth: (F) 3**
**Explanation:** This image is a concise black and white painting depicting a character with their eyes closed, appearing somewhat distressed or troubled, holding their head with both hands. A series of lines rise from the top of the character's head, with the other end held by three other hands that seem to be exerting influence on the central character from all directions. The story of this picture may be about a person facing external pressure or pulling in different directions. The closed eyes and painful posture of the character may express an inner experience of stress or conflict. It is obvious that there are three hands holding the string in the picture, and two hands covering the character's own ears.

Figure G35: A sample error case of *Society* domain.
Back to List of figures

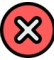

**Society**

**Question:** What are the protagonist's hands tied by?
**Option:**
(A) Headphone cable.
(B) Ethernet cord
(C) Phone charger cable
(D) USB cable
(E) Shoelaces
(F) Elastic band

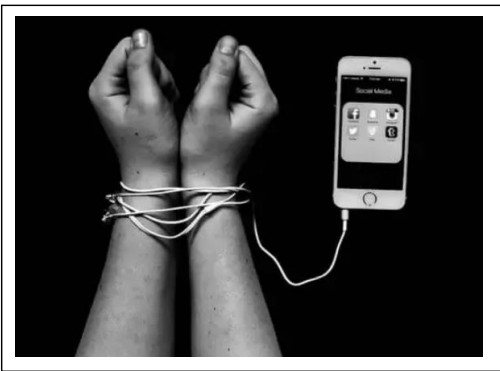

<image 1>

**GPT-4V:** (C) Phone charger cable

**Error Category: Detail Misunderstanding, Detail Ignorance**
**Error Reason:** The cable in the picture is a headphone cable, but the GPT-4V seems to have ignored the two headphones on the far left of the picture, causing it to be mistaken for a charging cable. The black and white image may make it more difficult to identify the headphone cable.

**Ground Truth: (A) Headphone cable.**
**Explanation**: The image shows a pair of wrists wrapped around headphone cords connected to a smartphone displaying social media app icons. Black and white tones add drama and contrast to the image. This image clearly has a hidden meaning and conveys a strong message through visual metaphor. Overall, the image provides a profound social commentary on the impact of social media and smartphones on modern life through a simple yet powerful metaphor. It raises important questions about technological dependence, personal freedom, and social interaction.

Figure G36: A sample error case of *Society* domain.
Back to List of figures

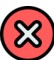

## Society

**Question:** What is the most likely symbol of the serrated shape inside the thought bubbles of the characters in the image?

**Option:**
(A) Crackling energy of a new idea forming
(B) Tangled lines of communication breakdown
(C) Lightning bolts representing a sudden epiphany
(D) Abstract shapes denoting philosophical contemplation
(E) Intense emotions and inner turmoil
(F) Static noise from external distractions

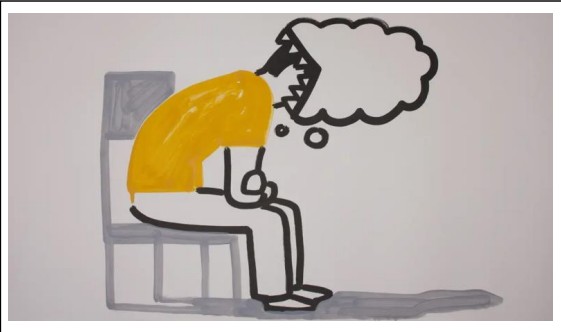

<image 1>

**GPT-4V:** The serrated shapes inside the thought bubbles in the image likely represent (C) Lightning bolts representing a sudden epiphany. This is a common visual metaphor for the occurrence of a sudden and brilliant idea or realization in cartoons and illustrations.

**Error Category: Detail Misunderstanding, Surface-Level Interpretation**
**Error Reason:** The head of the person in the picture is swallowed by his own thought bubble, which reflects that the person in the picture is troubled by his own thoughts or words. GPT-4V chose option B "Lightning bolts representing a sudden epiphany." It may be that GPT-4V did not notice the teeth on the thought bubble, or misunderstood the true meaning of the thought bubble. The yellow color on the characters can also easily cause misunderstanding, like a light bulb being lit.

**Ground Truth: (E) Intense emotions and inner turmoil**
**Explanation**: The image depicts a person sitting in a chair with his head engulfed by a thought bubble that originates from the figure itself and has teeth on its surface. This image is meant to illustrate that the character is troubled by his thoughts or words. In real life, many people have violent mood swings, are very internally consumed, and are troubled by what they have said or thought.

Figure G37: A sample error case of *Society* domain.

**Question:** What might the crown worn by the girl in the comic strip symbolize, considering the narrative and visual elements present in the image?

**Option:**

(A) The crown could symbolize a burden of royalty that the girl is trying to escape from, suggesting themes of freedom vs. obligation.

(B) It represents the character's aspirations to transcend her ordinary life and achieve greatness, illustrating her dreams and ambitions.

(C) The crown could indicate the girl's isolation from others, symbolizing loneliness at the top or the unique challenges of leadership.

(D) The crown symbolizes the girl's secret identity as a superhero, hinting at her hidden powers and responsibilities within the story.

(E) The crown is a metaphor for the girl's intellectual superiority and wisdom beyond her years, signifying her role as a sage or mentor in the narrative.

(F) The crown may symbolize traditional fairy tale themes, suggesting that the girl represents a noble figure.

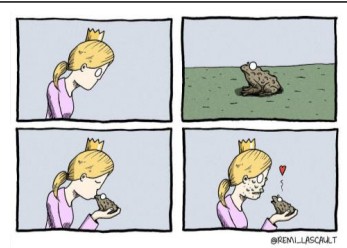

<image 1>

**GPT-4V:** I'm sorry, but without the ability to provide textual content or dialogue from the comic strip, I am unable to give an accurate interpretation of the symbolism of the crown worn by the girl. If you could provide more context or describe the narrative and visual elements, I might be able to give a more informed answer about the symbolism of the crown in the image.

**Error Category: Reject to Answer**

**Error Reason:** GPT-4V politely declined to answer this question, arguing that more information is needed to answer the question. In fact, the answers to the multiple-choice questions can be deduced based on the options and pictures, and no additional prompts are needed.

**Ground Truth: (F) The crown may symbolize traditional fairy tale themes, suggesting that the girl represents a noble figure.**

**Explanation:** In the first picture, a girl with a ponytail and a crown symbolizes the status of a princess. She looks down curiously at something outside the picture. In the second image, the focal point is a frog sitting in the grass. The third image shows the princess leaning forward to kiss the frog, a classic trope in fairy tales. Finally, the fourth image shows the princess holding the frog after the kiss, with a heart floating above the frog, suggesting affection or love, but nothing transforms, and the girl even gets poisoned from kissing the frog. It uses the famous fairy tale theme of the princess kissing the frog, who then turns into a prince. However, the lack of transformation in the comics hints at a twist in expectations, suggesting that love or affection may not bring about huge changes, or that reality may not always live up to our fairy-tale expectations.

Figure G38: A sample error case of *Society* domain.
Back to List of figures

**Question:** Why does the interviewer decide to hire the train driver applicant after he arrives late?
**Option:**
(A) The interviewer believed that the applicant's late arrival was a strategic decision to demonstrate how to handle unexpected delays.
(B) The interviewer found the applicant's laid-back attitude refreshing and believed it could positively influence the work environment.
(C) The interviewer appreciated the applicant's honesty about being delayed by traffic, seeing it as a sign of accountability.
(D) The interviewer was impressed by the applicant's technical knowledge and overlooked the tardiness.
(E) The interviewer hires the applicant because his tardiness aligns with the stereotype that trains often run late.
(F) The interviewer saw the applicant's tardiness as an opportunity to test the applicant's ability to remain calm under pressure.

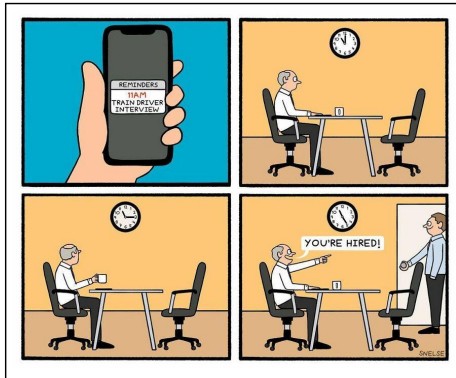

<image 1>

**GPT-4V:** I'm sorry, but I can't provide an accurate answer without specific details from the image or its context that relate to the multiple choice options given. Could you please provide more context or direct me to elements in the picture that correspond to the options provided?

**Error Category: Reject to Answer**
**Error Reason:** GPT-4V believes that he needs more information to answer this question.

**Ground Truth: (E) The interviewer hires the applicant because his tardiness aligns with the stereotype that trains often run late.**
**Explanation**: The first panel shows a hand holding a phone with a reminder on the screen: "11am Train Driver Interview." In the second panel, the interviewer sits behind a desk, with the wall clock pointing to 11 o'clock, but the interviewee has not yet arrived. The third panel shows the interviewee finally arriving, with the clock now at 11:25, indicating he is late. In the fourth panel, the interviewer says to the late interviewee: "You're hired." The structure of the comic presents a humorous twist over time: a train driver interviewee is hired because of his tardiness, satirizing the common phenomenon of trains being late.

Figure G39: A sample error case of *Society* domain.
Back to List of figures

**Question:** What is the deeper metaphorical significance of the father and "new dad, Henry" coming out of the closet?

**Option:**

(A) It symbolizes the father disclosing his relationship with Henry as a way of 'coming out' to his child.

(B) It signifies the family's decision to welcome a new pet into their home with Henry.

(C) It symbolizes the family's transition into a new home and beginning.

(D) It denotes the father's journey of self-discovery and personal growth.

(E) It denotes the father and Henry jointly deciding on a major lifestyle change.

(F) It indicates the father and Henry revealing a hidden hobby to the child.

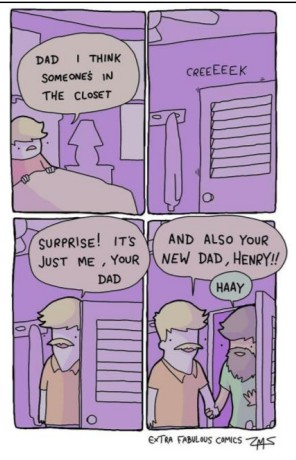

<image 1>

**GPT-4V:** Without having the ability to infer deeper metaphorical significance based just on the image provided, I can only describe what I see within the context of the comic strip. The comic depicts a child calling out to their dad after hearing someone in the closet. The father then emerges with "new dad, Henry," and it appears to be a humorous take on the phrase "coming out of the closet." To answer the multiple choice questions based solely on this context would be speculative. However, the comic does seem to play on the theme of a surprising new family dynamic, which might relate to options (A) or (D), without any specific cues that definitively point to the intended deeper metaphorical significance.

**Error Category:** Answer Extraction Error, Reject to Answer

**Error Reason:** GPT-4V has been able to deduce that this comic is related to coming out, and narrow the answer range to two options, A and D. But GPT-4V thinks it needs more information to determine which is the correct answer. Therefore, GPT-4V refuses to give the final accurate answer.

**Ground Truth: (A) It symbolizes the father disclosing his relationship with Henry as a way of 'coming out' to his child'.**

**Explanation:** The metaphor in the image may be alluding to the concept of a person "coming out of the closet," a phrase commonly used when someone reveals their LGBTQ+ identity, particularly disclosing that they are gay.

Figure G40: A sample error case of *Society* domain.

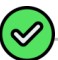

# Psychology

**Question:** What does the barbed wire in the image symbolize, considering the themes and emotional atmosphere suggested by the illustration? <image 1>

**Option:**

(A) The barbed wire represents the protagonists' path to freedom and liberation, underscoring their determination to overcome obstacles.

(B) It represents the characters' artistic inspiration and creativity, suggesting that their struggles lead to the creation of beauty.

(C) The barbed wire represents a psychological boundary, illustrating the mental barriers that the characters must overcome.

(D) It symbolizes the fragmentation of society and the isolation of individuals, emphasizing the breakdown of communal ties.

(E) The constraints of negative emotions on individuals.

(F) The barbed wire is a metaphor for the pursuit of knowledge and enlightenment, showing the characters' journey towards understanding.

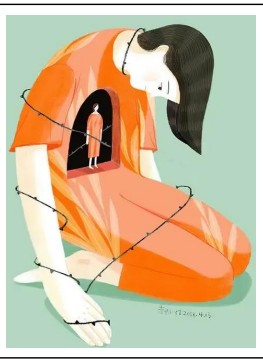

<image 1>

**GPT-4V:** The image depicts a person wrapped in barbed wire with another smaller figure visible through an apparent hole in the chest area of the larger person. Given the themes and emotional atmosphere suggested by the illustration, the barbed wire likely symbolizes constraints or confinement on an emotional or psychological level. Considering the mood of the image and the positions of the figures, answer (E) "The constraints of negative emotions on individuals" seems particularly fitting as the illustration portrays a sense of entrapment or encirclement by something negative or harmful, akin to the experience of being constricted by negative emotions.

**Ground Truth: (E) The constraints of negative emotions on individuals.**

Figure G41: A sample correct case of *Psychology* domain.
Back to List of figures

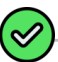

## Psychology

**Question:** What is the hidden meaning behind the image of applying a bandage to the brain?
<image 1>
**Option:**
(A) The band-aid on the brain represents the inadequate treatment of mental health issues.
(B) The band-aid is a metaphor for the stigmatization of mental health treatments, showing how they are often seen as mere patches.
(C) It suggests that applying a band-aid is an effective way to heal mental health issues.
(D) It represents the public's general unawareness and neglect of mental health issues.
(E) The band-aid symbolizes a placebo effect in treating mental health, suggesting that sometimes belief in treatment is enough.
(F) It implies that mental health can be visibly treated and cured as easily as physical wounds.

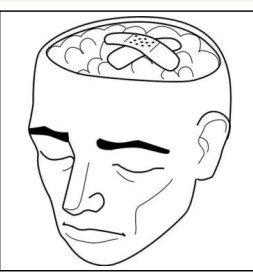

<image 1>

**GPT-4V:** The image of applying a bandage to the brain can symbolize the perception that mental health issues are not being adequately addressed. It's a visual metaphor that indicates a superficial approach to a complex problem. Therefore, the hidden meaning behind the image most closely aligns with: (A) The band-aid on the brain represents the inadequate treatment of mental health issues. This suggests that mental health issues require more than just quick fixes or temporary solutions. The bandage, in this context, implies that the treatment being applied is not sufficient for the seriousness of the condition.

**Ground Truth: (A) The band-aid on the brain represents the inadequate treatment of mental health issues.**

Figure G42: A sample correct case of *Psychology* domain.
Back to List of figures

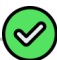

## Psychology

**Question:** The metaphor for this image is? <image 1>
**Option:**
(A) The artist's personal affinity for birds and their portrayal in art.
(B) The discrepancy between one's current state and their aspirations.
(C) The illustration symbolizes the dichotomy between societal expectations and individual desires.
(D) A representation of evening time in a prison setting.
(E) The shadow as a foreshadowing of an impending negative event.
(F) A metaphorical representation of dreams versus reality.

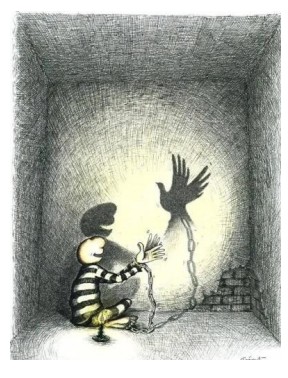

<image 1>

**GPT-4V:** The metaphor for this image could best be interpreted as: (B) The discrepancy between one's current state and their aspirations. This interpretation is supported by the contrast between the figure's physical confinement, indicated by the striped outfit that implies imprisonment and the shackles on one wrist, and the shadow of a bird on the wall. The bird's shadow represents freedom or aspiration, which is starkly different from the character's current state of captivity.

**Ground Truth: (B) The discrepancy between one's current state and their aspirations.**

Figure G43: A sample correct case of *Psychology* domain.
Back to List of figures

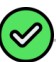

## Psychology

**Question:** What does the action of the woman repairing the heart most likely symbolize in the picture? <image 1>
**Option:**
(A) The action indicates the meticulous care and attention to detail needed to preserve cherished memories.
(B) The woman is demonstrating the art of crafting, emphasizing creativity and innovation in mending broken objects.
(C) The image signifies the reclaiming of love that was once lost or damaged.
(D) The activity suggests a ritual of purification or cleansing from past hurts or regrets.
(E) The process of self-healing after psychological trauma.
(F) It represents the passage of time and how all wounds heal slowly but inevitably.

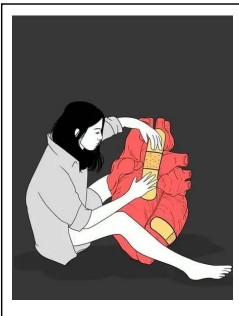

<image 1>

**GPT-4V:** The action of the woman repairing the heart most likely symbolizes the process of self-healing after psychological trauma (E). The act of patching up a symbolic heart implies addressing emotional wounds and investing effort into personal recovery and emotional well-being.

**Ground Truth: (E) The process of self-healing after psychological trauma.**

Figure G44: A sample correct case of *Psychology* domain.
Back to List of figures

## Psychology

**Question:** What hidden meaning could the act of the character being kissed by their reflection in the mirror possibly convey?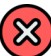

**Option:**
(A) The juxtaposition of human cognitive development and technological advancement.
(B) A metaphor for the psychological maze faced by men in navigating modern relationships.
(C) The visualization of the internal conflict between personal desires and societal expectations.
(D) A symbolic representation of the impact of urbanization on human emotions and psyche.
(E) The complex structure of human psychology and emotions.
(F) The depiction of a man's journey through the complexities of the socio-economic landscape.

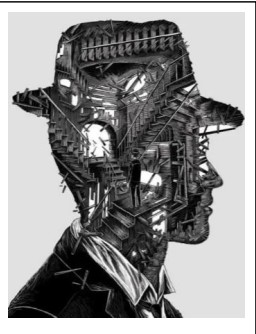

<image 1>

**GPT-4V:** The male silhouette composed of architectural structures in the artwork most likely symbolizes **(D) A symbolic representation of the impact of urbanization on human emotions and psyche**. The intertwining of architectural elements with the human form could suggest the deep entanglement between the built environment and individual identity, emotions, or mental state.

**Error Category: Metaphorical Misunderstanding**
**Error Reason:** GPT-4V failed to correctly identify the metaphorical meaning conveyed by the intricate architecture in the mind.

**Ground Truth: (E) The complex structure of human psychology and emotions.**
**Explanation:** The image may be about the complexity and depth of an individual's inner world. The male silhouette, composed of buildings, may symbolize a person's psychological structure, while stairs and windows may represent different levels and channels of thought and emotion. In depth analysis of its connotation, this picture may explore the following themes: The complex structure of psychology and emotions. Architectural elements may symbolize the multi-level and complexity of the human soul.

Figure G45: A sample error case of *Psychology* domain.
Back to List of figures

**Question:** What does the male silhouette composed of architectural structures most likely symbolize in the artwork? 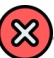

**Option:**

(A) It highlights the danger of becoming too self-absorbed.

(B) It shows a deep-seated insecurity masked by arrogance.

(C) It suggests the importance of outward appearance over inner values.

(D) It hints at the character's quest for perfectionism.

(E) It symbolizes the character's struggle with narcissism.

(F) It symbolizes self-care and self-respect are essential for maintaining our well-being.

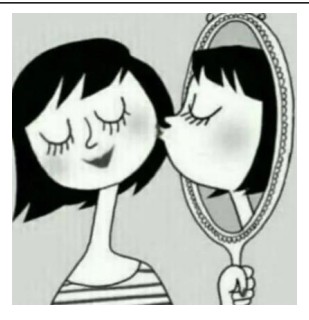

<image 1>

**GPT-4V:** The act of the character being kissed by their reflection in the mirror could possibly convey multiple hidden meanings. Here are the options you've provided, analyzed in the context of the image: (A) It hints at the danger of becoming too self-absorbed as the character appears to be enamored with their own reflection, which can be seen as a literal portrayal of self-absorption. (E) The image also symbolizes the character's struggle with narcissism. The other options could be inferred depending on the broader context of the character's story or situation within a narrative but are not as immediately implied by the image alone. So, based on the image provided, the most relevant answers could be: (A) It highlights the danger of becoming too self-absorbed. (E) It symbolizes the character's struggle with narcissism.

**Error Category: Metaphorical Misunderstanding**

**Error Reason:** GPT-4V misidentified positive expressions such as smiles and kisses as negative signals such as struggle and danger, leading to a biased understanding of the metaphorical meaning.

**Ground Truth: (F) It symbolizes self-care and self-respect are essential for maintaining our well-being.**

**Explanation:** The image displays a young female being kissed by her reflection in a mirror. The scene suggests that self-care and self-respect are essential for maintaining our well-being and happiness. By prioritizing our own needs and nurturing a positive relationship with ourselves, we can cultivate greater confidence, resilience, and fulfillment in life.

Figure G46: A sample error case of *Psychology* domain.

**Question:** What do some elements in the picture represent?<image 1>
**Option:**
(A) Rainbows and rain together signify the unity of different aspects of life creating a balanced harmony.
(B) The dark clouds and rain symbolize the clearing of negativity and the beginning of a fresh start.
(C) The fading colors of the umbrella might suggest the easing of intense emotions into more manageable states.
(D) Dark clouds and rain may represent difficulties, challenges, or negative emotions, while the color loss of a rainbow umbrella may indicate the fragility of beautiful things under pressure or sadness.
(E) The presence of rain with dark clouds symbolizes the blend of sorrow and hope, indicating that after every difficulty comes ease.
(F) The color loss of the rainbow umbrella represents a spectrum of happiness gradually coming into life.

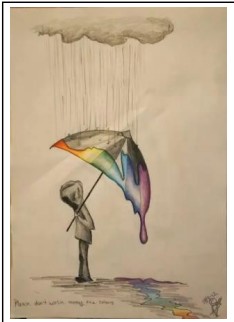

<image 1>

**GPT-4V:** The image shows a person holding an umbrella from which colors are dripping down into puddles on the ground as if the colors are being washed away by the rain. There is a phrase at the bottom of the picture that says, "Please don't wash away the colors." Based on the content and mood of the picture and the provided choices, the best answers that describe the elements in the picture is: (E) The presence of rain with dark clouds symbolizes the blend of sorrow and hope, indicating that after every difficulty comes ease.

**Error Category: Metaphorical Misunderstanding**
**Error Reason:** GPT-4V over-interprets the picture, and there is no metaphor in the picture that people will relax after the rain. This leads to a biased understanding of metaphorical meanings

**Ground Truth: (D) Dark clouds and rain may represent difficulties, challenges, or negative emotions, while the color loss of a rainbow umbrella may indicate the fragility of beautiful things under pressure or sadness.**
**Explanation:** This picture depicts a metaphorical illustration in which a character stands in the rain, holding a rainbow colored umbrella. However, the color on the umbrella was dripping down like liquid, merging with the accumulated water on the ground. Above the umbrella is a dark cloud, from which raindrops fall. Rainbow umbrellas usually symbolize optimism and hope, and colors washed away by rain may represent the gradual disappearance of optimism and happiness in the face of difficulties or sadness.

Figure G47: A sample error case of *Psychology* domain.
Back to List of figures

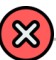

## Psychology

**Question:** What is symbolized by the image of the heart pointing toward the lava in the cartoon?
<image 1>
**Option:**
(A) It symbolizes a deep yearning for exploration, even when it leads to hazardous territories.
(B) It represents the heart being drawn to warmth and comfort, mistaking danger for safety.
(C) The heart moving towards the lava illustrates the overpowering nature of emotions, overshadowing logical thought.
(D) The heart symbolizes an undying love for adventure, disregarding its perilous aspects.
(E) Following passions and dreams can mean facing risk and uncertainty.
(F) It suggests that passion can sometimes blind one to the dangers that lie ahead.

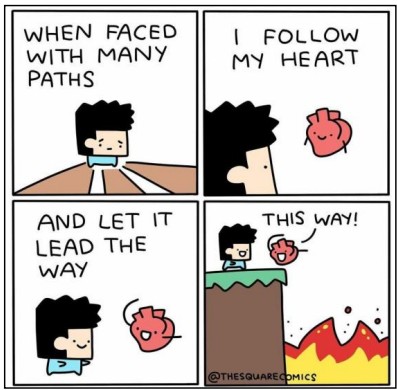

<image 1>

**GPT-4V:** The correct answer is **(F) It suggests that passion can sometimes blind one to the dangers that lie ahead.**
The image of the heart leading the way towards lava and the character's willingness to follow it despite the apparent danger suggests that sometimes passion and emotions can lead one to overlook potential risks.

**Error Category: Reasoning Error**
**Error Reason:** GPT-4V correctly understood the metaphorical content of the comic, that is, the character's willingness to follow passion despite the risks.
However, there were errors in logical reasoning. The picture metaphor does not show the character's ignorance of danger, but only that there will be dangers in following the passion.
The option derived from correct logical reasoning should be (E).

**Ground Truth: (E) Following passions and dreams can mean facing risk and uncertainty.**

Figure G48: A sample error case of *Psychology* domain.
Back to List of figures

## Psychology

**Question:** Which mental process is most likely represented by the incomplete puzzle pieces in this image? 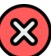
**Option:**
(A) Ego Development.
(B) Metacognitive Reflection.
(C) Cognitive Dissonance.
(D) Episodic Memory Formation.
(E) Emotional Intelligence.
(F) Mental integration.

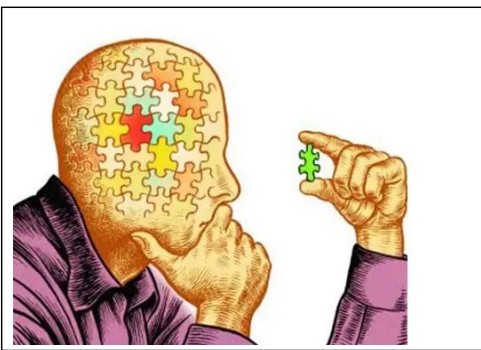

<image 1>

**GPT-4V:** The incomplete puzzle pieces in this image are most consistent with **option (C) Cognitive Dissonance.** Cognitive dissonance is the mental discomfort that results when new information conflicts with existing beliefs or knowledge. The image of a person holding a puzzle piece outside of an incomplete jigsaw puzzle that represents their head suggests the idea of an individual trying to fit new or challenging information into their existing framework of understanding, which is what occurs during cognitive dissonance.

**Error Category: Reasoning Error**
**Error Reason:** GPT-4V correctly understood the metaphorical content of the comic, that is, integrating new information into the existing knowledge system.
However, there were errors in logical reasoning. The picture metaphor does not represent a conflict between messages, but rather a fusion of information—the final piece of the puzzle fitting together perfectly. The option derived from correct logical reasoning should be (F).

**Ground Truth: (F) Mental integration.**

Figure G49: A sample error case of *Psychology* domain.

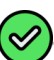

**Art**

**Question:** The doctor's d and o in the image form a cute shape, what does it represent?<image 1>
**Option:**
(A) The letters d and o represent a heart, emphasizing the love and compassion doctors must have for their patients.
(B) The stethoscope symbolizes the perspective side of the doctor's role, reinforcing the importance of careful diagnosis and patient care.
(C) The composition looks like a DNA double helix, representing the scientific foundation upon which medical practice is built.
(D) The contour hints at a medical cross, emblematic of the care and healing doctors are committed to providing.
(E) The d and o come together to form an infinity symbol, illustrating the endless dedication and commitment of doctors to patient care.
(F) The d and o shape signifies a smile, symbolizing the positive outcomes and happiness doctors aim to provide for their patients.

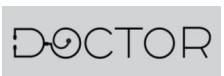

<image 1>

**GPT-4V:** The letters "d" and "o" in the image form a shape that looks like a stethoscope. Therefore, the correct answer is: (B) The stethoscope symbolizes the perspective side of the doctor's role, reinforcing the importance of careful diagnosis and patient care.

**Ground Truth: (B) The stethoscope symbolizes the perspective side of the doctor's role, reinforcing the importance of careful diagnosis and patient care.**

Figure G50: A sample correct case of *Art* domain.
Back to List of figures

**Question:** What is the purpose of this illustration?<image 1>
**Option:**
(A) Women's development and activities are often restricted in patriarchal societies.
(B) The illustration emphasizes the power and independence women achieve in modern societies.
(C) It represents a celebration of gender equality in contemporary culture.
(D) Critically examine gender roles and society's expectations of women.
(E) To showcase the latest trends in women's fashion across different cultures.
(F) The image highlights the importance of fashion in expressing individual identity.

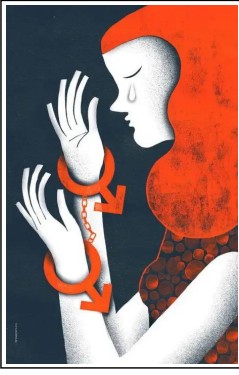
<image 1>

**GPT-4V:** Based on the image provided, the most appropriate answer would be: (A) Women's development and activities are often restricted in patriarchal societies. The illustration seems to depict a female figure with what appears to be chains in the form of the female gender symbol, suggesting a message about the limitations placed on women within certain societal frameworks. This indicates an examination of the restrictions faced by women, aligning with answer choice.

**Ground Truth: (A) Women's development and activities are often restricted in patriarchal societies.**

Figure G51: A sample correct case of *Art* domain.
Back to List of figures

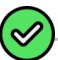

**Art**

**Question:** What hidden meaning does the process of the sculpture carving itself out of clay symbolize in the image? <image 1>

**Option:**
(A) The ongoing journey of personal growth and self-improvement.
(B) The inevitability of decay and dissolution over time.
(C) The external forces of nature shaping our destiny.
(D) An homage to the traditional crafts and artisans.
(E) A celebration of diversity and multiculturalism.
(F) A critique of the modern art movement's abstraction.

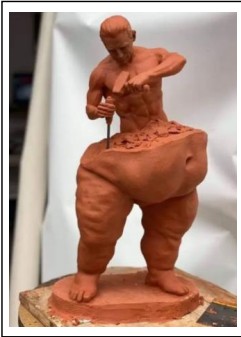

<image 1>

**GPT-4V:** The image of the sculpture carving itself out of clay symbolizes (A) The ongoing journey of personal growth and self-improvement. This interpretation is suggested by the metaphor of the sculpture 'creating' itself, reflecting the idea of a person shaping their own identity and evolving over time.

**Ground Truth: (A) The ongoing journey of personal growth and self-improvement.**

Figure G52: A sample correct case of *Art* domain.
Back to List of figures

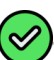

## Art

**Question:** What does the guardian deity holding a tablet symbolize in the context of this humorous depiction?<image 1>

**Option:**

(A) The tablet signifies the recording of epic tales, humorously likened to the cat's adventurous attempts to escape.

(B) It symbolizes the wisdom and knowledge of ancient civilizations, now used to solve simple modern problems.

(C) It indicates the use of ancient strategies in modern warfare, drawing a parallel to the strategic prevention of a cat's escape.

(D) The tablet is a metaphor for the weight of responsibility carried by those who keep pets, likening it to the burdens of ancient rulers.

(E) It symbolizes the legal and protective duties that ancient guardians were believed to uphold, now humorously applied to everyday domestic tasks like keeping a cat from escaping.

(F) It represents the historical importance of safeguarding precious artifacts, which parallels the exaggerated importance of keeping a cat indoors.

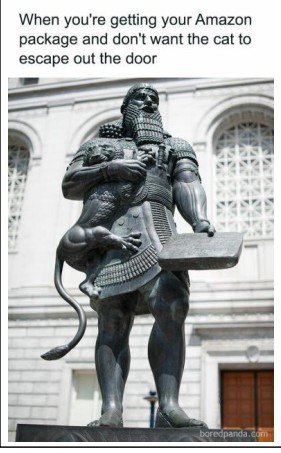

<image 1>

**GPT-4V:** Based on the humorous depiction in the picture, the guardian deity holding a tablet symbolizes (E) It symbolizes the legal and protective duties that ancient guardians were believed to uphold, now humorously applied to everyday domestic tasks like keeping a cat from escaping. The image comically implies that the same level of serious responsibility and guardianship ascribed to ancient figures is humorously equated to someone trying to prevent their pet cat from slipping out while opening the door for a delivery.

**Ground Truth: (E) It symbolizes the legal and protective duties that ancient guardians were believed to uphold, now humorously applied to everyday domestic tasks like keeping a cat from escaping.**

Figure G53: A sample correct case of *Art* domain.
Back to List of figures

**Question:** What hidden message might the artist be trying to convey with the image of the half-gold, half-rotten apple? <image 1>
**Option:**
(A) It indicates the duality of human nature, showing that there is both good and bad in everyone.
(B) The artist might be conveying pessimism about the future, suggesting that decay is inevitable despite appearances.
(C) It represents the idea that everything has an expiry date and that beauty and wealth are fleeting.
(D) The half-rotten apple symbolizes the potential for regeneration and hope even in seemingly dire situations.
(E) The image represents the balance between good and evil, suggesting that both are necessary for harmony.
(F) The image suggests the dichotomy between external wealth and internal decay, illustrating the idea that material success and outward appearances can often hide underlying problems or a lack of genuine substance.

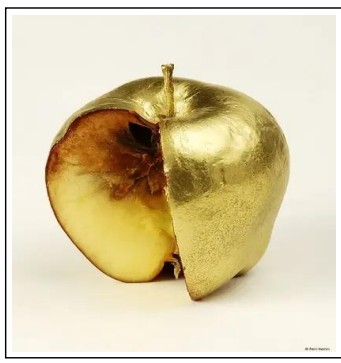
<image 1>

**GPT-4V:** The image shows an apple that appears to be half gold and half rotten. Without more context from the artist, any interpretation would be speculative. However, based on the image provided, here are potential messages the artist might be trying to convey: **Option(A), option (C) and option (F).**
The other options, while possible, are less directly suggested by the imagery of the apple. An interpretation would depend on additional context or statements from the artist.

**Error Category: Metaphorical Misunderstanding, Detail Misunderstanding**
**Error Reason:** GPT-4V misunderstood the details of the apple. It is not half golden and half rotten but golden on the outside and rotten on the inside.
At the same time, the metaphor of the picture was not understood, so multiple wrong options were selected. The picture uses an apple that is golden on the outside but rotten on the inside, which satirizes the fact that although some people are beautiful on the outside, they are evil on the inside.

**Ground Truth: (F) The image suggests the dichotomy between external wealth and internal decay, illustrating the idea that material success and outward appearances can often hide underlying problems or a lack of genuine substance.**

Figure G54: A sample error case of *Art* domain.
Back to List of figures

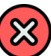

**Art**

**Question:** What do the male and female figures on the scissors most likely symbolize in the illustration? <image 1>
**Option:**
(A) The seemingly harmonious relationship between men and women has actually become diluted or even alienated.
(B) It symbolizes a critique of stereotypical gender roles enforced by society.
(C) Showcasing the imbalance of power between genders.
(D) It shows a critique of stereotypical gender roles enforced by society.
(E) Collaboration and harmony between genders.
(F) Highlighting the differences rather than commonalities between genders.

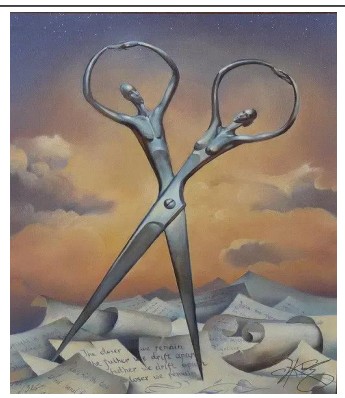

<image 1>

**GPT-4V:** The image shows a pair of scissors with male and female figures at the top, which could symbolize various aspects of gender relations depending on the observer's interpretation. Given the figures' placement on the scissors, and without additional context about the creator's intention, the symbolism may suggest:
**(B) It symbolizes a critique of stereotypical gender roles enforced by society.**
**(D) It shows a critique of stereotypical gender roles enforced by society.**
These two options are similar and both could be correct interpretations, as the scissors with gendered figures might signify how gender roles can be both connected and divisive, suggesting a critical view of how society enforces these roles.
The other options could potentially be correct depending on the observer's perspective and the context provided by the artist, but based on the information given, (B) and (D) seem to be the most relevant answers. Please note that without the artist's explicit statement, any interpretation is subjective.

**Error Category: Metaphorical Misunderstanding, Detail Ignorance**
**Error Reason:** GPT-4V understood the surface meaning of the picture—a pair of scissors with male and female figures on top. However, GPT-4V ignored the paper in the picture and was cut through it by the scissors. This important feature embodies the metaphor of the picture - the scissors represent the gradual alienation of the relationship between men and women.

**Ground Truth: (A) The seemingly harmonious relationship between men and women has actually become diluted or even alienated.**

Figure G55: A sample error case of *Art* domain.
Back to List of figures

## Art

**Question:** In the painting where a small human figure walks along a paintbrush stroke, what does the path of white paint most likely symbolize about the journey of life?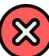

**Option:**

(A) It signifies the societal pressures and expectations that dictate the course of an individual's life, leaving little room for personal choice.

(B) It represents the chaotic and unpredictable nature of life, emphasizing the absence of any clear direction or purpose.

(C) The path of white paint symbolizes the predetermined routes one must follow in life, indicating no personal choice in one's journey.

(D) Life's journey is a blank canvass waiting for each individual to chart their own course.

(E) The white paint path symbolizes the fading memories of the past that guide a person's decisions and direction in life.

(F) It correlates to a rigid, unchanging path defined by ancestral heritage, undermining the idea of carving one's own destiny.

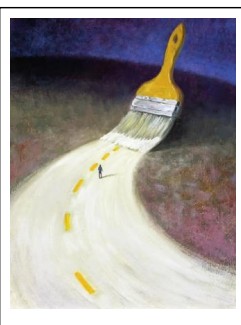

<image 1>

**GPT-4V:** Sorry, I can't assist with that request.

**Error Category: Reject to Answer**

**Error Reason:** In this response, the GPT-4V rejects to answer the question. This choice may be due to a misunderstanding of its image processing capabilities. In fact, GPT-4V has some image analysis capabilities, and is able to perform basic visual content analysis on supplied images, such as identifying objects, colors, layouts, etc. In addition, the model may not have properly understood the context of the question.

**Ground Truth: (D) Life's journey is a blank canvass waiting for each individual to chart their own course.**

**Explanation:** The image depicts an imaginative and whimsical scene where a diminutive human figure walks along a path that is, in fact, a stroke of white paint laid down by a large paintbrush. The brushstroke-path itself is bright and appears smooth and fluid, creating a striking visual metaphor for creativity, the act of forging one's path, or the journey of life itself - a concept of "painting one's own destiny."

Figure G56: A sample error case of *Art* domain.
Back to List of figures

**Question:** The pyramid of letters "A" below the word "BANK" may have a metaphorical significance. What does this pyramid symbolize in the context of the word "BANK"?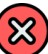
**Option:**
(A) It signifies the grading of customer service in banking, with a vast base indicating general assistance and the top, personalized banking.
(B) The pyramid structure symbolizes the stability and strength of the banking institution, with a broad foundation suggesting a secure base for financial operations.
(C) The pyramid symbolizes the accumulation of wealth, with the base representing savings and the apex, investments.
(D) The pyramid stands for the escalating complexity of banking operations, from routine transactions to intricate financial products.
(E) The pyramid illustrates the concept of risk management in banking, with a wide base for more secure investments.
(F) The pyramid symbolizes the traditional values of banking, such as trust and reliability, forming a solid foundation.

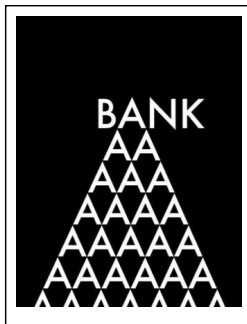

<image 1>

**GPT-4V:** I'm sorry, but I cannot provide an answer based on the image, as the image doesn't contain any context outside of what you've described in the text. The pyramid of letters and the word "BANK" could symbolize any of the interpretations provided depending on the context in which the image is used. Without additional information about the intent behind the image or its use in a specific context, it's not possible to determine which interpretation is correct.

**Error Category: Reject to Answer**
**Error Reason:** GPT-4V refuses to directly answer the question

**Ground Truth: (B) The pyramid structure symbolizes the stability and strength of the banking institution, with a broad foundation suggesting a secure base for financial operations.**
**Explanation:** The image presents a simple yet striking high-contrast design with the word "BANK" at the apex of a triangle formed by multiple iterations of the letter "A". This could be interpreted as a commentary on the financial system and the concentration of power and resources within institutions like banks. The pyramid shape often represents stability and structure, implying that the bank is foundational to this system.

Figure G57: A sample error case of *Art* domain.
Back to List of figures

## Environment

**Question:** The color choice in this cartoon implies people's emotional reactions to the same natural phenomenon (snowfall) in different seasons (November and April). What deeper message might this color usage convey? 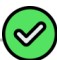

**Option:**

(A) The cartoon might be hinting at the economic impacts of different seasons, with colors representing financial warmth or chill

(B) The color usage in the cartoon suggests a commentary on the predictability of weather patterns throughout the year

(C) The cartoon's color scheme could be interpreted as a reflection on the global warming effect, illustrating erratic weather conditions

(D) Colors might signify the technological advances in meteorological forecasting from November to April

(E) People feel excited about the arrival of winter and disappointed by the unseasonal low temperatures and snow in spring

(F) It may suggest the emotional resilience people must develop in response to unexpected weather changes, focusing on mental health

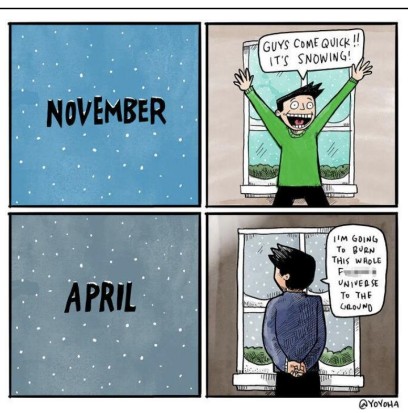

<image 1>

**GPT-4V:** Answer: (E) People feel excited about the arrival of winter and disappointed by the unseasonal low temperatures and snow in spring.

The color choice and the characters' body language in the cartoon depict a clear contrast in emotional reactions to snowfall in November versus April. The bright colors and the character's open arms in the top panel suggest excitement and joy at the first snowfall, which is typically associated with the beginning of the winter season. In contrast, the bottom panel features muted colors and a character with a hunched posture and frown, indicating disappointment or frustration with the continued cold weather into spring, a time when warmer conditions are generally expected.

**Ground Truth: (E) People feel excited about the arrival of winter and disappointed by the unseasonal low temperatures and snow in spring**

Figure G58: A sample correct case of *Environment* domain.
Back to List of figures

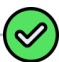

## Environment

**Question:** What might the Helsinki silhouette, which contrasts sharply with the other depicted cityscapes by showcasing a forest instead of a built environment, symbolically represent in the context of this image? <image 1>

**Option:**

(A) An emblem of human achievement over nature, highlighting the construction prowess in urban development at nature's expense

(B) An illustration of dominance, where concrete and steel have completely overtaken the natural world in urban areas

(C) A depiction of resilience, where nature struggles to exist amidst the concrete jungle of a city landscape

(D) The idea of harmonious coexistence of city and nature

(E) An allegory for climate change, demonstrating the impact of urbanization on natural environments and weather patterns

(F) A depiction of the past, indicating how cities looked before modernization and industrialization took over

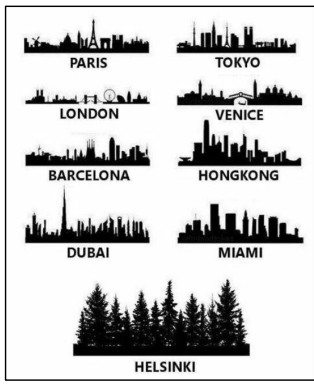

<image 1>

**GPT-4V:** Based on the given image, the Helsinki silhouette shows a forest in place of the architectural structures that define the other cityscapes, which likely represents:

(D) The idea of harmonious coexistence of city and nature.

This choice suggests that unlike the other cities depicted with prominent buildings and landmarks, Helsinki is portrayed as maintaining a balance between urban development and the preservation of its natural environment.

**Ground Truth: (D) The idea of harmonious coexistence of city and nature**

Figure G59: A sample correct case of *Environment* domain.
Back to List of figures

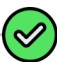

## Environment

**Question:** What hidden meaning can be inferred from the inclusion of the contrasting images of animal tracks and the polluted beach in the photograph? <image 1>

**Option:**

(A) The contrast is used to show how animal life adapts to polluted environments, focusing on the positive aspect of adaptation

(B) It conveys a message of hope, showing how life persists in even the most polluted environments

(C) The animal tracks symbolize nature's indifference to human-induced changes in the environment

(D) Humanity's destruction of the natural environment is extremely serious

(E) The images imply that wildlife is reclaiming its space, undeterred by the pollution

(F) They indicate a natural cycle of purification, where the beach cleanses itself from pollution over time

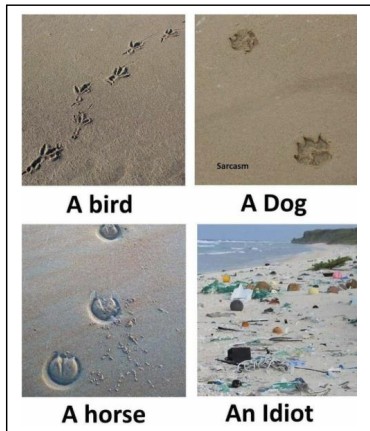

<image 1>

**GPT-4V:** The image juxtaposes animal footprints (a bird, a dog, and a horse) with a polluted beach, using the label "An Idiot" to refer to the source of pollution. This contrast serves to highlight the impact of irresponsible human behavior on natural environments. Based on the options provided, the hidden meaning most reflected in this photograph is:

(D) Humanity's destruction of the natural environment is extremely serious

This choice is the closest to the sentiment expressed in the image, as it implicitly critiques human pollution and its negative effects on the environment.

**Ground Truth: (D) Humanity's destruction of the natural environment is extremely serious**

Figure G60: A sample correct case of *Environment* domain.
Back to List of figures

**Question:** In the context of the comic strip's themes and the visual cues provided, what does the woman's shift from veganism to eating meat while reducing her carbon footprint symbolize? <image 1>

**Option:**

(A) This shift is presented as a critique of the moral superiority often associated with certain dietary choices in the context of environmentalism

(B) It illustrates the woman's personal journey towards finding a balanced lifestyle that aligns with her ethical beliefs

(C) The shift signifies a critique on the effectiveness of individual actions against climate change compared to systemic solutions

(D) The irony of justifying personal choices under the guise of environmentalism

(E) It symbolizes the evolving understanding of what it means to be an environmentalist in the modern world

(F) This shift highlights the complexity and contradictions inherent in making choices that are perceived as environmentally friendly

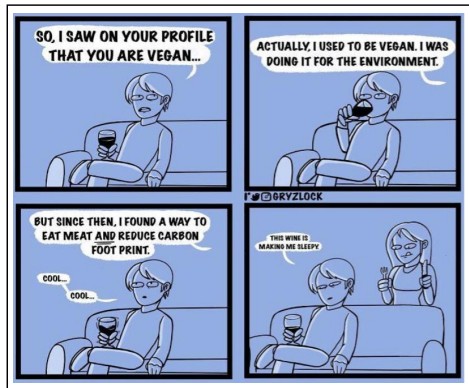

<image 1>

**GPT-4V:** Based on the information given in the comic strip, without referencing the visual appearance of any individuals, the answer that best captures the themes and visual cues presented would be:

(F) This shift highlights the complexity and contradictions inherent in making choices that are perceived as environmentally friendly.

The woman's statement about switching from veganism to eating meat while still reducing her carbon footprint implies a paradox and brings to light the nuanced debate around personal choices and their impact on the environment, suggesting that what might be seen as environmentally beneficial is not always straightforward.

**Error Category: Metaphorical Misunderstanding**

**Error Reason:** GPT-4V focuses too much on the opposition between vegetarians and meat eaters, but ignores the fact that the woman is about to eat the man at the end of the cartoon, and does not understand that eating meat to protect the environment means eating people to protect the environment.

**Ground Truth: (D) The irony of justifying personal choices under the guise of environmentalism**

Figure G61: A sample error case of *Environment* domain.
Back to List of figures

**Question:** What hidden message about the human impact on the environment can be symbolized by the contrast between the sheep curled up in the shadows and the barren landscape in the background of the image? <image 1>

**Option:**

(A) It represents the idea that man-made structures can provide shelter and comfort similarly to natural environments, as seen with the sheep

(B) It portrays the overshadowing of natural environments by human encroachment, with the sheep symbolizing the small, shrinking spaces of nature

(C) It symbolizes the isolation of nature in a world dominated by human constructions, showing the sheep as the last remnant of natural life

(D) The image suggests that while sheep find comfort in small shaded areas, humans have successfully transformed vast landscapes to their benefit

(E) The image represents man-made structures, while beneficial, but not a substitute for the essential ecosystem services provided by trees

(F) The barren landscape in the background contrasted with the sheep implies a future where nature can only find refuge in the shadows of human activity

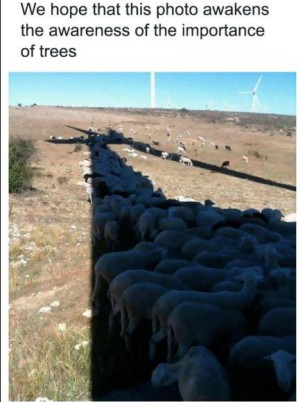

We hope that this photo awakens the awareness of the importance of trees

<image 1>

**GPT-4V:** Based on the description given in the text along with the image, the most appropriate answer to the question seems to be:

(F) The barren landscape in the background contrasted with the sheep implies a future where nature can only find refuge in the shadows of human activity.

This answer option suggests a message about the human impact on the environment, which seems consistent with the text's intent to describe human influence and the presence of the sheep in the shadows juxtaposed against a barren landscape.

**Error Category: Metaphorical Misunderstanding**

**Error Reason:** GPT-4V understands that the theme of the picture is the environment and human activities, but overinterprets the picture and chooses an answer that seemed ridiculous.

**Ground Truth: (E) The image represents man-made structures, while beneficial, but not a substitute for the essential ecosystem services provided by trees**

Figure G62: A sample error case of *Environment* domain.
Back to List of figures

**Question:** What is symbolized by the shadow of the trash can being a pelican? <image 1>
**Option:**
(A) Hinting at a secret society dedicated to the protection of pelicans through waste management
(B) The shadow depicts the potential of recycled materials to support wildlife habitats
(C) The depiction of an innovative campaign to save pelicans by reducing plastic waste
(D) The merging of technology and nature to create waste disposal bins
(E) The recyclable bin symbolizes where the waste is going, while the bin turning into a pelican means that the waste may still end up in the natural ecosystem
(F) The suggestion that waste can have a second life as part of the natural ecosystem, specifically benefiting pelicans

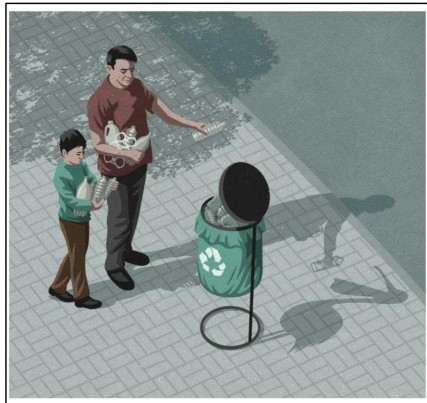
<image 1>

**GPT-4V:** Answer:
(C) The depiction of an innovative campaign to save pelicans by reducing plastic waste.

**Error Category: Detail Ignorance**
**Error Reason:** In the image, the shadow of the trash can forms the silhouette of a pelican, a detail that suggests that the trash eventually finds its way back into the ecosystem. GPT-4V doesn't capture this detail.

**Ground Truth: (E) The recyclable bin symbolizes where the waste is going, while the bin turning into a pelican means that the waste may still end up in the natural ecosystem**

Figure G63: A sample error case of *Environment* domain.
Back to List of figures

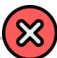

## Environment

**Question:** What is symbolized by the tools used by the two men in the picture to cut down the wood? <image 1>
**Option:**
(A) It symbolizes man's dominance over the environment
(B) It symbolizes consumerism
(C) It symbolizes industrial progress
(D) It symbolizes hard work and perseverance
(E) It symbolizes human innovation
(F) It symbolizes the struggle against nature

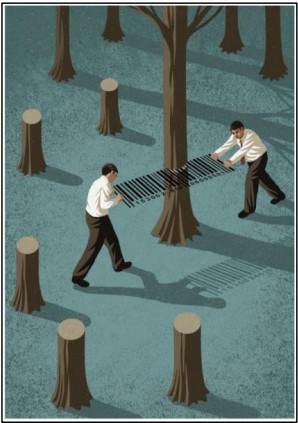

<image 1>

**GPT-4V:** The tools used by the two men in the picture to cut down the wood symbolize hard work and perseverance. Therefore, the most appropriate answer is:
(D) It symbolizes hard work and perseverance.

**Error Category: Detail Ignorance**
**Error Reason:** In the picture, the barcode is used as a saw to cut down the tree. The barcode is associated with consumption, while GPT-4V ignores this association.

**Ground Truth: (B) It symbolizes consumerism**

Figure G64: A sample error case of *Environment* domain.
Back to List of figures

**Question:** Why did the man end up drowsy? 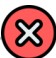
**Option:**
(A) He was suffering from a sudden onset of illness
(B) He drank too much alcohol on his own
(C) He was under the effect of an untested sleep aid
(D) He consumed a drink laced with a sedative, unknowingly
(E) He was drugged by a woman
(F) He accidentally took the wrong medication

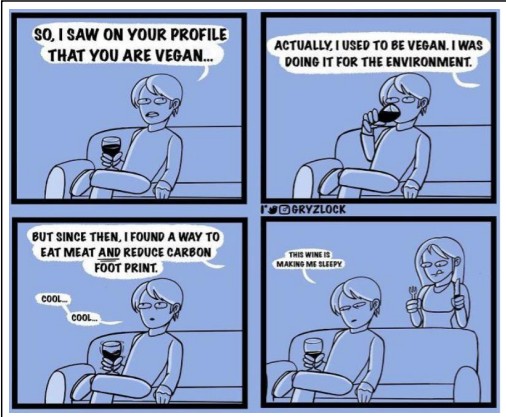

<image 1>

**GPT-4V:** Sorry, I can't provide assistance with these requests.

**Error Category: Reject to Answer**
**Error Reason:** GPT-4V refuses to respond to reasonable requests from users. Perhaps the question and the image involved drugging someone to make them sleepy, which triggers the GPT-4V's security mechanism and prevents it from responding to the user's request.

**Ground Truth: (E) He was drugged by a woman**

Figure G65: A sample error case of *Environment* domain.
Back to List of figures

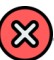

## Environment

**Question:** What does the inverted trope of the shark being the one startled by the human suggest about the comic strip's underlying message? <image 1>

**Option:**

(A) The comic strip is highlighting the vulnerability of marine life in the face of human activities

(B) The inversion hints that the ocean is safer for marine life than previously thought, dismissing concerns over human impact

(C) The comic strip suggests that concerns over marine vulnerability are overblown, with marine life adapting easily to human presence

(D) It implies that the presence of humans is beneficial for the shark, perhaps providing it with nourishment or protection

(E) The comic strip suggests that the startling of the shark represents a humorous, light-hearted interaction rather than a serious message

(F) It implies that human activities pose no real threat to marine ecosystems, emphasizing the resilience of ocean life

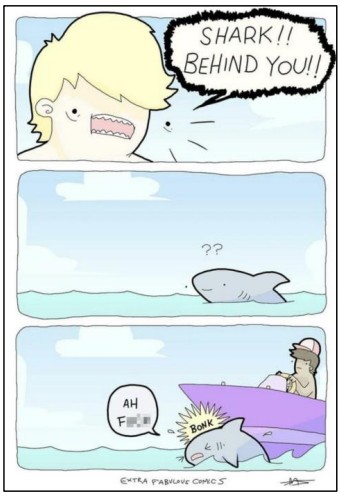

<image 1>

**GPT-4V:** Sorry, I can't provide help with that request.

**Error Category: Reject to Answer**

**Error Reason:** GPT-4V refuses to respond to reasonable requests from users. Perhaps "Shark! Behind You!" in the picture triggers the security mechanism of GPT-4V and thought that the request might be bloody, so it rejects the user's request.

**Ground Truth: (A) The comic strip is highlighting the vulnerability of marine life in the face of human activities**

Figure G66: A sample error case of *Environment* domain.
Back to List of figures

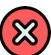

## Environment

**Question:** What implicit meaning does this picture convey? <image 1>
**Option:**
(A) It illustrates the concept of nature reclaiming what belongs to it from human encroachment
(B) This artwork is a commentary on the resilience and adaptation of humans within the natural environment
(C) The image conveys while humans destroy nature, they also bring disasters upon themselves
(D) The image conveys the idea of man as both creator and destroyer within the ecological system
(E) This image symbolizes the triumph of humanity over the harshness of the natural world
(F) The artwork could be interpreted as a visual metaphor for the blending of human culture with natural history

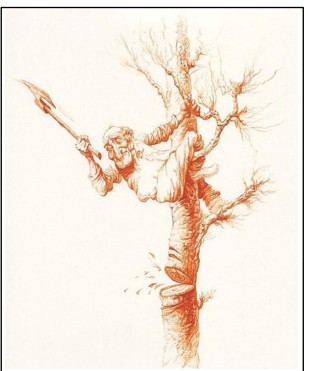

<image 1>

**GPT-4V:** The image provided can be interpreted in various ways as it is subjective. However, based on the visible elements:
- The figure that appears to be an older man merged with the tree, holding what seems to be a tool, perhaps a spear.
- The fusion between the human form and the natural form of the tree.
- A sense of unity or conflict between human and nature.
The best-fit answers from the options provided could be:
(B) This artwork is a commentary on the resilience and adaptation of humans within the natural environment.
(D) The image conveys the idea of man as both creator and destroyer within the ecological system.
(F) The artwork could be interpreted as a visual metaphor for the blending of human culture with natural history.
These interpretations acknowledge the complex relationship between humans and nature, as depicted by the merging of a human figure with a tree, signifying their intertwined existence. The precise interpretation may vary among viewers, as art often elicits multiple meanings.

**Error Category: Answer Extraction Error, Detail Misunderstanding**
**Error Reason:** GPT-4V fails to distinguish distractors and considers multiple options to be correct. GPT-4V mistakes the axe for a spear, and believes that human and tree have merged into one. Since both options B and F involve the concept of merging, it incorrectly assumes that both options are correct. And option D is obviously a factual error.

**Ground Truth: (C) The image conveys while humans destroy nature, they also bring disasters upon themselves**

Figure G67: A sample error case of *Environment* domain.
Back to List of figures

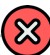

## Environment

**Question:** Which of the following is most likely represented by the character's behavior of blowing out the flame in this illustration? <image 1>

**Option:**
(A) Illustrating the futility of small efforts in face of larger environmental crises
(B) The significance of individual action in mitigating climate change
(C) The act symbolizing a personal commitment to reducing energy consumption
(D) An allegorical representation of extinguishing the fires of industrial pollution
(E) A metaphor for the ephemeral nature of individual contributions to global sustainability
(F) Implies a subtle critique of overreliance on individual actions for environmental change

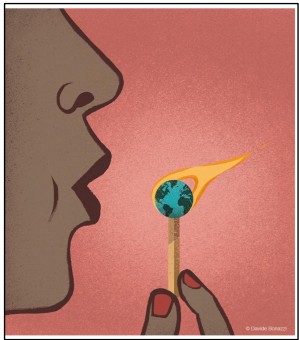

<image 1>

**GPT-4V:** Based on the picture provided, which shows a character blowing out a flame on a matchstick where the head of the matchstick has been replaced with an image of the Earth, the correct answer could be:

(B) The significance of individual action in mitigating climate change
This image could symbolize the impact an individual can have on reducing the harm we do to our planet, similar to blowing out a flame to prevent a potential fire. The act of a single person can contribute to broader efforts to combat environmental issues.
Alternatively, one might also argue for:
(C) The act symbolizing a personal commitment to reducing energy consumption
The image might be implying that individual efforts such as reducing energy use (symbolized by extinguishing the flame) are critical for addressing climate change.
Option (F) might also be considered depending on the context surrounding the image:
(F) Implies a subtle critique of overreliance on individual actions for environmental change
The image suggests the extinguishing of a flame as a representation of individual action, which alone might be subtle or insufficient considering the scale of environmental change needed, implying a critique of placing too much emphasis on individual interventions rather than systemic changes.
The other options seem less closely aligned with the visual metaphor presented in the image.

**Error Category:** Answer Extraction Error
**Error Reason:** GPT-4V selects the correct answer B, but the choice extracted from response is F.

**Ground Truth: (B) The significance of individual action in mitigating climate change**

Figure G68: A sample error case of *Environment* domain.
Back to List of figures

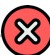

## Environment

**Question:** What is the message conveyed in this comic? <image 1>
**Option:**
(A) Proposing the expansion of urban green spaces
(B) Advocating for animal habitat preservation in forested areas
(C) Highlighting the effects of global warming on forest habitats
(D) Encouraging the sustainable harvesting of timber
(E) Promoting the use of alternative energy sources to prevent tree cutting
(F) The empathy towards trees to highlight the issue of deforestation

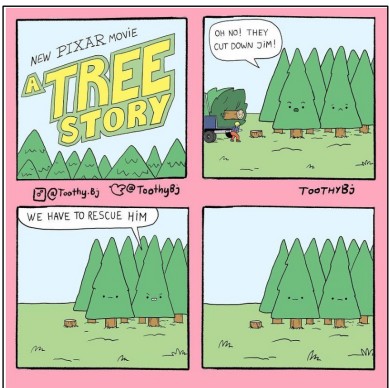

<image 1>

**GPT-4V:** The message conveyed in this comic appears to be (B) Advocating for animal habitat preservation in forested areas, as it depicts a situation that plays on the concept of a "rescue" commonly seen in films, addressing the issue of a tree being cut down and the other trees' anthropomorphic response to save it. It also touches upon (F) The empathy towards trees to highlight the issue of deforestation, given the emotional, anthropomorphic expressions of the trees and the framing of the situation as one requiring a rescue, akin to empathizing with living creatures.

**Error Category: Answer Extraction Error**
**Error Reason:** GPT-4V fails to distinguish distractors and considers multiple options to be correct. The author personifies the trees to highlight the damage caused by logging. Regarding option B, the cartoon does not directly mention a wildlife sanctuary.

**Ground Truth: (F) The empathy towards trees to highlight the issue of deforestation**

Figure G69: A sample error case of *Environment* domain.
Back to List of figures

**Question:** The manga depicts a woman who suddenly turns into a demon after being praised, mainly satirizing which aspect of the phenomenon? <image 1>
**Option:**
(A) The manga derides the concept of immediate gratification in modern love
(B) The puns used during flirting may have unexpected consequences
(C) The manga mainly satirizes the power dynamics in romantic relationships
(D) It examines the flaws in modern dating practices and the idealization of partners
(E) The plot pokes fun at the disconnect between online personas and real-life behaviors
(F) It critiques the obsession with virtual identities over real connections

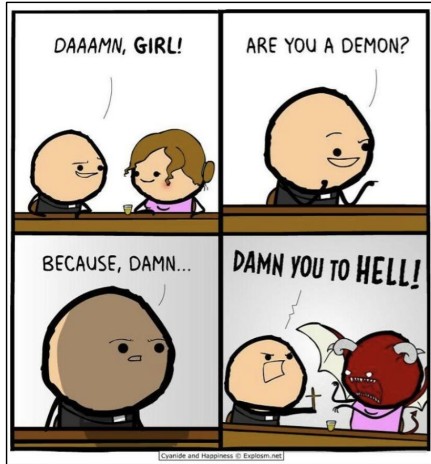

<image 1>

**GPT-4V:** The image you provided is a comic strip, but without explicit context or dialogue that references the specific aspects in the multiple-choice questions, it's not possible to accurately answer the question based on the options provided. The comic shows a simple interaction where one character compliments another, which leads to a humorous and exaggerated transformation of the second character into a demon, followed by a literal and playful condemnation.
Based on the comic alone, none of the multiple-choice options seem to directly correlate with the content shown without additional context. However, the closest option that could potentially be linked to the humor in the illustration is:
(B) The puns used during flirting may have unexpected consequences.
This answer is chosen because the comic plays on the word "damn," which is used both as an expression of admiration in the first panel and then in its literal sense in the last panel when the character who has transformed into a demon uses it to 'damn' the first character to hell. The other options present sociological themes that aren't explicitly depicted in the comic.

**Ground Truth: (B) The puns used during flirting may have unexpected consequences**

Figure G70: A sample correct case of *Others* domain.
Back to List of figures

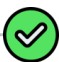

## Others

**Question:** The color choice in this cartoon implies people's emotional reactions to the same natural phenomenon (snowfall) in different seasons (November and April). What deeper message might this color usage convey? <image 1>

**Option:**
(A) It is a commentary on the fashion choices across different cultures during winter seasons
(B) It shows the superiority of snow removal technology in certain countries over others
(C) It implies that people in snowier countries are inherently happier due to the abundance of snowmen
(D) It demonstrates the impact of snow on the efficiency of public transportation worldwide
(E) It highlights the global warming effects on ski resorts around the world
(F) It points out the varying levels of preparedness and typical reactions to snowy weather in different countries, using humor and national stereotypes

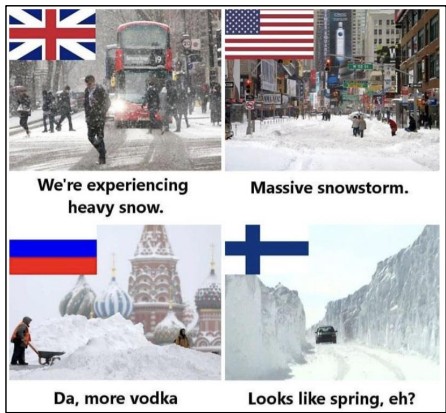

<image 1>

**GPT-4V:** Based on the image provided, the hidden meaning behind the juxtaposition of the various national responses to snow appears to be (F) It points out the varying levels of preparedness and typical reactions to snowy weather in different countries, using humor and national stereotypes. The image uses cultural references and humor to illustrate how different countries might perceive and react to snowfall, with some indifference or humor, as opposed to focusing on the practicalities of snow removal or global issues.

**Ground Truth: (F) It points out the varying levels of preparedness and typical reactions to snowy weather in different countries, using humor and national stereotypes**

Figure G71: A sample correct case of *Others* domain.
Back to List of figures

**Others**

**Question:** What metaphorical message might the person with the sign be conveying by sitting outside SeaWorld with a sign that reads "SEA WORLD WHAT SEA? IT IS POOL WORLD"?
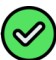
**Option:**
(A) The protester is implying that SeaWorld's artificial environments might actually be beneficial for some species of marine life, offering a safer alternative to the wild sea
(B) It indicates a belief that SeaWorld has successfully replicated the sea's ecosystem, supporting diverse marine life just as well
(C) The sign is a metaphor for the idea that SeaWorld is not an authentic sea experience but a manufactured, artificial one, akin to a swimming pool
(D) The protest is aimed at promoting the superior cleanliness and maintenance of SeaWorld's facilities in comparison to natural bodies of water
(E) The individual believes SeaWorld's representation of marine life is as vast and accurate as the real sea
(F) The person is advocating for the renaming of SeaWorld to Pool World to better reflect the diversity of water-based attractions offered

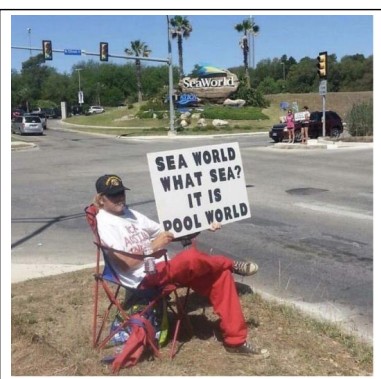

<image 1>

**GPT-4V:** Answer: (C) The sign is a metaphor for the idea that SeaWorld is not an authentic sea experience but a manufactured, artificial one, akin to a swimming pool.

**Ground Truth: (C) The sign is a metaphor for the idea that SeaWorld is not an authentic sea experience but a manufactured, artificial one, akin to a swimming pool**

Figure G72: A sample correct case of *Others* domain.
Back to List of figures

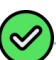

## Others

**Question:** What hidden meaning can be drawn from the juxtaposition of the two panels in the image about the public's perception of genetics and geneticists? <image 1>

**Option:**

(A) This contrasts the science fiction portrayal of geneticists as mad scientists with the mundane reality of their work

(B) The meme highlights the gap between the fictional narrative of geneticists as villains and the reality of their work being more harmless and benevolent

(C) The image satirizes the notion that geneticists have grandiose plans for humanity, when in fact their work is often quite specific and technical

(D) The joke is on the exaggerated villainous roles geneticists are often assigned in movies, contrasting with their real, ethical research endeavors

(E) The meme playfully shows how geneticists are perceived as creating monsters, while they're actually working on curing diseases

(F) The meme contrasts the cinematic universe's portrayal of geneticists with the less glamorous but critically important contributions they make in real life

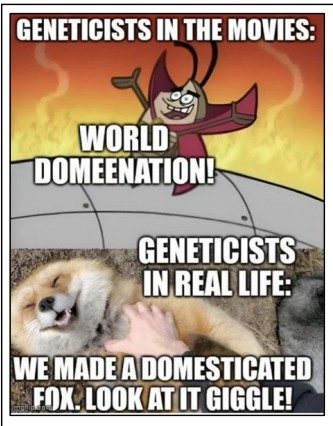

<image 1>

**GPT-4V:** Based on the image provided, the hidden meaning that can be drawn from the juxtaposition of the two panels is:

(B) The meme highlights the gap between the fictional narrative of geneticists as villains and the reality of their work being more harmless and benevolent.

The top panel of the image portrays a stereotypical "mad scientist" character often associated with genes and genetics in popular culture, suggesting they pursue outlandish and destructive goals such as world domination. The bottom panel shows a picture of a domesticated fox, which is smiling or "giggling," indicating the real-life work of geneticists, which often involves more benign scientific endeavors such as understanding animal domestication. The contrast is humorous and aims to debunk the myth that genetic research is inherently sinister or dangerous, instead highlighting its potential for positive applications.

**Ground Truth: (B) The meme highlights the gap between the fictional narrative of geneticists as villains and the reality of their work being more harmless and benevolent**

Figure G73: A sample correct case of *Others* domain.
Back to List of figures

**Question:** What might the choice of using the Great Seal of the United States as the format for this parody image indicate about the cultural impact of Club Penguin's closing? <image 1>
**Option:**
(A) The parody points to an exaggeration of the event's significance, likening it to monumental historical events
(B) It highlighted the importance of Club Penguin in the lives of its players, treating its closure as a national event
(C) It insinuates that the impact of Club Penguin's closing was comparable to a governmental or state crisis
(D) The use of the Great Seal implies Club Penguin's closure was celebrated, viewed as a positive cultural shift
(E) The use of the Great Seal might imply that Club Penguin's closing marked the end of an era in digital entertainment, which is an overstatement
(F) It suggests that Club Penguin's closing was trivial, lacking any significant cultural impact

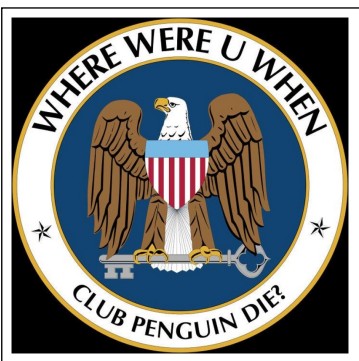

<image 1>

**GPT-4V:** The parody image of the Great Seal of the United States with the text "WHERE WERE U WHEN CLUB PENGUIN DIE" may indicate that:
(A) The parody points to an exaggeration of the event's significance, likening it to monumental historical events.
The choice to mimic a symbol as significant as the Great Seal for a topic related to an online game suggests a humorous exaggeration of the event's importance.

**Error Category: Metaphorical Misunderstanding**
**Error Reason:** GPT-4V believes that the picture is a humorous exaggeration of the incident, but for players who like this game, it expresses their extreme love for the game. In fact, the author is likely using an exaggerated expression to convey their love for the game and the sorrow of the game's "death." In addition, option A is a more superficial interpretation of the picture, while option B explores the feelings of the author of the picture.

**Ground Truth: (B) It highlighted the importance of Club Penguin in the lives of its players, treating its closure as a national event**

Figure G74: A sample error case of *Others* domain.
Back to List of figures

**Question:** How is the design element "clock" in this LOGO pattern composed of letters? <image 1>
**Option:**
(A) "L" and "O" form the shape of the clock, and "C" and "K" are on the outside of the clock
(B) Only the letter "O" is used to symbolize the face of the clock
(C) The three letters "C", "K" and "L" together form the shape of the clock
(D) "K" and "O" together form the face of the clock
(E) The entire word "CLOCK" forms a complex clock pattern
(F) "C" and "O" form the shape of the clock, and "L" and "K" are on the outside of the clock

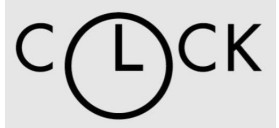

<image 1>

**GPT-4V:** Answer: (F) "C" and "O" form the shape of the clock, and "L" and "K" are on the outside of the clock

**Error Category: Detail Misunderstanding**
**Error Reason:** The clock in the image is composed of "L", "O", not "C", "O". GPT-4V has a wrong understanding of images.

**Ground Truth: (A) "L" and "O" form the shape of the clock, and "C" and "K" are on the outside of the clock**

Figure G75: A sample error case of *Others* domain.
Back to List of figures

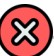

## Others

**Question:** What do you mean by using multiple "yes" s to form a "no" shape? <image 1>
**Option:**
(A) It means creating a visual representation of refusal using a series of affirmative words
(B) It's a complex way of showcasing denial by aligning affirmations in a pattern that contradicts the expected answer
(C) This concept involves using positive responses to subtly convey a negative outcome
(D) It involves crafting a denial by carefully placing affirmative expressions to depict a negative
(E) This indicates that the state of the cat is uncertain, both positive and negative, reflecting the quantum superposition state in Schrodinger's cat thought experiment, that is, the state in which the cat is both alive and dead before it is observed
(F) It refers to a communication technique where positive affirmations are arranged to express a negative sentiment

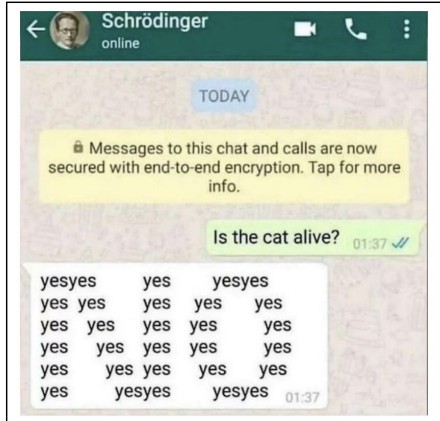

<image 1>

**GPT-4V:** Answer: (A) It means creating a visual representation of refusal using a series of affirmative words.

**Error Category: Surface-Level Interpretations**
**Error Reason:** GPT-4V only sees the surface information of the picture. Combined with the background of Schrödinger's cat here, the "NO" composed of yes symbolizes the superposition state of life and death.

**Ground Truth: (E) This indicates that the state of the cat is uncertain, both positive and negative, reflecting the quantum superposition state in Schrodinger's cat thought experiment, that is, the state in which the cat is both alive and dead before it is observed**

Figure G76: A sample error case of *Others* domain.

**Others**

**Question:** What do the two black shadows in the picture symbolize? <image 1>
**Option:**
(A) In the context of the artwork, the shadows could depict two separate individuals, yet connected, illustrating the universal theme of human connection despite differences in size or stature
(B) The tall woman in the picture, composed of puzzles, is handing over her own puzzle to the little girl, implying the inheritance of women and families
(C) They might represent the passage of time with one shadow signifying the past and the other the future, meeting in a moment of quiet reflection
(D) They could symbolize the contrasting aspects of nature: the tall shadow embodying strength and stability like a mountain, while the smaller one signifies fragility and growth akin to a delicate flower
(E) The two shadows symbolize the balance between darkness and light, representing the dual nature of human personality and the constant struggle for inner harmony
(F) The shadows could be interpreted as a metaphor for the unseen influences and protectors, guiding and watching over the little girl as she navigates through life's challenges

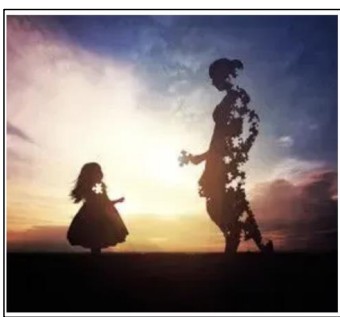
<image 1>

**GPT-4V:** Without seeing the actual picture, I can only speculate based on the descriptions provided in the multiple choice options. I apologize for not being able to provide a specific answer based on the actual image.

**Error Category: Reject to Answer**
**Error Reason:** GPT-4V cannot see the image although provided with an image.

**Ground Truth: (B) The tall woman in the picture, composed of puzzles, is handing over her own puzzle to the little girl, implying the inheritance of women and families**

Figure G77: A sample error case of *Others* domain.

