# Supplementary Materials

## A    General Datasheet of Dataset

### A.1    Motivation

**For what purpose was the dataset created?** Was there a specific task in mind? Was there a specific gap that needed to be filled? Please provide a description.

The dataset was created to fill a significant gap in the assessment of higher-order perceptual capabilities of Multimodal Large Language Models (MLLMs). Specifically, it aimed to evaluate these models' abilities to understand the nuanced emotional understanding and profound meaning extraction from images, which current benchmarks do not sufficiently explore.

**Who created the dataset (e.g., which team, research group) and on behalf of which entity (e.g., company, institution, organization)?**

The dataset is created by researchers from Shenzhen Institute of Advanced Technology (Chinese Academy of Science), 01.ai, and Multimodal Art Projection (M-A-P).

**Who funded the creation of the dataset?** If there is an associated grant, please provide the name of the grantor and the grant name and number.

No

### A.2    Composition

**What do the instances that comprise the dataset represent (e.g., documents, photos, people, countries)?** Are there multiple types of instances (e.g., movies, users, and ratings; people and interactions between them; nodes and edges)? Please provide a description.

The instances that comprise the II-Bench dataset represent a diverse collection of images, including illustrations, memes, posters, comics, logos, and paintings. These images span across six distinct domains: life, art, society, psychology, environment, and others, and are designed to challenge and evaluate the higher-order perceptual, reasoning, and comprehension abilities of Multimodal Large Language Models (MLLMs).

**How many instances are there in total (of each type, if appropriate)?**

II-Bench comprises a total of 1,222 images, with each image accompanied by 1 to 3 multiple-choice questions, resulting in a total of 1,434 questions. These images span across six distinct domains: life, art, society, psychology, environment, and others. Additionally, the images are classified based on the emotional tone they convey: Positive, Neutral, or Negative, and are annotated with rhetorical devices such as Metaphor, Exaggeration, Symbolism, Contrast, and others.

**Does the dataset contain all possible instances or is it a sample (not necessarily random) of instances from a larger set?** If the dataset is a sample, then what is the larger set? Is the sample representative of the larger set (e.g., geographic coverage)? If so, please describe how this representativeness was validated/verified. If it is not representative of the larger set, please describe why not (e.g., to cover a more diverse range of instances, because instances were withheld or unavailable).

The dataset contains all possible instances, which have been meticulously collected and annotated.

**What data does each instance consist of?** "Raw" data (e.g., unprocessed text or images)or features? In either case, please provide a description.

Each instance in the II-Bench dataset consists of the following data components:

- An image: Each image is a high-quality visual representation that falls into one of the specified categories such as illustrations, memes, posters, comics, logos, and paintings.
- Domain: Each image is classified into one of the six domains: Life, Art, Society, Psychology, Environment, or Others.
- Difficulty: Each image is classified into one of the three difficulties: Easy, Middle, or Hard.

- Emotion: Each image is classified into one of the three emotions: Positive, Neutral, or Negative.

- Rhetoric: Each image is annotated with the rhetoric used, such as Metaphor, Exaggeration, Symbolism, Contrast, Visual Dislocation, Antithesis, Analogy, Personification, or Others.

- Explanation: Each image contains an explanation of contained visual implications.

- Multiple-choice questions: Each image is associated with 1 to 3 multiple-choice questions designed to probe the understanding of the image's metaphorical implications. Each question has six options, including one correct answer and five distractors.

The images and their corresponding annotations are the "raw" data that form the basis of the II-Bench dataset, which is used to evaluate the higher-order perceptual capabilities of MLLMs. The dataset does not include processed features but relies on the richness of the images and the nuanced questions to assess model performance.

**Is there a label or target associated with each instance?** If so, please provide a description.

Each instance includes corresponding labels: domain, image type, difficulty, emotion, rhetoric, explanation and the correct answer.

**Is any information missing from individual instances?** If so, please provide a description, explaining why this information is missing (e.g., because it was unavailable). This does not include intentionally removed information, but might include, e.g., redacted text.

No

**Are there recommended data splits (e.g., training, development/validation, testing)?** If so, please provide a description of these splits, explaining the rationale behind them.

The dataset includes a total of 1,434 multiple-choice questions, with 1,399 questions used to construct the test set and 35 questions used to construct the development and validation set for few-shot tasks.

**Are there any errors, sources of noise, or redundancies in the dataset?** If so, please provide a description.

No

**Is the dataset self-contained, or does it link to or otherwise rely on external resources (e.g., websites, tweets, other datasets)?** If it links to or relies on external resources, a) are there guarantees that they will exist, and remain constant, over time; b) are there official archival versions of the complete dataset (i.e., including the external resources as they existed at the time the dataset was created); c) are there any restrictions (e.g., licenses, fees) associated with any of the external resources that might apply to a dataset consumer? Please provide descriptions of all external resources and any restrictions associated with them, as well as links or other access points, as appropriate.

The dataset is entirely self-contained.

**Does the dataset contain data that might be considered confidential(e.g., data that is protected by legal privilege or by doctor-patient confidentiality, data that includes the content of individuals' nonpublic communications)?** If so, please provide a description.

No

**Does the dataset contain data that, if viewed directly, might be offensive, insulting, threatening, or might otherwise cause anxiety?** If so, please describe why.

Although we have filtered and sifted as much as possible, some information may be potentially offensive. We are committed to continuous monitoring and improvement to mitigate such biases.

**Does the dataset identify any subpopulations (e.g., by age, gender)?**If so, please describe how these subpopulations are identified and provide a description of their respective distributions within the dataset.

No

**Is it possible to identify individuals (i.e., one or more natural persons), either directly or indirectly (i.e., in combination with other data) from the dataset?** If so, please describe how.

It is impossible to identify individuals from the dataset information.

**Does the dataset contain data that might be considered sensitive in any way (e.g., data that reveals race or ethnic origins, sexual orientations, religious beliefs, political opinions or union memberships, or locations; financial or health data; biometric or genetic data; forms of government identification, such as social security numbers; criminal history)?** If so, please provide a description

In developing II-Bench, we strictly adhere to ethical guidelines and legal regulations, ensuring fairness, transparency, inclusivity and respect for all stakeholders. However, we recognize that biases can inadvertently arise and some information may be potentially offensive. We are committed to continuous monitoring and improvement to mitigate such biases. Furthermore, we encourage users of our dataset to employ it responsibly and to consider the ethical implications of their work, particularly in applications that may impact individuals or communities.

### A.3   Collection Process

**How was the data associated with each instance acquired?** Was the data directly observable (e.g., raw text, movie ratings), reported by subjects (e.g., survey responses), or indirectly inferred/derived from other data(e.g., part-of-speech tags, model-based guesses for age or language)? If the data was reported by subjects or indirectly inferred/derived from other data, was the data validated/verified? If so, please describe how.

The data associated with each instance in the II-Bench dataset was acquired through a combination of direct observation and manual annotation. The images that constitute the dataset were directly observable, sourced from various renowned illustration websites. These images were manually collected by the research team, ensuring adherence to copyright and license regulations.

The annotations for each image, including domain, difficulty level, emotional tone, and rhetorical devices, were derived from the subjective interpretation of the annotators, who were instructed to identify and articulate the metaphorical implications of the images. Each image was annotated with multiple-choice questions and corresponding options, which required the annotators to create questions based on their understanding of the images' implications.

The dataset underwent a rigorous data curation process that included deduplication, text-to-image ratio control, and a visual inspection to ensure the images were relevant and of high quality. The annotation process involved multiple rounds of review and validation by the annotators to ensure consistency and accuracy in the annotations.

**What mechanisms or procedures were used to collect the data (e.g., hardware apparatuses or sensors, manual human curation, software programs, software APIs)?**How were these mechanisms or procedures validated?

We collect raw images using web scraping methods from various renowned illustration websites, ensuring a sufficiently extensive raw dataset.

**If the dataset is a sample from a larger set, what was the sampling strategy (e.g., deterministic, probabilistic with specific sampling probabilities)?**

The dataset is not a sample from a larger set. The dataset contains all possible instances, and each instance has been carefully collected and labeled by us.

**Who was involved in the data collection process (e.g., students, crowdworkers, contractors) and how were they compensated (e.g., how much were crowdworkers paid)?**

The authors of the paper were involved in the data collection and annotation process. All collectors and annotators are undergraduate students or have higher degrees, ensuring they can verify the annotated questions and related explanations.

**Over what timeframe was the data collected?** Does this timeframe match the creation timeframe of the data associated with the instances (e.g., recent crawl of old news articles)? If not, please describe the timeframe in which the data associated with the instances was created.

The data was collected from December 2023 to January 2024. However, the data associated with the instances was crawled during that period and may include images that were originally published earlier, similar to a recent crawl of old news articles.

**Did you collect the data from the individuals in question directly, or obtain it via third parties or other sources (e.g., websites)?**

The data for the II-Bench dataset was obtained via third-party sources, specifically from various renowned illustration websites. The collectors were instructed to adhere to copyright and license regulations, ensuring that the images were sourced ethically and legally.

### A.4 Preprocessing/cleaning/labeling

**Was any preprocessing/cleaning/labeling of the data done (e.g., discretization or bucketing, tokenization, part-of-speech tagging, SIFT feature extraction, removal of instances, processing of missing values)?** If so, please provide a description. If not, you may skip the remaining questions in this section.

The dataset underwent a rigorous preprocessing phase that included deduplication, text-to-image ratio control, and a visual inspection to ensure the images contained metaphorical or suggestive implications. Each image was annotated with difficulty, image type, domain, rhetorical devices, and multiple-choice questions with one correct answer and five distractors

**Was the "raw" data saved in addition to the preprocessed/cleaned/labeled data (e.g., to support unanticipated future uses)?** If so, please provide a link or other access point to the "raw" data.

No

### A.5 Uses

**Has the dataset been used for any tasks already?** If so, please provide a description.

No

**Is there a repository that links to any or all papers or systems that use the dataset?** If so, please provide a link or other access point.

No

**What (other) tasks could the dataset be used for?**

The dataset could be used for image implication understanding and image classification tasks.

**Is there anything about the composition of the dataset or the way it was collected and prepro-cessed/cleaned/labeled that might impact future uses?** For example, is there anything that a dataset consumer might need to know to avoid uses that could result in unfair treatment of individuals or groups (e.g., stereotyping, quality of service issues) or other risks or harms (e.g., legal risks, financial harms)? If so, please provide a description. Is there anything a dataset consumer could do to mitigate these risks or harms?

There is little risk here when we have cleaned the dataset.

### A.6 Distribution

**Will the dataset be distributed to third parties outside of the entity (e.g., company, institution, organization) on behalf of which the dataset was created?** If so, please provide a description.

The dataset is publicly available on the internet.

**How will the dataset will be distributed (e.g., tarball on website, API, GitHub)?** Does the dataset have a digital object identifier (DOI)?

The dataset is distributed via HuggingFace (`https://huggingface.co/datasets/m-a-p/II-Bench`), GitHub (`https://github.com/II-Bench/II-Bench`) and homepage (`https://ii-bench.github.io/`).

**When will the dataset be distributed?**

The dataset will be distributed in June 2024.

**Will the dataset be distributed under a copyright or other intellectual property (IP) license, and/or under applicable terms of use (ToU)?** If so, please describe this license and/or ToU, and provide a link or other access point to, or otherwise reproduce, any relevant licensing terms or ToU, as well as any fees associated with these restrictions.

This dataset is licensed under the Apache 2.0 License (`https://www.apache.org/licenses/LICENSE-2.0`). There is a request to cite the corresponding paper if the dataset is used.

**Have any third parties imposed IP-based or other restrictions on the data associated with the instances?** If so, please describe these restrictions, and provide a link or other access point to, or otherwise reproduce, any relevant licensing terms, as well as any fees associated with these restrictions.

No

**Do any export controls or other regulatory restrictions apply to the dataset or to individual instances?** If so, please describe these restrictions, and provide a link or other access point to, or otherwise reproduce, any supporting documentation.

Unknown to authors of the dataset.

## A.7 Maintenance

**Who will be supporting/hosting/maintaining the dataset?** Ziqiang Liu, Feiteng Fang, Xi Feng, Xinrun Du, Chenhao Zhang, Ge Zhang and Shiwen Ni are supporting/maintaining the dataset.

**How can the owner/curator/manager of the dataset be contacted (e.g., email address)?**

The manager of the dataset can be contacted via the following emails:

Ziqiang Liu: `zq.liu4@siat.ac.cn`
Feiteng Fang: `feitengfang@mail.ustc.edu.cn`
Xi Feng: `fengxi@ustc.edu`
Xinrun Du: `duxinrun2000@gmail.com`
Chenhao Zhang: `ch_zhang@hust.edu.cn`
Ge Zhang: `gezhang@umich.edu`
Shiwen Ni: `sw.ni@siat.ac.cn`

**Is there an erratum?** If so, please provide a link or other access point.

Not yet found.

**Will the dataset be updated (e.g., to correct labeling errors, add new instances, delete instances)?** If so, please describe how often, by whom, and how updates will be communicated to dataset consumers (e.g., mailing list, GitHub)?

This will be posted on the Dataset webpage (`https://ii-bench.github.io/`.

**Will older versions of the dataset continue to be supported/hosted/maintained?** If so, please describe how. If not, please describe how its obsolescence will be communicated to dataset consumers.

We do not maintain old versions of the dataset, if we update the version of the dataset, we will put the specific details of the dataset update on the relevant GitHub (`https://github.com/II-Bench/II-Bench`).

**If others want to extend/augment/build on/contribute to the dataset, is there a mechanism for them to do so?** If so, please provide a description. Will these contributions be validated/verified? If so, please describe how. If not, why not? Is there a process for communicating/distributing these contributions to dataset consumers? If so, please provide a description.

If others want to extend/augment/build on/contribute to the dataset, please contact the original authors about incorporating fixes/extensions.

## B  URL to website/platform

The code to reproduce the results of the paper is available at `https://github.com/II-Bench/II-Bench`.

We have also created a webpage for the dataset: `https://ii-bench.github.io/`.

## C  URL to Croissant metadata

The II-Bench dataset in croissant format is publicly available at `https://huggingface.co/datasets/m-a-p/II-Bench`.

## D  Author Statement and Data License

As the authors of this research, we hereby declare:

- We confirm that the dataset used in this study has been legally obtained and complies with the relevant data licensing agreement. We have thoroughly reviewed and ensured that the use of the data adheres to all applicable laws and regulations.
- We acknowledge that the dataset is licensed under the Apache License 2.0 (`https://www.apache.org/licenses/LICENSE-2.0`). We have adhered to the terms and conditions of this license in our use of the dataset.
- We bear full responsibility for any violations of rights or any other issues that may arise from the use of this dataset. We commit to addressing any such issues promptly and in accordance with legal and ethical standards.

Please refer to Datasheet A.6 for more details.

## E  Maintenance plan

Please refer to Datasheet A.7 for more details.

## F  Reproducibility

The code to reproduce the results of the paper is available at `https://github.com/II-Bench/II-Bench`.