# OpenReview forum: "II-Bench: An Image Implication Understanding Benchmark for Multimodal Large Language Models"
_NeurIPS.cc/2024/Datasets_and_Benchmarks_Track — NeurIPS 2024 Track Datasets and Benchmarks Poster_

### Official Review · Reviewer_C8Pk · 2024-07-24
**Overall good article, experimentation could be enhanced.**

**Rating:** 8
**Confidence:** 4

**Review:**

The paper presents a high-quality, clearly written, and easy-to-understand exposition of a novel image understanding benchmark, II-Bench, along with its publicly available dataset. The work provides significant insights and contributions to the community, particularly in the realm of multimodal large language models (MLLMs) and their ability to comprehend complex image semantics

**Strengths:**

1.	The paper identifies the gap between LLMs and human performance in understanding image implications. Through comparative experiments, it highlights the discrepancies in complex semantic image understanding between humans and various existing LLM versions, introducing a new metric focused on image implication understanding capabilities.
2.	The dataset includes a variety of modalities, styles, domains, and emotional inclinations, making it a valuable resource for the community. Its comprehensive nature contributes significantly to the advancement of research in this field.
3.	The study is well-supported by a diverse range of experiments, including analyses of mainstream prompt skills, ablation studies on the information provided, and detailed error analyses. This thorough approach enhances the credibility and robustness of the findings.

**Additional Feedback:**

NO

**Clarity:**

The paper is well-written, with a clear presentation of ideas and findings. The language is accessible, making it easy for readers to understand the authors' intentions and conclusions.

**Correctness:**

The claims made in the submission are well-supported by the data and experimental results. The construction of the dataset appears sound, and the evaluation methods and experimental design are appropriate and correctly implemented. The work is rigorous in its approach, ensuring that the findings are reliable and valid.

**Documentation:**

The paper provides sufficient detail on the data collection and organization processes for the II-Bench dataset. It includes descriptions of the types of images used, the sources of the images, and the criteria for selection. The dataset comprises a diverse set of images, encompassing various modalities, styles, domains, and emotional inclinations, making it a comprehensive resource for evaluating MLLMs.

**Ethics:**

The paper addresses ethical considerations related to the dataset, including potential biases and the responsible use of the data. It provides guidelines for the ethical use of the dataset, emphasizing transparency and the importance of considering the ethical implications of research conducted with the dataset.

**Limitations:**

1.	The absence of experiments involving fine-tuning on highly relevant datasets limits the paper's ability to fully explore the models' potential improvements. Including this aspect could have provided deeper insights into the models' capabilities and limitations.

**Opportunities For Improvement:**

1.	The paper could improve by including tests on model performance after fine-tuning with data highly relevant to the task of understanding image implications. This would provide a clearer picture of how adaptable and improveable the models are with targeted training.

**Relation To Prior Work:**

The paper clearly discusses how its contributions differ from previous work, especially in terms of introducing a new benchmark specifically focused on the higher-order perceptual abilities of MLLMs. It situates its contributions within the broader context of existing research, highlighting the unique aspects of the II-Bench dataset and its potential impact on future studies.

**Summary And Contributions:**

II-Bench is a novel benchmark focusing on evaluating the higher-order perception of MLLMs, addressing a previously unexplored area. Through extensive experimentation, the paper highlights the shortcomings of current MLLMs in understanding abstract and complex images and their inherent deficiencies in sentiment comprehension.

---

> ### Author Response · Authors · 2024-08-17
> **Response to Reviewer C8Pk**
>
> > **Concern1**: The paper could improve by including tests on model performance after fine-tuning with data highly relevant to the task of understanding image implications. This would provide a clearer picture of how adaptable and improveable the models are with targeted training.
>
> Thank you for your insightful feedback. We appreciate your suggestion to include tests on model performance after fine-tuning with data highly relevant to the task of understanding image implications. We would like to address your points as follows:
>
> 1. **Scope of Our Work**: Our primary objective in this study is to explore whether general MLLMs possess the higher-order capability to perceive image implications without task-specific fine-tuning.
>
> 2. **General & Task-Specific Capability**: As Reviewer `SyRn` pointed out, scaling this dataset is challenging. We believe that the understanding of visual implications is a general cognitive ability rather than a domain-specific skill. Our emphasis is on investigating the emergent capabilities of general MLLMls rather than their performance after fine-tuning.
>
> 3. **Future Work**: Actually, we agree that fine-tuning on the target task is a valuable direction for future research. We plan to explore this in subsequent studies to further investigate the potential improvements in model performance with targeted training.
>
> Thank you once again for your valuable feedback. We hope this clarifies our approach and the rationale behind it.
>
> Finally, we are very grateful for your careful review and valuable suggestions. Please feel free to comment if there are any further confusions and concerns, we will respond as soon as possible. We are committed to continuing our contributions to the LLM and open-source communities.

---

> ### Author Rebuttal · Authors · 2024-08-17
>
> # Response to Reviewer C8Pk
>
> > **Concern1**: The paper could improve by including tests on model performance after fine-tuning with data highly relevant to the task of understanding image implications. This would provide a clearer picture of how adaptable and improveable the models are with targeted training.
>
> Thank you for your insightful feedback. We appreciate your suggestion to include tests on model performance after fine-tuning with data highly relevant to the task of understanding image implications. We would like to address your points as follows:
>
> 1. **Scope of Our Work**: Our primary objective in this study is to explore whether general MLLMs possess the higher-order capability to perceive image implications without task-specific fine-tuning.
>
> 2. **General & Task-Specific Capability**: As Reviewer `SyRn` pointed out, scaling this dataset is challenging. We believe that the understanding of visual implications is a general cognitive ability rather than a domain-specific skill. Our emphasis is on investigating the emergent capabilities of general MLLMls rather than their performance after fine-tuning.
>
> 3. **Future Work**: Actually, we agree that fine-tuning on the target task is a valuable direction for future research. We plan to explore this in subsequent studies to further investigate the potential improvements in model performance with targeted training.
>
> Thank you once again for your valuable feedback. We hope this clarifies our approach and the rationale behind it.
>
> Finally, we are very grateful for your careful review and valuable suggestions. Please feel free to comment if there are any further confusions and concerns, we will respond as soon as possible. We are committed to continuing our contributions to the LLM and open-source communities.

---

> ### Author Response · Authors · 2024-08-22
> **Kindly Reminder to Reviewer C8Pk**
>
> Thank you once again for dedicating your valuable time to reviewing our paper. We sincerely hope you can participate in the Reviewer-Author discussion, as it is crucial for us to fully address your concerns and improve our paper during this period. As the deadline approaches, we kindly request your feedback on our rebuttal. We are eager to engage in further discussion and address any additional concerns you may have.

---

### Official Review · Reviewer_SyRn · 2024-07-25
**Review of II-Bench: An Image Implication Understanding Benchmark for Multimodal Large Language Models**

**Rating:** 5
**Confidence:** 5
**Correctness:** None
**Clarity:** None

**Review:**

## Strengths：

1. There has been little exploration of metaphor understanding in images within MLLM evaluation studies. The test set proposed by this work is highly innovative.

2. A meticulous image selection strategy has been established, including diversity screening, using OCR models to filter out images with a high proportion of text, and manual judgment.

3. The paper sets multiple evaluation methods for the models, including few-shot, CoT, Domain, Emotion, and Rhetoric, allowing a more comprehensive exploration of the factors affecting models' metaphor understanding capabilities.

## Weaknesses：

1. The dataset is created in a multiple-choice format, which in some cases can hinder evaluation, as models may understand the question and provide the correct answer but fail to output the correct option.

2. The dataset is challenging to expand, as each piece of data requires careful manual creation, consuming significant time and effort.

3. The data comes from well-known illustration websites, which might have been included in some models' training sets, posing a risk of data leakage and leading to unfair evaluations.

4. Each question matches six options, resulting in overly long contexts. Some models have low capabilities for handling long contexts, leading to unfairness during the evaluation process.

**Strengths:**

## Strengths：

1. There has been little exploration of metaphor understanding in images within MLLM evaluation studies. The test set proposed by this work is highly innovative.

2. A meticulous image selection strategy has been established, including diversity screening, using OCR models to filter out images with a high proportion of text, and manual judgment.

3. The paper sets multiple evaluation methods for the models, including few-shot, CoT, Domain, Emotion, and Rhetoric, allowing a more comprehensive exploration of the factors affecting models' metaphor understanding capabilities.

## Weaknesses：

1. The dataset is created in a multiple-choice format, which in some cases can hinder evaluation, as models may understand the question and provide the correct answer but fail to output the correct option.

2. The dataset is challenging to expand, as each piece of data requires careful manual creation, consuming significant time and effort.

3. The data comes from well-known illustration websites, which might have been included in some models' training sets, posing a risk of data leakage and leading to unfair evaluations.

4. Each question matches six options, resulting in overly long contexts. Some models have low capabilities for handling long contexts, leading to unfairness during the evaluation process.

**Additional Feedback:**

None

**Documentation:**

None

**Opportunities For Improvement:**

None

**Relation To Prior Work:**

None

**Summary And Contributions:**

This paper introduces the Image Metaphor Understanding Benchmark (II-Bench), which aims to assess models' advanced perception, reasoning, and understanding capabilities when dealing with complex metaphorical images. It uses benchmarks such as similarity and visual information proportion, combined with human evaluation, to select high-quality images. Questions and answers for these images are designed by humans, covering fields such as life, art, society, psychology, the environment, and others.

---

> ### Author Response · Authors · 2024-08-17
> **Response to Reviewer SyRn(1/n)**
>
> We sincerely thank you for your time and constructive suggestions. The insights provided have been instrumental in refining our work. We would like to provide our detailed responses and explanations to the issues raised.
>
> > **Concern1**: The dataset is created in a multiple-choice format, which in some cases can hinder evaluation, as models may understand the question and provide the correct answer but fail to output the correct option.
>
>
> Thank you for your insightful feedback. We appreciate your concern regarding the use of a multiple-choice format for our dataset evaluation. We would like to address your points as follows:
>
> 1. **Adherence to Established Evaluation Methods**: We have followed the prevalent evaluation methodologies used in the field, such as those employed in MMLU[1] and MMMU[2]. These methods utilize multiple-choice questions to objectively assess model performance. We believe that the ability of models to follow instructions is an important aspect of their overall capability.
>
> 2. **Robust Answer Extraction**: To mitigate the issue of models understanding the question but failing to output the correct option, we have implemented a robust answer extraction method. This method is designed to accurately extract the answer from the model's output. The specific extraction technique we used can be found in our evaluation script (https://github.com/II-Bench/II-Bench/blob/main/eval.py). We will add more details about our extraction method in the next version of our paper.
>
> 3. **Model Output Analysis**: In our experiments, we observed that the majority of models were able to output the final option. We have provided a detailed statistical analysis of the model outputs in **Appendix E**.
>
> We hope this addresses your concerns and clarifies our methodology. Thank you once again for your valuable feedback.
>
> [1] Measuring Massive Multitask Language Understanding. *https://arxiv.org/abs/2009.03300*
>
> [2] MMMU: A Massive Multi-discipline Multimodal Understanding and Reasoning Benchmark for Expert AGI. *https://arxiv.org/abs/2311.16502*
>
> > **Concern2**: The dataset is challenging to expand, as each piece of data requires careful manual creation, consuming significant time and effort.
>
> Thank you for your insightful feedback. We fully acknowledge your point regarding the challenges associated with expanding the dataset due to the significant time and effort required for careful manual creation. From another perspective, this indeed underscores the importance and value of our dataset. To ensure the highest quality, we dedicated a team of 15 annotators and quality inspectors who worked diligently over a period of four months to annotate and refine the dataset. This rigorous process was essential to create a reliable and challenging benchmark for image implication.
>
> Our goal is to provide the open-source community with a high-quality dataset that can serve as a robust benchmark. Our experimental results indicate a substantial gap between the performance of current MLLMs and human, suggesting that there is no immediate need to expand the dataset in the short term. Looking ahead, we are committed to exploring more automated methods to expand our dataset, which will be a focus of our future work.
>
> We believe that our benchmark will significantly contribute to advancing MLLMs towards higher-order cognitive understanding, and potentially even Artificial General Intelligence (AGI).
>
> Thank you once again for your valuable feedback.

---

> ### Author Response · Authors · 2024-08-17
> **Response to Reviewer SyRn(2/n)**
>
> > **Concern3**: The data comes from well-known illustration websites, which might have been included in some models' training sets, posing a risk of data leakage and leading to unfair evaluations.
>
>
> Thank you for your insightful feedback regarding the potential risk of data leakage from using well-known illustration websites. We understand the importance of ensuring fair evaluations and have made efforts to mitigate this risk:
>
> 1. **Recent Data Collection**: We have made an effort to collect data from a recent time, which aims to reduce the likelihood that these specific images are included in the training sets of existing models.
>
> 2. **Brand new, Manually Constructed Questions**: The questions we posed are entirely new and manually constructed, which are not present in any model's training set. Even if the images themselves are part of the training data, it is unlikely to correctly answer our high-level implication questions.
>
> 3. **Multimodal Data Leakage Assessment**: To address concerns about multimodal data leakage, we conduct evaluations where models are tested without providing the corresponding images. The experimental results are shown in the table below.
>
> | Model | Mode | Accuarcy |
> | --- | --- | --- |
> | deepseek-vl-7b-chat | MLLM | 60.3 |
> | deepseek-vl-7b-chat | text-only | 36.8(**-23.5**) |
> | Qwen-VL-Chat | MLLM | 53.4 |
> | Qwen-VL-Chat | text-only | 38.0(**-15.4**) |
> | idefics2-8b | MLLM | 67.7 |
> | idefics2-8b | text-only | 47.8(**-19.9**) |
>
> The results indicate that models perform badly without the images, suggesting that the models do not possess pre-existing knowledge of the specific questions and images, thereby indicating that there is no risk of data leakage.
>
> These measures collectively help ensure that our evaluations are fair and that the risk of data leakage is minimized.
>
> > **Concern4**: Each question matches six options, resulting in overly long contexts. Some models have low capabilities for handling long contexts, leading to unfairness during the evaluation process.
>
> Thank you for your insightful feedback. We appreciate your concern regarding the potential issue of overly long contexts due to each question matching six options, which might affect the evaluation fairness for models with lower capabilities in handling long contexts.
>
> We would like to address this concern with the following points:
>
> 1. **Option Length and input length**: As demonstrated in **`Table 1`** of our paper, the average length of our options is **`only 14 words`**, which indicates that the options themselves are not significantly contribute to overly long contexts. In most cases, the total input length **remains within the window size of the models**.
>
> 2. To further substantiate the fairness of our evaluation, we have provided detailed statistics on model outputs in **`Appendix E`**. These statistics show that under most settings, the models are able to **successfully generate final answer**, suggesting that the input lengths are **generally within the models' window size**, thereby supporting the fairness of our comparisons.
>
> We hope this clarifies our approach and addresses your concern. We are committed to ensuring a fair and rigorous evaluation process and appreciate your feedback in helping us achieve this goal.
>
> Finally, we are very grateful for your careful review and valuable suggestions. If you feel that our newly added responses have addressed your concerns and questions, we sincerely hope you might consider raising your score. Please feel free to comment if there are any further confusions and concerns, we will respond as soon as possible. We are committed to continuing our contributions to the LLM and open-source communities.

---

> ### Author Rebuttal · Authors · 2024-08-17
>
> # Rebuttal to Reviewer SyRn
> We sincerely thank you for your time and constructive suggestions. The insights provided have been instrumental in refining our work. We would like to provide our detailed responses and explanations to the issues raised.
>
> 1. Regarding `Concern 1.`
>
>
> Thank you for your insightful feedback. We appreciate your concern regarding the use of a multiple-choice format for our dataset evaluation. We would like to address your points as follows:
>
> - **Adherence to Established Evaluation Methods**: Our dataset employs the multiple-choice format, which is widely used in prominent benchmarks such as ARC, MMLU, Helleswag, MMMU, CMMMU, CEVAL and so on. We have adopted a similar answer extraction method to ensure consistency and robustness in our evaluations.
>
> - **Robust Answer Extraction**: To mitigate the issue of models understanding the question but failing to output the correct option, we have implemented a robust answer extraction method. Specifically, our extraction method involves the following steps:
>     - Pattern Matching: We first search for the presence of option labels within the model's output using specific patterns.
>     - Frequency Analysis: If multiple option labels are found, we count their occurrences and select the most frequently occurring label.
> We will add more details about our extraction method in the next version of our paper.
>
> - **Model Output Analysis**: In our experiments, we observed that the majority of models were able to output the final option. We have provided a detailed statistical analysis of the model outputs in **`Appendix E`**.
>
> We hope this addresses your concerns and clarifies our methodology. Thank you once again for your valuable feedback.
>
> 2. Regarding `Concern 2.`
>
> Thank you for your insightful feedback. We fully acknowledge your point regarding the challenges associated with expanding the dataset due to the significant time and effort required for careful manual creation. From another perspective, this indeed underscores the importance and value of our dataset. To ensure the highest quality, we dedicated a team of **15** annotators and quality inspectors who worked diligently over a period of four months to annotate and refine the dataset. This rigorous process was essential to create a reliable and challenging benchmark for image implication.
>
> Our goal is to provide the open-source community with a high-quality dataset that can serve as a robust benchmark. Our experimental results indicate a substantial gap between the performance of current MLLMs and human, suggesting that there is no immediate need to expand the dataset in the short term. Looking ahead, we are committed to exploring more automated methods to expand our dataset, which will be a focus of our future work.
>
> We believe that our benchmark will significantly contribute to advancing MLLMs towards higher-order cognitive understanding, and potentially even Artificial General Intelligence (AGI).
>
> 3. Regarding `Concern 3.`
>
> Thank you for your insightful feedback regarding the potential risk of data leakage from using well-known illustration websites. We understand the importance of ensuring fair evaluations and have made efforts to mitigate this risk:
>
> - **Recent Data Collection**: We have made an effort to collect data from a recent time, which aims to reduce the likelihood that these specific images are included in the training sets of existing models.
>
> - **Brand new, Manually Constructed Questions**: The questions we posed are entirely new and manually constructed, which are not present in any model's training set. Even if the images themselves are part of the training data, it is unlikely to correctly answer our high-level implication questions.
>
> - **Multimodal Data Leakage Assessment**: To address concerns about multimodal data leakage, we conduct evaluations where models are tested without providing the corresponding images. The experimental results are shown in the pdf file.
>
> The results indicate that models perform badly without the images, suggesting that the models do not possess pre-existing knowledge of the specific questions and images, thereby indicating that there is no risk of data leakage.
>
> These measures collectively help ensure that our evaluations are fair and that the risk of data leakage is minimized.
>
> 4. Regarding `Concern 4.`
>
> Thank you for your insightful feedback. We appreciate your concern regarding the potential issue of overly long contexts due to each question matching six options, which might affect the evaluation fairness for models with lower capabilities in handling long contexts.
>
> We would like to address this concern with the following points:
>
> 1. **Option Length and input length**: As demonstrated in **`Table 1`** of our paper, the average length of our options is **`only 14 words`**, which indicates that the options themselves are not significantly contribute to overly long contexts. In most cases, the total input length **remains within the window size of the models**.
>
> 2. To further substantiate the fairness of our evaluation, we have provided detailed statistics on model outputs in **`Appendix E`**. These statistics show that under most settings, the models are able to **successfully generate final answer**, suggesting that the input lengths are **generally within the models' window size**, thereby supporting the fairness of our comparisons.
>
> We hope this clarifies our approach and addresses your concern. We are committed to ensuring a fair and rigorous evaluation process and appreciate your feedback in helping us achieve this goal.
>
> Finally, we are very grateful for your careful review and valuable suggestions. Please feel free to comment if there are any further confusions and concerns, we will respond as soon as possible. We are committed to continuing our contributions to the LLM and open-source communities.

---

> ### Author Response · Authors · 2024-08-29
> **Further Explanation on Data Leakage and Multiple-Choice Format(3/n)**
>
> 1. Regarding `The data comes from well-known illustration websites, which might have been included in some models' training sets, posing a risk of data leakage and leading to unfair evaluations.`
>
> Thank you for your insightful feedback regarding the potential risk of data leakage from using well-known illustration websites. We understand the importance of ensuring fair evaluations and have made efforts to address this concern.
>
> We conduct a more comprehensive assessment of multimodal data leakage by comparing the performance of several models with and without image inputs. The experimental results are as follows:
>
> | Model | Mode | Accuarcy |
> | --- | --- | --- |
> | deepseek-vl-7b-chat | MLLM | 60.3 |
> | deepseek-vl-7b-chat | text-only | 36.8(**-23.5**) |
> | Qwen-VL-Chat | MLLM | 53.4 |
> | Qwen-VL-Chat | text-only | 38.0(**-15.4**) |
> | idefics2-8b | MLLM | 67.7 |
> | idefics2-8b | text-only | 47.8(**-19.9**) |
> | Mantis-8B-siglip-Llama3 | MLLM | 57.5 |
> | Mantis-8B-siglip-Llama3 | text-only | 43.3(**-14.2**) |
> | Yi-VL-6B-Chat | MLLM | 61.3 |
> | Yi-VL-6B-Chat | text-only | 44.9(**-16.4**) |
> | Yi-VL-34B-Chat | MLLM | 67.9 |
> | Yi-VL-34B-Chat | text-only | 52.2(**-15.7**) |
>
> The results indicate that models perform poorly without the images, suggesting that they do not possess pre-existing knowledge of the specific questions and images. This finding supports the conclusion that there is no significant risk of data leakage.
>
> These measures collectively help ensure that our evaluations are fair and that the risk of data leakage is minimized. We have made every effort to avoid data leakage and are committed to providing a detailed explanation of these steps in the next version of our paper.
>
> 2. Regarding `The dataset is created in a multiple-choice format, which in some cases can hinder evaluation, as models may understand the question and provide the correct answer but fail to output the correct option.`
>
> Thank you for your insightful feedback regarding the evaluation format of our dataset. We understand your concern about the potential limitations of using a multiple-choice format.
>
> Our dataset employs the multiple-choice format, which is widely used in prominent benchmarks such as ARC[1], MMLU[2], Helleswag[3], MMMU[4], CMMMU[5], CEVAL[6] and so on. We have adopted a similar answer extraction method to ensure consistency and robustness in our evaluations.
>
> Specifically, our extraction method involves the following steps:
>
>  - **Pattern Matching**: We first search for the presence of option labels (e.g., A, B, C) within the model's output using specific patterns.
>  - **Frequency Analysis**: If multiple option labels are found, we count their occurrences and select the most frequently occurring label.
>
> This method ensures that the most relevant option label is selected, thereby mitigating the risk of incorrect option extraction due to minor variations in the model's output.
>
> We acknowledge the importance of providing a detailed explanation of our answer extraction method. Therefore, we will include a comprehensive description of this method in our next paper to enhance clarity and transparency. For more details, you can refer to `https://github.com/II-Bench/II-Bench/blob/main/eval.py`.
>
> As the deadline is approaching, we look forward to your prompt response so that we can address your concerns in a timely manner.
>
>
> Thank you once again for your valuable feedback. We appreciate your efforts in helping us improve our work.
>
> [1] Think you have Solved Question Answering? Try ARC, the AI2 Reasoning Challenge. *https://arxiv.org/abs/1803.05457*
>
> [2] Measuring Massive Multitask Language Understanding. *https://arxiv.org/abs/2009.03300*
>
> [3] HellaSwag: Can a Machine Really Finish Your Sentence? *https://arxiv.org/abs/1905.07830*
>
> [4] MMMU: A Massive Multi-discipline Multimodal Understanding and Reasoning Benchmark for Expert AGI. *https://arxiv.org/abs/2311.16502*
>
> [5] CMMMU: A Chinese Massive Multi-discipline Multimodal Understanding Benchmark. *https://arxiv.org/abs/2401.11944*
>
> [6] CEval: A Benchmark for Evaluating Counterfactual Text Generation. *https://arxiv.org/abs/2404.17475*

---

> ### Author Response · Authors · 2024-08-30
> **Kindly Reminder to Reviewer SyRn**
>
> Dear Reviewer `SyRn`:
>
> Sorry to bother you. Thank you once again for dedicating your valuable time to reviewing our paper. We have carefully addressed all your concerns in detail and hope that you find the response satisfactory. We sincerely hope you can participate in the Reviewer-Author discussion, as it is crucial for us to fully address your concerns and improve our paper during this period. As the deadline approaches, we kindly request your feedback on our rebuttal. We are eager to engage in further discussion and address any additional concerns you may have. We sincerely appreciate your constructive suggestions and believe that the additional experiments, analysis, and explanations significantly improve the quality of our submission. We truly hope you might consider raising your score. Thank you!
>
> Best Regards, Authors

---

### Official Review · Reviewer_472N · 2024-07-26
**a benchmark for high-order perception**

**Rating:** 5
**Confidence:** 5
**Correctness:** Yes
**Clarity:** Yes

**Review:**

**Summary Of Strengths:**

- II-Bench is comprehensively described, covering various image types and domains.
- The whole work is complete, with fine-grained annotations such as data difficulty, data categorization, image sentiment and diverse evaluation settings such as zero-shot, COT, and few-shot.
- The study introduces the motivation clearly, emphasizing the gap in evaluating higher-order perceptual capabilities of MLLMs.

**Summary Of Weaknesses:**
- Why is it necessary to evaluate the performance of high-order perception for AGI? There is little discussion about the relationship between image implication and AGI.
- For some questions in II-Bench, there may not be a standard answer (such as those related to art, memes, etc., where everyone has their perspective). For such questions with strong subjective elements, the multiple-choice format may not be suitable. If the incorrect options are set to be very different from the correct option, it may result in the questions being too easy.
- As many images and questions are too subjective, how to control the consistency of answers across different subjects? The authors are encouraged to provide more details about the annotation process.
- Although MLLMs perform worse than humans, the best-performing model Qwen-VL-Max achieves 74.8 accuracy already. The proposed benchmark is not challenging for MLLMs. Moreover, it is a bit surprising that many open-source MLLMs such as LLaVA-1.6-34B exceed GPT4V. Do the authors use GPT4V with high quality?
- What about the performance of LLMs on II-Bench?
- Why do most models perform worse when using CoT? Is it because the process of extracting options is not robust enough? The author should provide more details on the steps for extracting options here (whether there is a difference in extracting answers from CoT and non-CoT responses).

**Strengths:**

see Review

**Additional Feedback:**

see Review

**Documentation:**

Yes

**Limitations:**

Yes

**Opportunities For Improvement:**

see Review

**Relation To Prior Work:**

Yes

**Summary And Contributions:**

To improve the image implication understanding in multimodal large language models (MLLMs), the II-Bench is proposed in this study. II-Bench is a comprehensive evaluation benchmark for MLLMs comprising 1,222 images across six domains: life, art, society, psychology, environment, and others, with a total of 1,434 questions. This study also evaluates 20 popular MLLMs on II-Bench, revealing a significant gap between the performance of MLLMs and humans. The best MLLM achieves 74.8% accuracy, while human accuracy averages 90%, peaking at 98%. The study highlights the limitations of MLLMs in understanding high-level semantics and capturing image details and the enhancement of model accuracy when image sentiment polarity hints are incorporated into the prompts.

---

> ### Author Response · Authors · 2024-08-17
> **Response to Reviewer 472N(1/n)**
>
> We sincerely thank you for your time and constructive suggestions. The insights provided have been instrumental in refining our work. We would like to provide our detailed responses and explanations to the issues raised.
>
> > **Concern1**: Why is it necessary to evaluate the performance of high-order perception for AGI? There is little discussion about the relationship between image implication and AGI.
>
>
> Thank you for your insightful question regarding the necessity of evaluating high-order perception for AGI and the relationship between image implication and AGI. We apologize for any confusion caused by the lack of detailed explanation in our initial submission. We appreciate the opportunity to clarify this important aspect of our work.
>
> The evaluation of high-order perception is crucial for AGI because it encompasses the ability to understand complex, non-literal aspects of visual information, much like humans do. This includes recognizing and processing implications, emotional cues, synesthesia or other cognitive features that go beyond mere object recognition or factual knowledge extraction. Our work draws on the critique of universalism, which highlights that important epistemological concepts are often rooted in metaphorical thinking[1,2]. In the context of AGI, this implies that a machine's understanding of the world should also incorporate these metaphorical dimensions to achieve a more human-like comprehension, which fosters a deeper, more nuanced understanding. For AGI, this means that the ability to understand implications is essential for interacting with humans.
>
> Our proposed II-Bench specifically addresses this gap by measuring the ability of MLLMs to capture and understand high-order, non-literal associations or implications based on visual input. Unlike existing metrics that focus on factual knowledge and detailed image analysis, II-Bench evaluates the human-like understanding of visual information, which includes cognitive features such as emotion, subject substitution, and synaesthesia. By incorporating these dimensions, II-Bench provides a more comprehensive assessment of an MLLM's progress towards AGI, ensuring that it not only processes visual data accurately but also interprets it in a way that aligns with human cognitive and perceptual experiences.
>
> We hope this explanation clarifies the importance of evaluating high-order perception for AGI and the role of image implication in this context. Thank you for your valuable feedback, and we look forward to any further questions or comments you may have.
>
> [1] *Metaphor and Continental Philosophy: From Kant to Derrida, New York: Routledge, 2007.*
>
> [2] *Heidegger and Derrida on Philosophy and Metaphor, Amherst, New York: Humanity Books, 2000.*

---

> > ### Author Response · Authors · 2024-08-29
> > **Further Explanation on the relationship between image implication and AGI.**
> >
> > 1. Regarding `Why is it necessary to evaluate the performance of high-order perception for AGI? There is little discussion about the relationship between image implication and AGI.`
> >
> > Thank you for your insightful question. The necessity to evaluate the performance of high-order perception for AGI stems from the fundamental nature of human intelligence, which encompasses not only the ability to recognize objects and their attributes but also to understand and interpret the deeper meanings and implications.
> >
> > An image implication refer to the underlying meanings that extend beyond the superficial interpretation, probing into the essence of the image[1]. Recent studies[2] have shown that understanding the implications of images is a manifestation of high-level human intelligence, serving as an important means of exploration from perceptual intelligence to cognitive intelligence.
> >
> > However, previous efforts in visual understanding have mainly focused on surface-level aspects of images, such as object attributes and relationship reasoning. These approaches do not fully capture the complexity and depth of human visual cognition. To address these limitations and fill the current research gap, we introduce II-Bench, a benchmark designed to evaluate the model's higher-order perception of images, which is crucial for advancing AGI towards more sophisticated and human-like cognitive abilities.
> >
> > We hope this clarifies the importance of evaluating high-order perception for AGI and the relationship between image implication and AGI. Thank you for your consideration.
> >
> > [1] Roland Barthes. 1968. Elements of semiology. Macmillan.
> >
> > [2] Can Large Multimodal Models Uncover Deep Semantics Behind Images? *https://arxiv.org/abs/2402.11281*

---

> ### Author Response · Authors · 2024-08-17
> **Response to Reviewer 472N(2/n)**
>
> > **Concern2**: For some questions in II-Bench, there may not be a standard answer (such as those related to art, memes, etc., where everyone has their perspective). For such questions with strong subjective elements, the multiple-choice format may not be suitable. If the incorrect options are set to be very different from the correct option, it may result in the questions being too easy.
>
> Thank you for your insightful feedback regarding the potential ease of questions due to the distinctiveness of incorrect options. We apologize for not providing a detailed explanation of our data annotation process in the paper, which may have led to some confusion. We understand the importance of maintaining an appropriate level of challenge in our benchmark.
>
> To ensure the quality and difficulty of our distractor options, we have implemented a rigorous multi-step process. Initially, each set of options undergoes multiple rounds of meticulous manual annotation，which is designed to ensure that the distractors are sufficiently challenging and not easily distinguishable from the correct option.
>
> Following the annotation, we conduct a thorough quality assurance review. During this phase, if any question is found to be insufficiently challenging, it is revised accordingly. This iterative process continues until at least 4 out of 5 independent reviewers agree that the question is accurate, reasonable, and presents an adequate level of difficulty.
>
> We believe that this comprehensive approach effectively mitigates the risk of questions being too easy.
>
> > **Concern3**: As many images and questions are too subjective, how to control the consistency of answers across different subjects? The authors are encouraged to provide more details about the annotation process.
>
> Thank you for your insightful feedback. We acknowledge that some images and questions may indeed be subjective, but we have made efforts to mitigate the impact of this subjectivity on the consistency of answers.
>
> Specifically, each question in our dataset is annotated by at least five annotators. If their answers are not consistent, we either discard those questions or re-annotate them until at least four out of the five annotators reach a consensus.  Therefore, images with deep semantics but significant controversy will not be included. This quality control process helps to reduce the influence of individual subjectivity and ensures greater consistency across different annotators.
>
> We appreciate your suggestion and will provide a more detailed explanation of our annotation process in the next version of our paper to further clarify how we address this issue.

---

> ### Author Response · Authors · 2024-08-17
> **Response to Reviewer 472N(3/n)**
>
> > **Concern4**: Although MLLMs perform worse than humans, the best-performing model Qwen-VL-Max achieves 74.8 accuracy already. The proposed benchmark is not challenging for MLLMs. Moreover, it is a bit surprising that many open-source MLLMs such as LLaVA-1.6-34B exceed GPT4V. Do the authors use GPT4V with high quality?
>
>
> **Regarding `the benchmark difficulty`**:
> We appreciate your observation regarding the benchmark's difficulty. To address this, we have already conducted a series of experiments to explore the performance of MLLMs across questions of varying difficulty levels in **`Table 6`**. The results show that MLLMs perform relatively well on simple and medium-difficulty questions but still lag behind human performance. Notably, their performance significantly drops on difficult questions, where the gap between MLLMs and human performance remains substantial. Therefore, we believe our benchmark continues to present a significant challenge for MLLMs.
>
> **Regarding `the performance of open-source models compared to GPT4V`**:
> 1. In our experiments, we used GPT4V and GPT4o under completely identical conditions to ensure a fair comparison. The results indicate that GPT4o outperforms GPT4V.
> 2. Additionally, it is noteworthy that some great open-source models surpass GPT4V. This phenomenon suggests two important points:
>
> - Not all closed-source models necessarily outperform open-source models.
> - Open-source models are progressively closing the performance gap with closed-source models, which is an encouraging sign for the continued improvement of open-source MLLMs.
>
> > **Concern5**: What about the performance of LLMs on II-Bench?
>
> Thank you for your insightful question regarding the performance of large language models (LLMs) on the II-Bench tasks.
>
> We conducted a series of experiments to evaluate the performance of LLMs on II-Bench. The results of these experiments are summarized in the table below:
>
> | Model | Mode | Accuarcy |
> | --- | --- | --- |
> | vicuna-7b-v1.5 | LLM | 10.8 |
> | Llama-2-7b-chat-hf | LLM | 34.6 |
> | MAP-Neo-7b-Instruct-v0.1 | LLM | 36.1 |
> | Meta-Llama3-8b-Instruct | LLM | 43.7 |
> | Qwen2-7b-Instruct | LLM | 44.7 |
> | Yi-1.5-6b-Chat | LLM | 44.9 |
> | deepseek-llm-7b-chat | LLM | 36.8 |
> | deepseek-vl-7b-chat | MLLM | 60.3 |
> | deepseek-vl-7b-chat | text-only | 36.8(**-23.5**) |
> | Qwen-VL-Chat | MLLM | 53.4 |
> | Qwen-VL-Chat | text-only | 38.0(**-15.4**) |
> | idefics2-8b | MLLM | 67.7 |
> | idefics2-8b | text-only | 47.8(**-19.9**) |
>
>
> 1. We test the performance of current popular LLMs on II-Bench and find that the performance of LLMs is far inferior to that of MLLMs. The reason some models perform better than random selection is that some questions may contain descriptions of image details, leading to accuracy higher than random selection. Nevertheless, the performance of LLMs is still very poor.
> 2. We also test MLLMs by providing only the questions and options as input, without the images. The results show that the performance of the MLLM significantly decreases when the images are not provided.
>
> Both experimental phenomena indicate that the information in the images is indeed necessary for the model to answer correctly. We believe these results emphasize the need for multimodal approaches that integrate both textual and visual information to effectively tackle the challenges presented by II-Bench.

---

> ### Author Response · Authors · 2024-08-17
> **Response to Reviewer 472N(4/n)**
>
> > **Concern6**: Why do most models perform worse when using CoT? Is it because the process of extracting options is not robust enough? The author should provide more details on the steps for extracting options here (whether there is a difference in extracting answers from CoT and non-CoT responses).
>
>
> Thank you for your insightful comments and questions regarding our work. We apologize for any confusion caused by the lack of detailed explanation in the original. We appreciate the opportunity to clarify and provide more details.
>
> **Regarding `the performance of models with CoT`**:
>
> 1. **Adherence to Format**: One primary reason for the observed decline in performance with Chain-of-Thought (CoT) prompting is that many models fail to adhere to the required format. Specifically, CoT prompts require models to output both an analysis and a final answer in a fixed format. However, many models only provide the analysis without concluding with the final answer. This issue is highlighted in **`Appendix E`**, where we present statistics on model outputs. The lack of a final answer in CoT responses leads to extraction failures, which negatively impacts performance metrics.
>
> 2. **Effectiveness of CoT**: Another contributing factor is that CoT prompting **does not universally enhance performance across all types of tasks**. In evaluations such as MMLU[1], C-Eval[2] and E-Eval[3], where the primary focus is not on logical reasoning or mathematical problem-solving, Chain-of-Thought (CoT) prompting often does not lead to better results. The same phenomenon is also observed in our experiments. This contrasts with tasks that inherently benefit from step-by-step reasoning, where CoT can be more effective.
>
> [1] Scaling Instruction-Finetuned Language Models. *https://arxiv.org/abs/2210.11416*
>
> [2] C-EVAL: A Multi-Level Multi-Discipline Chinese Evaluation Suite for Foundation Models. *https://arxiv.org/abs/2305.08322*
>
> [3] E-EVAL: A Comprehensive Chinese K-12 Education Evaluation Benchmark for Large Language Models. *https://arxiv.org/abs/2401.15927*
>
> **Regarding `the extraction of options`**:
>
> We use the same extraction method for both CoT and non-CoT responses to ensure consistency. Specifically, we employ regular expressions to extract the final answers from the model outputs. The detailed extraction process can be found in our evaluation script available at https://github.com/II-Bench/II-Bench/blob/main/eval.py.
>
> In the next version of our paper, we will include a more detailed explanation of the answer extraction steps to provide greater clarity on this process.
>
> Finally, we are very grateful for your careful review and valuable suggestions. If you feel that our newly added responses have addressed your concerns and questions, we sincerely hope you might consider raising your score. Please feel free to comment if there are any further confusions and concerns, we will respond as soon as possible. We are committed to continuing our contributions to the LLM and open-source communities.

---

> ### Author Rebuttal · Authors · 2024-08-17
>
> We sincerely thank you for your time and constructive suggestions. The insights provided have been instrumental in refining our work. We would like to provide our detailed responses and explanations to the issues raised.
> 1. Regarding `Concern 1.`
>
> Thank you for your insightful question. The evaluation of high-order perception is crucial for AGI because it encompasses the ability to understand complex, non-literal aspects of visual information, much like humans do. This includes recognizing and processing implications, emotional cues, synesthesia or other cognitive features that go beyond mere object recognition or factual knowledge extraction. In the context of AGI, this implies that a machine's understanding of the world should also incorporate these metaphorical dimensions to achieve a more human-like comprehension, which fosters a deeper, more nuanced understanding. For AGI, this means that the ability to understand implications is essential for interacting with humans.
>
> We introduce II-Bench, a benchmark designed to evaluate the model's higher-order perception of images, which is crucial for advancing AGI towards more sophisticated and human-like cognitive abilities.
>
> 2. Regarding `Concern 2.`
>
> To ensure the quality and difficulty of our distractor options, we have implemented a rigorous multi-step process. Initially, each set of options undergoes multiple rounds of meticulous manual annotation，which is designed to ensure that the distractors are sufficiently challenging and not easily distinguishable from the correct option.
>
> During quality review phase, if any question is found to be insufficiently challenging, it is revised accordingly. This iterative process continues until at least 4 out of 5 independent reviewers agree that the question is accurate, reasonable, and presents an adequate level of difficulty.
>
> We believe that this comprehensive approach effectively mitigates the risk of questions being too easy.
>
> 3. Regarding `Concern 3.`
>
> We acknowledge that some images and questions may indeed be subjective, but we have made efforts to mitigate the impact of this subjectivity on the consistency of answers.
>
> Specifically, each question in our dataset is annotated by at least five annotators. If their answers are not consistent, we either discard those questions or re-annotate them until at least four out of the five annotators reach a consensus. Therefore, images with deep semantics but significant controversy will not be included. This quality control process helps to reduce the influence of individual subjectivity and ensures greater consistency across different annotators.
>
> 4. Regarding `Concern 4.`
>
> - We appreciate your observation regarding the benchmark's difficulty. To address this, we have already conducted a series of experiments to explore the performance of MLLMs across questions of varying difficulty levels in **`Table 6`**. The results show that MLLMs perform relatively well on simple and medium-difficulty questions but still lag behind human performance. Notably, their performance significantly drops on difficult questions, where the gap between MLLMs and human performance remains substantial. Therefore, we believe our benchmark continues to present a significant challenge for MLLMs.
> - In our experiments, we used GPT4V and GPT4o under completely identical conditions to ensure a fair comparison. Additionally, it is noteworthy that some great open-source models surpass GPT4V. This phenomenon suggests two important points: (1)Not all closed-source models necessarily outperform open-source models. (2)Open-source models are progressively closing the performance gap with closed-source models.
>
> 5. Regarding `Concern 5.`
>
> We conduct a more comprehensive assessment of several LLMs with and without image. The result is in the PDF file. We test the performance of current popular LLMs on II-Bench and find that their performance is significantly inferior to that of MLLMs. We also test MLLMs by providing only the questions and options as input, without the images. The results show that the performance of the MLLMs significantly decreases when the images are not provided. Both experimental phenomena indicate that **the information in the images is indeed necessary** for the model to answer correctly.
>
> 6. Regarding `Concern 6.`
>
> - **Adherence to Format**: One primary reason for the observed decline in performance with CoT prompting is that many models fail to adhere to the required format. Specifically, CoT prompts require models to output both an analysis and a final answer in a fixed format. However, many models only provide the analysis without concluding with the final answer. This issue is highlighted in **`Appendix E`**, where we present detailed statistics on model outputs. The lack of a final answer in CoT responses leads to extraction failures, which negatively impacts performance metrics.
>
> - **Effectiveness of CoT**: Another factor is that CoT prompting does not universally enhance performance across all types of tasks. In evaluations such as MMLU, C-Eval and E-Eval, where the primary focus is not on logical reasoning or mathematical problem-solving, **CoT prompting often does not lead to better results**. The same phenomenon is also observed in our experiments. This contrasts with tasks that inherently benefit from step-by-step reasoning, where CoT can be more effective.
>
> - **Option Extraction**: We use the same extraction method for both CoT and non-CoT responses to ensure consistency. Specifically, our extraction method involves the following steps:
>      - **Pattern Matching**: We first search for the presence of option labels within the model's output using specific patterns.
>      - **Frequency Analysis**: If multiple option labels are found, we count their occurrences and select the most frequently occurring label.
>
> For more details, you can refer to `https://github.com/II-Bench/II-Bench/blob/main/eval.py`.
>
> Finally, we are very grateful for your careful review.

---

> ### Author Response · Authors · 2024-08-29
> **Further Explanation on LLMs Performance and Option Extraction(5/n)**
>
> 1. Regarding `the performance of LLMs on II-Bench.`
>
> We conduct a more comprehensive assessment of several LLMs with and without image. The results are as follows:
>
> | Model | Mode | Accuarcy |
> | --- | --- | --- |
> | vicuna-7b-v1.5 | LLM | 10.8 |
> | Llama-2-7b-chat-hf | LLM | 34.6 |
> | MAP-Neo-7b-Instruct-v0.1 | LLM | 36.1 |
> | Meta-Llama3-8b-Instruct | LLM | 43.7 |
> | Qwen2-7b-Instruct | LLM | 44.7 |
> | Yi-1.5-6b-Chat | LLM | 44.9 |
> | deepseek-llm-7b-chat | LLM | 36.8 |
> | deepseek-vl-7b-chat | MLLM | 60.3 |
> | deepseek-vl-7b-chat | text-only | 36.8(**-23.5**) |
> | Qwen-VL-Chat | MLLM | 53.4 |
> | Qwen-VL-Chat | text-only | 38.0(**-15.4**) |
> | idefics2-8b | MLLM | 67.7 |
> | idefics2-8b | text-only | 47.8(**-19.9**) |
> | Mantis-8B-siglip-Llama3 | MLLM | 57.5 |
> | Mantis-8B-siglip-Llama3 | text-only | 43.3(**-14.2**) |
> | Yi-VL-6B-Chat | MLLM | 61.3 |
> | Yi-VL-6B-Chat | text-only | 44.9(**-16.4**) |
> | Yi-VL-34B-Chat | MLLM | 67.9 |
> | Yi-VL-34B-Chat | text-only | 52.2(**-15.7**) |
>
> We test the performance of current popular LLMs on II-Bench and found that their performance is significantly inferior to that of MLLMs. The reason some models perform better than random selection is that some questions may contain descriptions of image details, leading to accuracy higher than random selection. Nevertheless, the performance of LLMs remains very poor.
>
> We also test MLLMs by providing only the questions and options as input, without the images. The results show that the performance of the MLLMs significantly decreases when the images are not provided.
>
> Both experimental phenomena indicate that **the information in the images is indeed necessary** for the model to answer correctly. We believe these results emphasize the need for multimodal approaches that integrate **both textual and visual information** to effectively tackle the challenges presented by II-Bench.
>
> 2. Regarding `the steps for extracting options.`
> Our benchmark employs the multiple-choice format, which is widely used in prominent benchmarks such as ARC[1], MMLU[2], Helleswag[3], MMMU[4], CMMMU[5], CEVAL[6] and so on. We have adopted a similar answer extraction method to ensure consistency and robustness in our evaluations.
>
> Specifically, our extraction method involves the following steps:
>  - **Pattern Matching**: We first search for the presence of option labels (e.g., A, B, C) within the model's output using specific patterns.
>  - **Frequency Analysis**: If multiple option labels are found, we count their occurrences and select the most frequently occurring label.
>
> This method ensures that the most relevant option label is selected, thereby mitigating the risk of incorrect option extraction due to minor variations in the model's output.
>
> We acknowledge the importance of providing a detailed explanation of our answer extraction method. Therefore, we will include a comprehensive description of this method in next version of our paper to enhance clarity and transparency. For more details, you can refer to `https://github.com/II-Bench/II-Bench/blob/main/eval.py`.
>
> As the deadline is approaching, we look forward to your prompt response so that we can address your concerns in a timely manner.
>
> Thank you once again for your valuable feedback. We appreciate your efforts in helping us improve our work.
>
> [1] Think you have Solved Question Answering? Try ARC, the AI2 Reasoning Challenge. *https://arxiv.org/abs/1803.05457*
>
> [2] Measuring Massive Multitask Language Understanding. *https://arxiv.org/abs/2009.03300*
>
> [3] HellaSwag: Can a Machine Really Finish Your Sentence? *https://arxiv.org/abs/1905.07830*
>
> [4] MMMU: A Massive Multi-discipline Multimodal Understanding and Reasoning Benchmark for Expert AGI. *https://arxiv.org/abs/2311.16502*
>
> [5] CMMMU: A Chinese Massive Multi-discipline Multimodal Understanding Benchmark. *https://arxiv.org/abs/2401.11944*
>
> [6] CEval: A Benchmark for Evaluating Counterfactual Text Generation. *https://arxiv.org/abs/2404.17475*

---

> ### Author Response · Authors · 2024-08-30
> **Kindly Reminder to Reviewer 472N**
>
> Dear Reviewer `427N`:
>
> Sorry to bother you. Thank you once again for dedicating your valuable time to reviewing our paper. We have carefully addressed all your concerns in detail and hope that you find the response satisfactory. We sincerely hope you can participate in the Reviewer-Author discussion, as it is crucial for us to fully address your concerns and improve our paper during this period. As the deadline approaches, we kindly request your feedback on our rebuttal. We are eager to engage in further discussion and address any additional concerns you may have. We sincerely appreciate your constructive suggestions and believe that the additional experiments, analysis, and explanations significantly improve the quality of our submission. We truly hope you might consider raising your score. Thank you!
>
> Best Regards, Authors

---

### Author Rebuttal · Authors · 2024-08-29

We sincerely thank the reviewers for their time and constructive feedback. The insights provided have been instrumental in refining our work. We appreciate the recognition of the key contributions of our paper:

- **II-Bench:** We introduce II-Bench, the first Image Implication Understanding Benchmark, which is very challenging for current Multimodal Large Language Models (MLLMs). This benchmark is designed to evaluate the higher-order perception capabilities of MLLMs, pushing the boundaries of what these models can achieve in terms of understanding and interpreting complex visual information.
- **Complete and Detailed Work:** The whole work is complete, with fine-grained annotations such as data difficulty, data categorization, image sentiment, and diverse evaluation settings such as zero-shot, chain-of-thought (COT), and few-shot. These detailed annotations and varied evaluation settings ensure a thorough and nuanced assessment of MLLMs' capabilities.
- **Comprehensive Experiments:** We have conducted extensive qualitative and quantitative experiments to demonstrate the effectiveness and efficiency of our proposed benchmark
- **Key Findings:** Our experiments reveal key findings that highlight the significant performance differences between humans and MLLMs. Additionally, we observe that closed-source models often outperform open-source ones, although the performance gap between the leading closed-source model and the leading open-source model is minimal.

Our aim with II-Bench is to evaluate MLLMs’ higher-order perception of images. We believe that II-Bench will inspire the community to create the next generation of MLLMs, propelling us further on the path toward sophisticated artificial general intelligence (AGI).

We have **addressed all concerns** raised, incorporating further experiments and clarifications, and believe the revisions have improved the quality and clarity of our paper. We would like to highlight the following points:

- We provided a detailed discussion on the relationship between higher-order perception abilities and AGI.
- We offered a detailed data annotation process to address issues of difficulty and subjectivity.
- We presented a robust answer extraction method that aligns with the current mainstream evaluation approaches.
- We provided detailed experiments to evaluate the performance of LLMs and discussed how to avoid data leakage issues.

Finally,  we firmly believe that our detailed responses can address all the reviewers' concerns and we are very grateful for your careful review and valuable suggestions. This work and these experiments have taken a lot of effort from our author team. If you feel that our newly added responses and results have addressed your concerns and questions, we sincerely hope you might consider raising your score. It means a lot to us and we would like to offer our sincerest thanks to you. Please feel free to comment if there are any further confusions and concerns, we will respond as soon as possible.

---

### Decision · Program_Chairs · 2024-09-26

**Decision:**

Accept (Poster)

**Comment:**

Recommendation: accept

The authors present II-Bench, a novel benchmark designed to evaluate the ability of VLM to understand metaphors in images. This work addresses an important gap in the current landscape of VLMs evaluation, focusing on complex semantic understanding that goes beyond simple object recognition or factual knowledge extraction.

Reviewers agree on these strenghts:
- This is the first benchmark of its kind, addressing a critical aspect of visual understanding that has been largely unexplored in previous MLLM evaluations.
- Comprehensive dataset: The benchmark includes a diverse range of images across multiple domains, with careful curation and human annotation processes to ensure quality and relevance.
- Various open source and close source VLMs are evaluated along with various prompting techniques. The results show a strong gap between human performance and the model performance, leaving space for progress.

the reviewers have noted a few weaknesses:

- Scalability challenges: The dataset's reliance on manual annotation makes it difficult to expand, which could limit its long-term applicability as models improve.

Despite these limitations, the authors have provided thorough and convincing responses to reviewer concerns (possible data leakage, multi-choice format, long context, subjectivity of the questions/answers) and I don’t believe these are deal-breaking concerns. Unfortunately, none of the reviewers have engaged in the discussion and as such, did not modify their initial score.

The potential impact of this benchmark on advancing VLMs capabilities towards more human-like visual understanding is significant and would benefit from the visibility of this conference.